# Why DDIM Hallucinates More Than DDPM:
# A Theoretical Analysis of Reverse Dynamics

**Muhammad H. Ashiq** [* 1]  **Samanyu Arora** [* 1]  **Abhinav N. Harish** [1]  **Ishaan Kharbanda** [1]  **Hung Yun Tseng** [1]
**Grigorios G. Chrysos** [1]

## Abstract

We theoretically study the hallucination phenomena in two canonical diffusion samplers: the stochastic Denoising Diffusion Probabilistic Model (DDPM) and the deterministic Denoising Diffusion Implicit Model (DDIM). We analyze the reverse ODE (DDIM) and SDE (DDPM) for a Gaussian mixture target, proving that after a critical time $\tau$, (a) DDIM can become stuck on the segment connecting the two nearest modes and (b) DDPM *stochasticity* helps it become unstuck from this region, thus avoiding hallucination. Our empirical validation verifies that DDPM has a significantly lower hallucination rate than DDIM when this region is entered. Building on our observations, we exhibit how using additional stochastic steps can help DDIM avoid hallucinations and offer new insights on how to design improved samplers.

## 1. Introduction

Despite significant progress, state-of-the-art diffusion models still exhibit hallucinations which do not abide by the structural, semantic constraints present in their training distribution. This includes distorted hands, implausible object boundaries, and temporal inconsistencies in video. Concretely, hallucinations can be characterized by generations which arise when the reverse diffusion trajectory is guided toward regions outside the set of constraints implied by the target distribution.

To address this issue, prior work has focused on a variety of different strategies. In the domain of text generation, Lu et al. (2025) identify a local generation bias in image diffusion models by highlighting their inability to model global

context in MNIST digits. Aithal et al. (2024) characterize hallucinations as mode interpolation in Gaussian mixture models and propose a metric for detecting interpolations. Triaridis et al. (2025) explore dynamic guidance as a means of mitigating hallucination. Although these works provide strategies to detect hallucinations, none of them explore *why* hallucinations arise by systematically studying the design and mechanics of the reverse diffusion process.

We focus on the distinction between two popular samplers: the stochastic Denoising Diffusion Probabilistic Models (DDPM) (Ho et al., 2020) and the deterministic Denoising Diffusion Implicit Models (DDIM) (Song et al., 2021a). These samplers are trained identically; however, they differ in the sampling strategy they employ during the reverse process. DDPM conducts an equal number of forward and reverse steps, matching the introduction of noise in the forward diffusion SDE. DDIM, on the other hand, does *not* include this noise in the reverse process. DDIM introduces step skipping through a non-Markovian reverse process, offering faster inference. We identify that removing the noise in the reverse process contributes significantly to hallucinations. This observation has been made empirically by prior works in the context of counting hallucinations (Fu et al., 2025). However, to the best of our knowledge, a solid theoretical characterization of this phenomenon remains absent. We fill this gap and address hallucination as a *structural artifact* of the reverse process. Concretely, we ask the following question:

*Can we formally characterize why DDIM hallucinates more than DDPM?*

To answer this question, we study the reverse Probability Flow (PF) ODE (DDIM) and the reverse Variance Preserving (VP) SDE (DDPM) when the data distribution is an $N$-mode Gaussian mixture[1]. This setup lends itself to rigorous theoretical analysis. Additionally, we study hallucinations formulated as mode interpolations: a phenomenon where a sampled diffusion trajectory ends up at the line connecting modes of the data distribution (Aithal et al., 2024).

---
[*]Equal contribution [1]University of Wisconsin-Madison, Madison, WI, USA. Correspondence to: Grigorios G. Chrysos <chrysos@wisc.edu>.

*Proceedings of the 43rd International Conference on Machine Learning*, Seoul, South Korea. PMLR 306, 2026. Copyright 2026 by the author(s).

---
[1]Under mild regularity conditions, the marginal densities are identical for DDIM and DDPM.

As demonstrated in Figure 1b, our analysis reveals that DDIM/DDPM convergence occurs in two phases: (i) an early convergence phase dominated by attraction to the nearest line segment between two data modes and (ii) a late convergence phase governing motion along the line connecting these two modes. We demonstrate that the final steps of a trajectory under the late convergence phase determine the outcome of sampling and are a crucial difference between these two samplers. Specifically, in this final phase, the noise component of DDPM helps it escape regions on this line segment where DDIM gets stuck, thus avoiding hallucinations. Our theory demonstrates that this is the foundational reason that DDPM has an edge over DDIM, *regardless of the number of reverse steps employed.*

These findings rule out several previously held beliefs on DDIM hallucinations. For example, although practitioners often skip DDIM steps and thus incur numerical error when sampling, we demonstrate that this does not explain the hallucination rate gap. We find empirically that the DDIM and DDPM hallucination rate gap persists under the same discretization, and our theoretical results are derived under zero numerical error. Additionally, Aithal et al. (2024) attribute mode interpolation to the smoothing of the score function due to approximation with a neural network. Our results, however, are derived under the *exact score function*, i.e, the expected score upon using infinite samples from the distribution. We find that, with the exact score, the mechanism causing mode interpolation can be cleanly characterized.

In summary, our contributions are as follows:

1. We rigorously study the source of DDIM hallucinations as observed in an $N$-mode Gaussian mixture, demonstrating that after a critical time $\tau$, DDIM trajectories converge to the nearest line segment joining two modes and then can become stuck near the midpoint, thus hallucinating. This is illustrated in Figure 1.

2. We leverage this to provide a theoretical justification for why DDIM hallucinates more than DDPM: the noise of DDPM can help it become unstuck from this hallucination region around the midpoint. This is illustrated in Figure 1c.

3. Empirically, we invalidate that the DDIM hallucination rate gap can be explained by step skipping and demonstrate that adding a few DDPM steps after DDIM converges near the midpoint neighborhood can help the trajectory escape, lowering hallucination rate.

Code for our experiments is available for reproducibility at https://github.com/diffusion-hallucination/ddim_vs_ddpm_hallucinations.

## 2. Problem Setting

**Notation**: Bolded lowercase letters $\boldsymbol{x} \in \mathbb{R}^{\varpi}$ denote vectors of dimension $\varpi$. $[N]$, for scalar $N \in \mathbb{N}$, denotes the integer subset $\{1, 2, ..., N\}$. For a function $f$, we use $\nabla_{\boldsymbol{x}} f$ to denote its gradient if $f : \mathbb{R}^{\varpi} \to \mathbb{R}$ and its Jacobian if $f : \mathbb{R}^{\varpi} \to \mathbb{R}^{o}$. We denote by $x_t : [0, T] \to \mathbb{R}$ and $\boldsymbol{x}_t : [0, T] \to \mathbb{R}^{\varpi}$ time-dependent scalar-valued and vector-valued functions, respectively, from scalar time $T$ to $0$ unless otherwise specified. We use $\dot{x}_t$ to denote the time derivative of $x_t$ and similarly use $\dot{\boldsymbol{x}}_t$ for $\boldsymbol{x}_t$; for all other derivatives, we use the notation $\frac{d}{dx} f(x)$. $\mathcal{N}(\boldsymbol{x}; \boldsymbol{\mu}, \sigma^2 \boldsymbol{I})$ is used to denote the probability density function of a $\varpi$-dimensional Gaussian centered at $\boldsymbol{\mu}$ with variance $\sigma^2 \boldsymbol{I}$. We write stochastic differential equations (SDE) as $d\boldsymbol{x}_t = b(t, \boldsymbol{x}_t) dt + \sigma(t, \boldsymbol{x}_t) d\boldsymbol{W}_t$ where $\boldsymbol{W}_t \in \mathbb{R}^{\varpi \times 1}$ is the standard Brownian motion. We provide a full list of notation in Sec. A and a symbol table in Sec. J.

**Diffusion Models**: The goal of diffusion models is to learn to transport a simple base distribution $p_{\text{base}}$ to a more complicated target distribution $p_{\text{data}}$. Throughout, we consider $p_{\text{base}}$ to be the standard Gaussian $\mathcal{N}(0, \boldsymbol{I})$ (Sohl-Dickstein et al., 2015). We first define the DDPM Variance Preserving (VP) forward SDE (Song et al., 2021b):

$$dx_t = -\tfrac{1}{2}\beta_t \, \boldsymbol{x}_t \, dt + \sqrt{\beta_t} \, d\boldsymbol{W}_t, \qquad (1)$$

where $\beta_t$ is a noise schedule and $\boldsymbol{W}_t$ is the standard Brownian motion. We then denote $\bar{\alpha}_t := \exp(-\int_0^t \beta(s)ds)$. We consider the standard setting where $\bar{\alpha}_t \in C^2$, $\bar{\alpha}_0 = 1$, and $\bar{\alpha}_T = \gamma$ for some $\gamma \approx 0$. By Anderson's theorem (Anderson, 1982), Eq. (1) can be reversed as:

$$d\boldsymbol{x}_t = -\frac{1}{2}\beta_t \left( \boldsymbol{x}_t + 2\nabla_{\boldsymbol{x}_t} \log p_t(\boldsymbol{x}_t) \right) dt + \sqrt{\beta_t} \, d\bar{\boldsymbol{W}}_t, \qquad (2)$$

for reversed Brownian motion $\bar{\boldsymbol{W}}_t$. Here, the marginals $p_t$ are given by convolving the target distribution $p_{\text{data}}$ with a standard Gaussian: $p_t(\boldsymbol{x}) = \int p_{\text{data}}(\boldsymbol{y})\mathcal{N}(\boldsymbol{x}; \sqrt{\bar{\alpha}_t}\boldsymbol{y}, (1 - \bar{\alpha}_t)\boldsymbol{I})d\boldsymbol{y}$. Since access to the true score $\nabla_{\boldsymbol{x}_t} \log p_t(\boldsymbol{x}_t)$ is usually unavailable in practice, one instead learns the score field $s_{\boldsymbol{\theta}}(\boldsymbol{x}_t)$. Then, discretizing Eq. (2) with $s_\theta$, using Euler-Maruyama's method, provides a stochastic algorithm to sample from target distribution $p_{\text{data}}$, beginning with a base instance $\boldsymbol{x}_T \sim \mathcal{N}(0, \boldsymbol{I})$.

For DDIM, we define the (reverse-time) probability flow ordinary differential equation (PF ODE):

$$\dot{\boldsymbol{x}}_t = -\frac{1}{2}\beta_t(\boldsymbol{x}_t + \nabla_{\boldsymbol{x}_t} \log p_t(\boldsymbol{x}_t)), \qquad (3)$$

where the marginals $p_t$ are the same as those in Eq. (2). Notice that Eq. (3) is exactly Eq. (2) without the noise term, up to a constant multiplication of the score. Similarly,

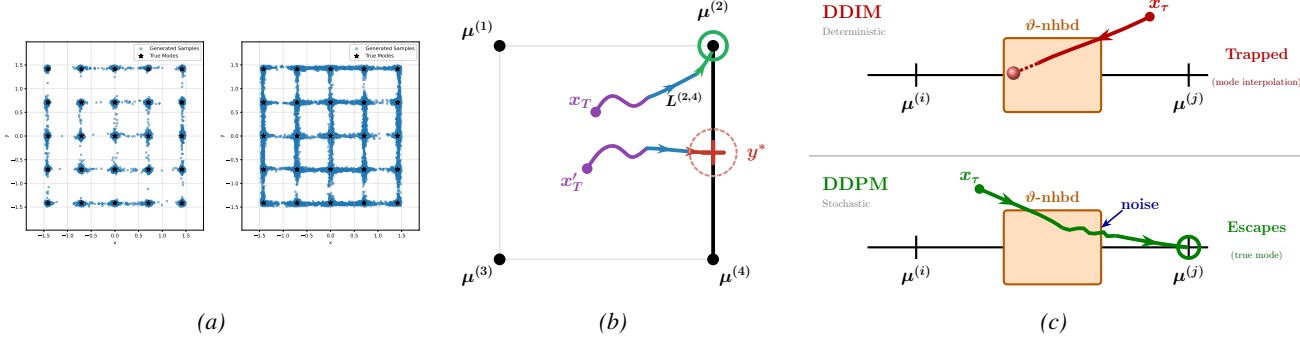

*Figure 1.* (a) In 100,000 generated samples for a 25-mode Gaussian mixture target, despite using the *same pretrained model*, DDPM (left) hallucinates significantly less than DDIM (right). (b) Towards the beginning of the reverse process, the trajectory selects a line segment to converge to. After that, the trajectory converges rapidly to the nearest line segment: either the true mode or the midpoint neighborhood. (c) Within the midpoint neighborhood, DDIM can get stuck and hallucinate, while DDPM can escape with added noise and avoid hallucination.

one estimates the score field $s_\theta(x_t)$ and discretizes Eq. (3) with a first-order exponential integrator (Lu et al., 2022) to obtain a *deterministic* sampling method for $p_{\text{data}}$ (Song et al., 2021a). In the main paper, we theoretically study the dynamics of Eq. (2) and Eq. (3) assuming access to the exact score function. We discuss this regularity condition in Sec. I. We also provide an empirical study under the inexact score function in the main paper experiments (Sec. 6) and theoretically characterize the inexact score function dynamics in Sec. C.

Finally, we consider a Gaussian mixture target distribution of $N$ components, $p_{\text{data}}(x) := \sum_{i=1}^{N} \pi_i \mathcal{N}(x; \mu^{(i)}, \sigma^2 I)$. Applying linearity of convolution yields that $p_t(x) = \sum_{i=1}^{N} \pi_i \mathcal{N}(x; \mu_t^{(i)}, \sigma_t^2 I)$, where $\mu_t^{(i)} := \sqrt{\bar{\alpha}} \mu^{(i)}$ and $\sigma_t^2 = \sigma^2 \bar{\alpha}_t + (1 - \bar{\alpha}_t)$. We then denote $\tilde{\sigma}_t^2 := \frac{\sigma_t^2}{\bar{\alpha}_t}$ as the effective variance; note that this starts large at the beginning of the reverse process, but becomes $\sigma^2$ at the end, i.e, the variance of an individual Gaussian mode. In the main paper, we consider the equal weights case where $\pi_i = \pi_j \ \forall i, j \in [N]$; we extend this to the unequal weights case in Sec. B of the appendix.

## 3. Related Work

**Diffusion Models**: Diffusion models generate samples by reversing a noise corruption process, with DDPM (Ho et al., 2020) and its deterministic fewer-step counterpart DDIM (Song et al., 2021a). Score-based generative modeling unifies DDPM-style reverse SDE and DDIM-style PF ODE (Song et al., 2021b); recent theory studies the gap between ODE- and SDE- sampling distributions (Deveney et al., 2025), convergence guarantees for DDIM vs. DDPM (Beyler & Bach, 2025), and statistical guarantees on the PF ODE (Chen et al., 2023; Cai & Li, 2025). Recent work also explores differences between the ODE and SDE in terms of performance (Cao et al., 2023) and proposes hy-

brid samplers (Xu et al., 2023). However, these works do not explicitly study hallucinations. Additionally, instead of studying how ODE/SDE trajectories evolve, they compare statistical distances (e.g., Wasserstein distance (Deveney et al., 2025)). These analyses do not provide insight into *where* and *when* hallucinations arise. To our knowledge, our work is the first to mathematically study the differences in hallucination mechanisms between DDIM and DDPM from the lens of the reverse ODE and SDE.

In our work, we also isolate key times in the reverse process trajectories for DDIM and DDPM. Several recent works isolate key times in the reverse trajectory as well: semantic features emerge in narrow time windows (Li & Chen, 2024) and, given the memorizing score function, there are distinct regimes of feature emergence and memorization (Biroli et al., 2024). Notably, these times and reverse process regimes are obtained under different regularity conditions and differ from ours. Bonnaire et al. (2025) study critical timesteps during diffusion *training* (e.g., the relevant steps of SGD), but this differs from our study of relevant steps in the reverse process.

Aside from quantifying key timesteps in the reverse process, $N$-mode Gaussian mixtures remain a central controlled testbed for understanding diffusion sampler behavior, with recent theory directly analyzing diffusion guidance (Wu et al., 2024), smoothness (Liang et al., 2025), convergence (Li et al., 2025), memorization (Buchanan et al., 2025), and generalization (Zhang et al., 2026) properties in $N$-Gaussian-mixture regimes. Our work adds to this thread of research, filling a gap in the mathematical understanding of hallucinations across different types of diffusion inference procedures.

**Hallucinations in Diffusion Models**: A growing line of work defines diffusion hallucinations as "mode interpolations", or sampled instances that are interpolations between

modes of the target distribution (Aithal et al., 2024). For example, in our Gaussian mixture setting, modes are the means of the Gaussian components and mode interpolations are instances which lie on the line segment joining any two modes. This is distinct from "mode collapse" (Zhang et al., 2018; Thanh-Tung & Tran, 2020), which describes a loss of diversity due to collapsing onto a subset of the true modes. In line with these recent works, we use mode interpolation as our working definition of hallucination, finding both empirically and theoretically that DDIM interpolates more often than DDPM. In this work, we assume a *well-separated Gaussian mixture* (Shah et al., 2023). This assumption is natural for any rigorous analysis of mode interpolation hallucinations; if regions of high probability mass around the modes overlap, then there are no mode interpolations since all samples are in these valid regions. Furthermore, while Aithal et al. (2024) attribute mode interpolation solely to the smoothing of the learned diffusion score field, we find that the mechanism underlying mode interpolation can arise even with the exact, non-smoothed score field.

Furthermore, many papers study detection and mitigation of hallucinations (Betti et al., 2026; Chandran.C et al., 2025; Triaridis et al., 2025; Somepalli et al., 2023); in contrast, we study how hallucinations arise specifically due to the choice of inference procedure. This emphasis is further supported by concurrent work showing higher counting hallucination rates for DDIM than DDPM (Fu et al., 2025), distributional differences in ODE- and SDE-based samplers (Deveney et al., 2025; Beyler & Bach, 2025), and empirical evidence that stochastic samplers outperform deterministic ones in scientific generation tasks (Shehata et al., 2025). Additionally, Triaridis et al. (2025) highlight the need to understand the difference in hallucination mechanisms between DDIM and DDPM for future mitigation methods, rather than treating mitigation purely as a post-hoc problem.

## 4. DDIM Hallucinations

In what follows, we aim to study how DDIM hallucinates. To do so, we first demonstrate that after the variance of $p_t$ becomes sufficiently small, DDIM in Eq. (3) converges to the $\varepsilon$-tube around the nearest $i, j$-mode segment, which is the line segment joining the two nearest modes $\boldsymbol{\mu}_t^{(i)}$ and $\boldsymbol{\mu}_t^{(j)}$ illustrated in Figure 2. Then, the rest of the DDIM dynamics remain within this tube, parallel to the nearest $i, j$-mode segment. Thus, if DDIM does not converge directly to one of the modes, then mode interpolation can happen on the $i, j$-mode segment. Next, we demonstrate that there exists a radius around the midpoint of the nearest $i, j$-mode segment such that DDIM cannot escape, thus hallucinating. In Sec. 5, we move to studying DDPM, demonstrating that the noise term in Eq. (2) leads to DDPM escaping the midpoint neighborhood, thus avoiding hallucination.

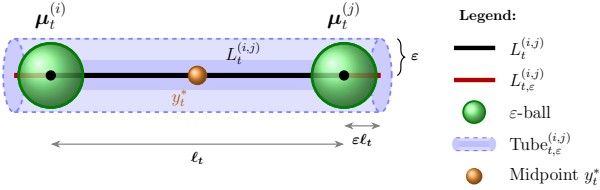

*Figure 2.* **(1)** In black, we have the line segment $L_t^{(i,j)}$ joining two modes. **(2)** Together with the red portion, this forms $L_{t,\varepsilon}^{(i,j)}$. **(3)** We then have the $\varepsilon$-ball surrounding modes $i$ and $j$. **(4)** Next, we have Tube$_{t,\varepsilon}^{(i,j)}$. **(5)** We also illustrate the midpoint of the line segment $\boldsymbol{y}_t^*$ (where $\boldsymbol{w}_t = 0$), discussed in Prop. 4.7. This provides a high-level description of key objects used throughout Sec. 4, and is not intended to describe the boundary between true and interpolated samples.

To begin, we prove that DDIM with Eq. (3) converges to the tube around the nearest $i, j$-mode segment after sufficient time. Firstly, we define an $\varepsilon$-extended $a, b$-mode segment for $a, b \in [N]$ as $L_{t,\varepsilon}^{(a,b)} := \{\boldsymbol{\mu}_t^{(b)} + v(\boldsymbol{\mu}_t^{(a)} - \boldsymbol{\mu}_t^{(b)}) : v \in [-\varepsilon, 1+\varepsilon]\}$. When $\varepsilon = 0$, this refers to just the line segment joining the two modes; we denote this by $L_t^{(a,b)}$. Next, we define the orthogonal distance of a trajectory to an $\varepsilon$-extended $a, b$-mode segment as $d_\perp(\boldsymbol{x}_t, L_{t,\varepsilon}^{(a,b)}) := \inf_{\boldsymbol{y} \in L_{t,\varepsilon}^{(a,b)}} ||\boldsymbol{y} - \boldsymbol{x}_t||_2$ and a $\zeta$-tube around $L_t^{(a,b)}$ as Tube$_{t,\zeta}^{(a,b)} := \{\boldsymbol{y} \in \mathbb{R}^\varpi : \exists \boldsymbol{k} \in L_t^{(a,b)}$ s.t. $||\boldsymbol{y} - \boldsymbol{k}||_2 \leq \zeta\}$. We similarly define the $\varepsilon$-ball around a mode. These key objects are illustrated in Figure 2. We then state our primary assumption.

**Assumption 4.1** (Two-Mode Dominance). *Consider a reverse diffusion process governed by Eq.* (3) *with Gaussian mixture target distribution $p_{data}$. Assume that there exists a time $\tau_1 \in (0, T)$ such that for $t \leq \tau_1$ there exists $i, j \in [N]$, $i \neq j$, where, for any $k \notin \{i, j\}$:*

$$||\boldsymbol{x}_t - \boldsymbol{\mu}_t^{(k)}||_2^2 \geq \min\left(||\boldsymbol{x}_t - \boldsymbol{\mu}_t^{(i)}||_2^2, ||\boldsymbol{x}_t - \boldsymbol{\mu}_t^{(j)}||_2^2\right) + \Delta_t,$$

*for some margin $\Delta_t \geq 2\sigma_t^2 \kappa$ where $\kappa > 1$.*

**Remark**: In the reverse process, the variance of $p_t$ begins large but decays to $\sigma^2$ by $t = 0$. Thus, we assume a critical time such that $\sigma_t$ is sufficiently small with respect to the distances from the modes. Note that in the case that the trajectory is very close to one particular mode, Asm. 4.1 can hold for several pairs of modes, for example modes $i, j$ and modes $i, k$. In this case, as we demonstrate later, the instance will simply converge to mode $i$, thus converging to the line segment between $i$ and any associated mode, including $i, j$. Otherwise, Asm. 4.1 says that, after sufficient time in the reverse process, the trajectory is nearer to a pair of modes than any other modes. Next, the bound on the margin is say-

ing that this distance is sufficiently large with respect to the variance of the mixture. This well-separation assumption is critical for any analysis of mode interpolations, ensuring that the rest $N-2$ modes do not have regions of high probability mass which overlap with modes $i$ and $j$. If this assumption does not hold, a mode interpolation between modes $i$ and $j$ may not be a hallucination and may instead be a high probability instance under $p_{\text{data}}$. Such well-separation conditions are common in diffusion analyses (Shah et al., 2023). We empirically verify this assumption in Figure E.6, finding that it holds for large $\kappa \gg 1$.

Given this assumption, we prove our first core result, demonstrating that DDIM converges fast to the $\varepsilon$-tube around the $i, j$-mode segment after $t < \tau_1$ in Asm. 4.1.

**Theorem 4.2** (Convergence to $\text{Tube}_{t,\varepsilon}^{(i,j)}$). *Suppose Asm. 4.1 holds. Then, there exists a time $\tau \leq \tau_1$ such that for all $t \in [0, \tau)$, we have exponential decay of the orthogonal distance to the $\varepsilon$-tube of the nearest $i, j$-mode segment, i.e.:*

$$d_\perp(\boldsymbol{x}_t, L_{t,\varepsilon}^{(i,j)}) \leq \mathcal{O}\left(d_\perp(\boldsymbol{x}_T, L_{T,\varepsilon}^{(i,j)}) \exp(-u(t)) + \varepsilon\right),$$

*where $\varepsilon \in \mathcal{O}(N \exp(-\kappa))$ and $u(t) := \int_t^T \frac{\beta_s}{2\sigma_s^2} ds$, which is monotonically increasing as $t \to 0$. This yields that, for any $\phi \in (0, 1)$, with probability at least $1 - 2\exp(-(\varpi)\phi^2/8)$, we have that:*

$$d_\perp(\boldsymbol{x}_t, L_{t,\varepsilon}^{(i,j)}) \leq \mathcal{O}\left(\sqrt{(\varpi)(1+\phi)} \exp(-u(t)) + \varepsilon\right).$$

*Proof Sketch*: Rescale Eq. (3) and then compute the dynamics around the nearest $i, j$-mode segment. Then, decouple the dynamics into parallel and orthogonal components. Finally, solve a first-order linear ODE for the orthogonal component and show that the parallel component remains restricted to only the $\varepsilon$-extended nearest $i, j$-mode segment. A full proof is provided in Sec. H.1.

Note that the rate of convergence to $\text{Tube}_{t,\varepsilon}^{(i,j)}$ is damped by the initial distance from the segment, which scales with $\sqrt{\varpi}$ with high probability. Next, note that after $\tau$, if the trajectory enters $\text{Tube}_{t,\varepsilon}^{(i,j)}$, the trajectory stays there.

**Corollary 4.3** (Trapping in $\text{Tube}_{t,\varepsilon}^{(i,j)}$). *For any $t \in [0, \tau]$, where $\tau$ is defined in Theorem 4.2, if $\boldsymbol{x}_t$ governed by the dynamics of Eq. (3) enters $\text{Tube}_{t,\varepsilon}^{(i,j)}$, where $\varepsilon \in \mathcal{O}(N \exp(-\kappa))$, it stays there.*

As such, once the trajectory is in $\text{Tube}_{t,\varepsilon}^{(i,j)}$, the only nontrivial component of the dynamics are the parallel dynamics to the nearest $i, j$-mode segment $L_{t,\varepsilon}^{(i,j)}$ within $\text{Tube}_{t,\varepsilon}^{(i,j)}$. We thus study these parallel dynamics; however, before doing so, we introduce a new assumption.

**Assumption 4.4** (Modes Sufficiently Far Apart). *Consider a reverse diffusion process governed by Eq. (3) with Gaussian mixture target distribution $p_{data}$. Then, consider time $\tau_2 \in (0, T)$, such that for $t \leq \tau_2$:*

$$\ell^2 := ||\boldsymbol{\mu}^{(i)} - \boldsymbol{\mu}^{(j)}||_2^2 \geq 4\kappa \tilde{\sigma}_t^2 \varpi,$$

*where $\kappa > 1$ from Asm. 4.1 and $\tilde{\sigma}_t^2 = \frac{\sigma_t^2}{\bar{\alpha}_t}$.*

**Remark**: Here, we assume that the effective variance is small enough such that the *distance between the two modes of interest*, $\boldsymbol{\mu}^{(i)}$ and $\boldsymbol{\mu}^{(j)}$, is larger. Furthermore, $\ell$ is *not* time-dependent. While $\kappa$ in Asm. 4.4 could be different, we take it to be the same as Asm. 4.1 for simplicity. Asm. 4.4 scales with dimension $\varpi$ in order to ensure regions of high probability mass do not overlap between modes $i$ and $j$ in high dimensions (Saremi & Hyvärinen, 2019), so that mode interpolation remains a valid definition of hallucination. We empirically verify this assumption in Figure E.6, finding that it holds for large $\kappa \gg 1$.

Next, we find that there are stable equilibria near the true modes:

**Proposition 4.5** (Mode Stability). *Suppose Asm. 4.1 and Asm. 4.4 hold. Consider $t \leq \hat{\tau}$ where $\hat{\tau} = \min\{\tau, \tau_2\}$. Then, with respect to the parallel dynamics to $L_{t,\varepsilon}^{(i,j)}$ while the trajectory is in $\text{Tube}_{t,\varepsilon}^{(i,j)}$, there exist instantaneous stable equilibria in a $\varepsilon$-ball around modes $\boldsymbol{\mu}_t^{(i)}$ and $\boldsymbol{\mu}_t^{(j)}$, where $\varepsilon \in \mathcal{O}(N \exp(-\kappa))$, for $\kappa \geq 2 \log N + \Theta(1)$.*

*Proof Sketch*: Apply the intermediate value theorem to show existence, then bound the drift derivative to check stability. See Sec. H.3 for a full proof.

**Remark**: Note that these parallel dynamics are the *exact* parallel dynamics in Eq. (G.50), incorporating an additional term due to the influence of the $N-2$ other modes, which is bounded by $\varepsilon$.

Furthermore, trajectories in a neighborhood of these instantaneous equilibria converge to a *stable trajectory* around the equilibria.

**Corollary 4.6** (Tracking Stability). *With the same conditions as Prop. 4.5, there exists a stable trajectory in a neighborhood around $\xi_t^{*,i}$ such that any two trajectories, in that same neighborhood, converge to the stable trajectory. This holds similarly for $\xi_t^{*,j}$.*

**Remark**: In particular, this means that trajectories which end up near $\xi_t^{*,i}$ stay near $\xi_t^{*,i}$, and same for $\xi_t^{*,j}$.

Next, we find that there is a neighborhood around the midpoint for which the parallel dynamics are *trapped*. Effectively, if an instance is trapped around the midpoint, this yields mode interpolation.

**Proposition 4.7** (Trapping in Midpoint Neighborhood).
*Suppose Asm. 4.1 and Asm. 4.4 hold. Let $\hat{\tau} = \min\{\tau, \tau_2\}$. Furthermore, suppose $\pi_i = \pi_j$. Then:*

1. *For $t \leq \hat{\tau}$, consider the approximate parallel dynamics in $\mathrm{Tube}_{t,\varepsilon}^{(i,j)}$, and denote these dynamics by $\xi_t$. Then, the point $\boldsymbol{y}_t^* = \frac{\boldsymbol{\mu}^{(i)} + \boldsymbol{\mu}^{(j)}}{2} + \boldsymbol{w}_t$, where $\boldsymbol{w}_t$ is the component of Eq. (3) orthogonal to $L_{t,\varepsilon}^{(i,j)}$, is a hyperbolic (instantaneously) unstable equilibrium with respect to these dynamics, with unstable eigenvalue $\lambda_t := \frac{\ell^2}{4\hat{\sigma}_t^2} - 1$. Note that $\boldsymbol{y}_t^*$ is exactly where $\xi_t = \frac{1}{2}$.*

2. *Suppose $\tau_3 \leq \hat{\tau}$. Consider $\xi(\cdot)$ as the approximate parallel dynamics in $\mathrm{Tube}_{t,\varepsilon}^{(i,j)}$. Choose a (nondimensionalized) radius $\vartheta \in (0, \frac{1}{2}]$ satisfying $0 < \vartheta < \frac{\underline{\lambda}}{2(1+\bar{\lambda})^2}$, where $\underline{\lambda} := \min_{\{t \in [0, \tau_3]\}} \lambda_t$ and $\bar{\lambda} := \max_{\{t \in [0, \tau_3]\}} \lambda_t$. Let $\Lambda_+ := \bar{\lambda} + 2(1 + \bar{\lambda})^2 \vartheta$. Then, if $0 < |\xi_{\tau_3} - \frac{1}{2}| < \vartheta \exp(-\Lambda_+ \tau_3)$, then:*

$$|\xi_t - \frac{1}{2}| < \vartheta \text{ for every } t \in [0, \tau_3].$$

*Equivalently, to possibly exit the midpoint neighborhood of radius $\vartheta$, a necessary condition is:*

$$\tau_3 \geq \frac{1}{\Lambda_+} \log(\frac{\vartheta}{|\xi_{\tau_3} - \frac{1}{2}|}).$$

*Proof Sketch*: Compute the root of the approximate parallel dynamics and check its derivative's positivity. Compute its unstable Jacobian eigenvalue $\lambda_t$. Bound the second derivative of the parallel dynamics and use this to bound the derivative of $|\xi_t - \frac{1}{2}|$; applying Grönwall's and leveraging our choice of $\vartheta$ yields the result. A full proof is provided in Sec. H.5.

**Discussion**: This proposition demonstrates that, for a sufficiently small time budget $\tau_3$, DDIM trajectories cannot leave a $\vartheta$-neighborhood of the midpoint. Even though the midpoint repels the instance along the line as an unstable equilibria, it does not do so sufficiently fast enough to reach the true modes. Additionally, note that if $\boldsymbol{w}_t$ is nonzero, this becomes a hyperbolic saddle: the negative eigenvalues represent attraction to $L_t^{(i,j)}$. This corresponds with the fact that the component of Eq. (3) orthogonal to $L_t^{(i,j)}$ vanishes in sufficient time per Theorem 4.2. We empirically demonstrate this in Figure E.12.

Note that we use the "approximate" parallel dynamics given in Eq. (H.234) to prove this result. The approximate dynamics ignore the $\varepsilon$ term induced by the $N - 2$ modes. In Prop. B.1, we prove that using the exact parallel dynamics in Eq. (G.50) only shifts the location of the point which

trajectories get stuck near. Empirically, as demonstrated in Figure E.12, we find that the location shift is negligible. We provide a comprehensive theoretical study of this in Sec. B. Additionally, the saddle still exists under score estimation error, as demonstrated in Sec. C.

In particular, this defines two regimes of convergence in the diffusion reverse process for the $N$ component Gaussian mixture, which we illustrate in Figure 1: (a) After $\tau$, $\boldsymbol{x}_t$ moves fast towards $\mathrm{Tube}_{t,\varepsilon}^{(i,j)}$ (b) If $\boldsymbol{x}_t$ converges outside a $\vartheta$-neighborhood of the midpoint, near one of the true modes $\boldsymbol{\mu}_t^{(i)}$ or $\boldsymbol{\mu}_t^{(j)}$, it converges fast to this true mode. Otherwise, if it converges near the midpoint $\boldsymbol{y}_t^*$, the trajectory gets "stuck" and requires large remaining time $\tau_3$ to leave.

## 5. DDPM Hallucinations

Recall that DDPM follows the reverse SDE Eq. (2), which shares the same drift as the PF ODE Eq. (3) (up to a constant factor) but includes an additional noise term. Consequently, DDPM exhibits the same two qualitative convergence stages as DDIM, described in Sec. 4, which we also verify empirically in Sec. 6. Here we focus on the final stage, when the trajectory is governed by the parallel dynamics within $\mathrm{Tube}_{t,\varepsilon}^{(i,j)}$. In this regime, we show that DDPM is substantially less likely than DDIM to terminate in the midpoint neighborhood responsible for mode interpolation hallucinations.

For modes $\boldsymbol{\mu}^{(i)}$ and $\boldsymbol{\mu}^{(j)}$, we define their bisector hyperplane as $H^{(i,j)} := \left\{ \boldsymbol{y} \in \mathbb{R}^\varpi : \|\boldsymbol{y} - \boldsymbol{\mu}^{(i)}\|_2^2 = \|\boldsymbol{y} - \boldsymbol{\mu}^{(j)}\|_2^2 \right\}$. Inside $\mathrm{Tube}_{t,\varepsilon}^{(i,j)}$, Theorem 4.2 implies that after $\tau$, the relevant motion reduces to a one-dimensional displacement from $H^{(i,j)}$ along its unit normal. In what follows, we make this reduction explicit and then bound the probability of ending near the midpoint.

Formally, let $\boldsymbol{u} := \frac{\boldsymbol{\mu}^{(j)} - \boldsymbol{\mu}^{(i)}}{\|\boldsymbol{\mu}^{(j)} - \boldsymbol{\mu}^{(i)}\|_2}$ and define the signed bisector coordinate $A_t := \left\langle \boldsymbol{x}_t - \frac{1}{2}(\boldsymbol{\mu}^{(i)} + \boldsymbol{\mu}^{(j)}), \boldsymbol{u} \right\rangle$ and note that $A_t = 0$ if and only if $\boldsymbol{x}_t \in H^{(i,j)}$, ranging from $-\frac{(\ell+\varepsilon)}{2}$ and $\frac{\ell+\varepsilon}{2}$ along the $\epsilon$-extended $i, j$-mode segment. Applying Itô's formula to $A_t$ and defining the one-dimensional Brownian motion $dB_t := \langle \boldsymbol{u}, d\bar{\boldsymbol{W}}_t \rangle$ yields:

$$dA_t = b(A_t, t)\, dt + \sqrt{2\eta(t)}\, dB_t, \qquad A_T = 0, \quad (4)$$

where $\eta(t) = \frac{1}{2}\beta_t$ and $b(A_t, t) = \left\langle -\frac{1}{2}\beta_t \boldsymbol{x}_t - \beta_t \nabla_{\boldsymbol{x}_t} \log p_t(\boldsymbol{x}_t), \boldsymbol{u} \right\rangle$. We then leverage this SDE to bound the probability of DDPM terminating in a $\vartheta$-neighborhood of $\boldsymbol{y}_t^*$.

**Proposition 5.1** (DDPM Terminal Midpoint Bound). *Let $A_t$ satisfy Eq. (4) on $t \in [0, \tau_3]$, with $\eta \in L^1([0, \tau_3])$ and $\eta(t) \leq \eta_{\max}$ for all $t \in [0, \tau_3]$. Fix $\vartheta > 0$. Assume there*

*exists a measurable function $K : [0, \tau_3] \to [0, \infty)$ such that for every $t \in [0, \tau_3]$ and every $a \in [-2\vartheta, 2\vartheta]$,*

$$|b(a, t)| \leq K(t) |a|. \qquad (5)$$

*Assume further there exists $\lambda_{\mathrm{rep}} > 0$ such that for all $t \in [0, \tau_3]$,*

$$\begin{aligned} b(a, t) &\leq -\lambda_{\mathrm{rep}}\, a & \forall\, a \geq \vartheta, \\ b(a, t) &\geq -\lambda_{\mathrm{rep}}\, a & \forall\, a \leq -\vartheta. \end{aligned} \qquad (6)$$

*Then the terminal midpoint event satisfies*

$$\begin{aligned} \mathbb{P}\big(|A_0| \leq \vartheta\big) \leq{} & \frac{4}{\pi} \exp\left( -\frac{\pi^2 \int_0^{\tau_3} \eta(s)\, ds}{16\left(1 + \int_0^{\tau_3} K(s)\, ds\right)^2 \vartheta^2} \right) \\ & + 2 \exp\left( -\frac{\lambda_{\mathrm{rep}}\, \vartheta^2}{2\eta_{\max}} \right). \end{aligned} \qquad (7)$$

*Proof Sketch*: We upper bound the probability of terminating in the midpoint neighborhood by splitting trajectories into two cases and combining their bounds: (i) those that remain within a $2\vartheta$-slab around the bisector for the entire reverse-time interval, and (ii) those that exit this slab. On the *confinement event* (i), Eq. (5) implies that staying trapped forces the stochastic martingale term to also remain trapped. Using Dubins & Schwarz (1965), we view this martingale as a time-changed Brownian motion with time horizon given by its quadratic variation $2\int_0^T \eta(s)\, ds$, and a standard Brownian confinement estimate yields the first exponential term. On the *exit event* (ii), to still terminate near the midpoint, the path must leave to distance $2\vartheta$ and later return to within $\vartheta$. The strong Markov property at the exit time reduces this to a return probability from $\pm 2\vartheta$. Under Eq. (6), an exponential supermartingale and optional stopping bound this return, giving the second exponential term. A full proof can be found in Sec. H.6.

**Discussion.** Prop. 5.1 is the DDPM analogue of the DDIM midpoint persistence result in Prop. 4.7. After trajectories enter $\mathrm{Tube}_{t,\varepsilon}^{(i,j)}$, DDIM's remaining failure mode is to terminate near the bisector, producing mode interpolation samples. This proposition shows that DDPM makes this terminal midpoint event exponentially unlikely, so trajectories in the stage where they are within $\mathrm{Tube}_{t,\varepsilon}^{(i,j)}$ overwhelmingly tend to end near one of the two true modes instead. We validate this empirically in Sec. 6. The assumptions of Prop. 5.1 are empirically verified in Sec. E.7.

## 6. Experiments

We provide experiments that justify our theoretical assumptions; demonstrate our theoretical results; and validate our subsequent interpretations. We primarily experiment with a Gaussian mixture consisting of 25 modes arranged on a

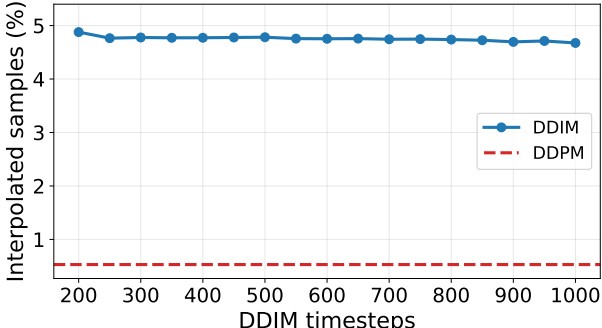

*Figure 3.* Hallucination rate for varying number of DDIM steps used in the reverse process. Notice that the number of DDIM interpolated samples is consistently larger than that of DDPM. Thus, this invalidates the idea that the gap between DDIM and DDPM hallucination rates arises due to skipping steps.

uniform grid in 2 dimensions; we study this dataset with higher dimensions in Sec. E.2. Unless otherwise stated, we use 1000 DDPM steps ($T = 1000$) and 50 DDIM sampling steps with quadratic skipping. Firstly, we show that mode interpolation is a primary source of hallucinations during sampling. We also demonstrate that the high hallucination rate gap between DDIM and DDPM is *not* due to DDIM skipping steps (i.e., PF ODE discretization) in the reverse process. Finally, we demonstrate that DDPM noise helps escape the $\vartheta$-neighborhood around the midpoint, thus avoiding hallucinations, as predicted by our theoretical results. Further experimental details are provided in Sec. F; we also provide additional experiments on images as well as ablations in Sec. E.

**DDIM Hallucinates More Than DDPM**: In Figure 1a, we find that DDIM hallucinates more than DDPM.

We classify a sample as a *mode interpolation* if it lies more than $5\sigma$ from every true mode but within $5\sigma$ of a line segment joining two modes. Note that we omit $< 0.01\%$ of instances which land outside $5\sigma$ of any line segment; these instances arise due to score error, and we do not focus on them for the purposes of this work. We expand on our choice of threshold and characterization of these samples in Sec. F. Additionally, we reproduce the recent results of Fu et al. (2025) in Sec. E.3, demonstrating that DDIM has more counting hallucinations than DDPM as well for an image dataset.

Crucially, in Figure 3, we demonstrate that the significantly higher hallucination rate of DDIM is *not due to skipping steps*. Specifically, from coarse ODE discretizations to fine ODE discretizations, the hallucination rate of DDIM remains larger than that of DDPM.

Finally, we observe that for an axis-aligned grid, hallucinations rarely happen on the diagonals joining two modes.

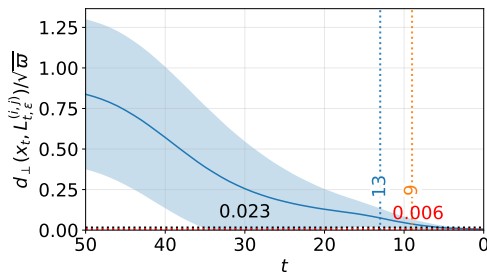 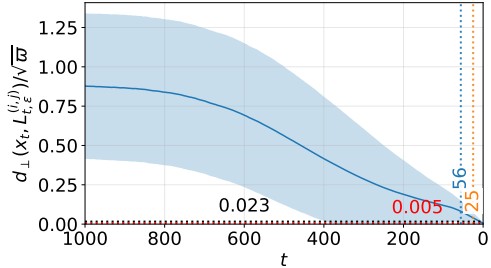 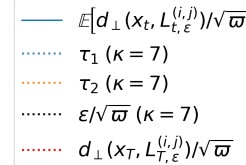

*(a)* DDIM convergence to the $i, j$-mode segment.  *(b)* DDPM convergence to the $i, j$-mode segment.

*Figure 4.* For both DDIM (Figure 4a) and DDPM (Figure 4b), we plot the convergence rate to the nearest $i, j$-mode segment across 100,000 trajectories, finding that convergence occurs after $\tau_1$ and thus validating Theorem 4.2. Note that $i, j$ change across time in these figures; however, as expected, after $\tau_1$ they become fixed. We plot $\varepsilon / \varpi$ as a dotted black line, finding that convergence to $\text{Tube}_{t,\varepsilon}^{(i,j)}$ is after $\tau_2$; thus, only the parallel dynamics become relevant at this stage. We also find that DDPM ends up closer to the nearest $i, j$-mode segment (dotted red line). Note that DDIM has 50 steps due to quadratic skipping. Uncertainty is quantified in the blue region as the $95\%$ confidence interval across trajectories.

This is also observed by Aithal et al. (2024). We provide theoretical justification for this in Sec. D.

**Convergence to Nearest $i, j$-Mode Segment**: In Figure 4, we validate Theorem 4.2 empirically, firstly finding that Assumptions 4.1 and 4.4 hold for large $\kappa = 7$. In Sec. E.1, we find that these assumptions hold for a range of other large $\kappa$ as well. We then find that DDIM and DDPM converge to the $\varepsilon$-extended nearest $i, j$-mode segment after $\tau_1$, where $\varepsilon = N \exp(-\kappa)$. Note that this line segment varies with time, and $i, j$ can change over the reverse process. Importantly, to verify Theorem 4.2, we compute the distance (and $\varepsilon$) rescaled by dividing $\sqrt{\varpi}$. We also find that the trajectories pass into the $\varepsilon$-tube after $\tau_2$, empirically verifying that the phenomena described in Prop. 4.7 are relevant as soon as the trajectory enters the $\varepsilon$-tube. These results are for $\varpi = 2$; results for higher dimensions are in Sec. E.2.

**DDPM Noise Helps Avoid Hallucinations**: In what follows, we aim to demonstrate that Prop. 4.7 and Prop. 5.1 hold for a nontrivial radius $\vartheta$. That is, for $\tau_3 = 9$ (since $\tau_2 = 9$ in Figure 4) remaining steps, DDIM trajectories get stuck within a $\vartheta$-neighborhood of the midpoint $y_t^*$ while DDPM trajectories escape. Specifically, we start DDIM and DDPM (with appropriately adjusted $\beta_t$) along $L_t^{(a,b)}$ at time $\tau_3$ and evaluate the hallucination rate after starting at this point. We do so for 100,000 trajectories across all $a, b \in [N]$. Critically, in Figure 5, we find that there exists a $\vartheta(\tau_3) > 0$ such that if a trajectory starts within $\vartheta(\tau_3)$ of the midpoint, it hallucinates. Next, the figure demonstrates that within this radius, DDPM has a significantly lower hallucination rate than DDIM. We thus conclude that the noise term in DDPM helps it escape the $\vartheta$-neighborhood around the midpoint, thus avoiding hallucinations. In Figure 5, we leverage this insight to evaluate the following DDIM-DDPM hybrid procedure: we add $z$ DDPM steps to DDIM after $\tau_3$, and find that this helps avoid hallucinations as well. In

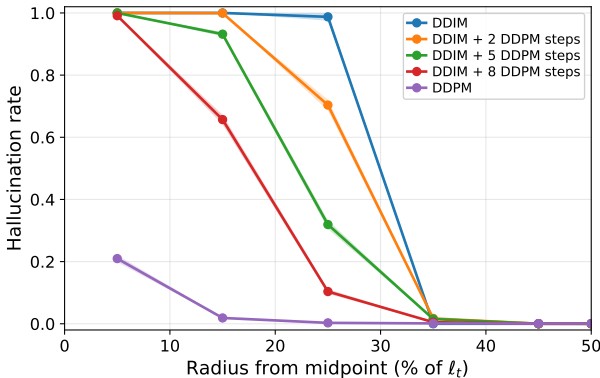

*Figure 5.* Starting DDIM at $\tau_3 = 9$, we find that for $\vartheta = 0.15\ell_t$, DDIM gets stuck before it can reach the true modes, i.e., it hallucinates, as predicted by Prop. 4.7. Furthermore, DDPM has a lower hallucination rate within this same $\vartheta$. Thus, we conclude that DDPM noise helps escape the $\vartheta$-neighborhood around the midpoint, as predicted by Prop. 5.1. Given this, we find that adding $z$ DDPM steps after starting DDIM at $\tau_3$ helps mitigate hallucination. Uncertainty is quantified by the standard error across mode pairs.

Sec. E.4, we demonstrate that our results hold across various choices of $\tau_3$. Note that $\tau_3$ is taken with respect to DDIM steps throughout our experiments.

## 7. Discussion

We present a geometric analysis of hallucinations in diffusion models through the lens of Gaussian mixtures. In particular, after sufficient time, diffusion trajectories undergo two key convergence phases: rapid collapse onto a nearby line segment connecting two modes, followed by dynamics along this line. Next, if a trajectory does not converge to one of the modes, it can get stuck near a point; in the case that Gaussian mixture weights are equal, this is at the

midpoint. This provides insight into the core mechanisms which differ between DDIM and DDPM: stochastic noise can help DDPM escape the neighborhood of the midpoint to one of the true modes and avoid hallucination.

Our results thus imply that the noise term is beneficial for reducing mode interpolation hallucinations. Empirically, we find this to be the case as well: adding only a few diffusion steps at the end of the reverse process significantly lowers the hallucination rate. We hope that these results highlight to the community the pitfalls of preferring DDIM over DDPM and inspire future work on designing flow-based generative models which avoid the midpoint neighborhood entirely, thus mitigating hallucinations.

**Limitations and Future Work**: Our primary analysis assumes a Gaussian mixture target distribution with well-separated modes. While we prove some results under an inexact score in Sec. C, a full characterization of what changes under score error is left to future work. Additionally, Aithal et al. (2024) highlight that mode interpolation can also happen in image latent space. Thus, a natural extension of our theoretical analysis to images is studying latent diffusion (Rombach et al., 2022) with Gaussian mixture latents. Furthermore, we believe that our work provides a first step towards designing adaptive samplers or models which detect when trajectories are stuck and inject DDPM noise accordingly, thus mitigating hallucinations while keeping the computational benefits of DDIM.

## Acknowledgments

We are truly thankful to the reviewers for their thoughtful and constructive feedback. We are also thankful to Zulip for supporting our online communication.

## Impact Statement

Hallucinations, generations that violate structural or semantic constraints of the training distribution, are a key obstacle in the adoption of state-of-the-art generative models, like diffusion, in safety critical contexts. Our work contributes to removing this obstacle by giving a mechanistic account of one category of hallucination: mode interpolation, where trajectories terminate between modes of the target distribution. We identify why DDIM, a commonly used diffusion inference procedure, has a higher hallucination rate than its stochastic counterpart, DDPM. Concretely, the noise in the DDPM reverse process can help it get unstuck from a region around the midpoint, where DDIM would otherwise hallucinate. Empirically, we highlight that even mild noise injection can be beneficial, guiding practitioners to safer generative modeling. Altogether, we believe our work has positive societal impact and might lead to a better understanding of the generalization of generative models as well

(Vedantam et al., 2018; Zhao et al., 2018; Georgopoulos et al., 2020; Deschenaux et al., 2024).

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

## Contents of the Appendix

# A. Full Notation

We let $x \in \mathbb{R}$ denote a scalar. Bolded lowercase letters $\boldsymbol{x} \in \mathbb{R}^{\varpi}$ denote vectors of dimension $\varpi$. Bolded upper case letters $\boldsymbol{X} \in \mathbb{R}^{\varpi \times \varpi}$ denote matrices with $\varpi$ rows and columns, unless dimensionalities are explicitly specified. $[N]$, for scalar $N \in \mathbb{R}$, denotes the integer subset $\{1, 2, ..., N\}$. We use $||\boldsymbol{x}||_2$ to denote the $\ell_2$ norm of a vector $\boldsymbol{x}$. We use $\langle \boldsymbol{x}_1, \boldsymbol{x}_2 \rangle$ to denote the inner product between vectors $\boldsymbol{x}_1, \boldsymbol{x}_2$, which is the dot product $\boldsymbol{x}^T \boldsymbol{x}$ unless explicitly specified otherwise. For a function $f$, we use $\nabla_{\boldsymbol{x}} f$ to denote its gradient if $f : \mathbb{R}^{\varpi} \to \mathbb{R}$ and its Jacobian if $f : \mathbb{R}^{\varpi} \to \mathbb{R}^o$. We denote by $x_t : [0, T] \to \mathbb{R}$ and $\boldsymbol{x}_t : [0, T] \to \mathbb{R}^{\varpi}$ time-dependent scalar-valued and vector-valued functions, respectively, from time scalar time $T$ to $0$ unless otherwise specified. We use $\dot{x}_t$ to denote the time derivative of $x_t$ and similarly use $\dot{\boldsymbol{x}}_t$ for $\boldsymbol{x}_t$; for all other derivatives, we use the notation $\frac{d}{dx} f(x)$. $\mathcal{N}(\boldsymbol{x}; \boldsymbol{\mu}, \sigma^2 \boldsymbol{I})$ is used to denote the probability density function of a $\varpi$-dimensional Gaussian centered at $\boldsymbol{\mu}$ with variance $\sigma^2 \boldsymbol{I}$, where $\boldsymbol{I}$ is the identity matrix. We write stochastic differential equations (SDE) as $d\boldsymbol{x}_t = b(t, \boldsymbol{x}_t) dt + \sigma(t, \boldsymbol{x}_t) d\boldsymbol{W}_t$ where $\boldsymbol{W}_t \in \mathbb{R}^{\varpi \times 1}$ is the standard Brownian motion, which is equivalent to the weak form $\boldsymbol{x}_t = \boldsymbol{x}_0 + \int_0^t b(s, \boldsymbol{x}_s) ds + \int_0^t \sigma(s, \boldsymbol{x}_s) d\boldsymbol{W}_t$. We say that, for $f, g : \mathbb{R}^{\varpi} \to \mathbb{R}^o$, $f \in \mathcal{O}(g)$ if $\exists c_1 > 0$ s.t. $f(\boldsymbol{x}) \le c_1 g(\boldsymbol{x})$ for $x$ sufficiently large; $f \in \Omega(g)$ if $\exists c_2 > 0$ s.t. $f(x) \ge c_2 g(x)$ for $x$ sufficiently large; and $f \in \Theta(g)$ if $f \in \mathcal{O}(g)$ and $f \in \Omega(g)$.

# B. Saddle Existence for Unequal Weights and Exact Parallel Dynamics

In Prop. 4.7, we demonstrated that, after sufficient time in the reverse process, the midpoint was an equilibrium with respect to the approximate parallel dynamics, under equal weights of the $i, j$-mode segment $\pi_i = \pi_j$. Here:

1. Under unequal weights, we demonstrate a sufficient condition for an equilibrium to exist.

2. Under the exact dynamics, we demonstrate that there exists an equilibrium, and that it does not differ greatly from the midpoint. This is also empirically justified by Figure E.12.

**Proposition B.1** (Equilibria Location under Unequal Weights and Exact Dynamics)**.** *Suppose Asm. 4.1 and Asm. 4.4 hold and fix $t \le \hat{\tau}$ (with $\hat{\tau}$ same as in Asm. 4.4). Consider the induced 1D parallel dynamics inside the tube.*

1. *(**Unequal weights**). Let $\ell := \|\boldsymbol{\mu}^{(i)} - \boldsymbol{\mu}^{(j)}\|_2$ . For the approximate parallel dynamics in $\mathrm{Tube}_{t,\varepsilon}^{(i,j)}$, there exists a point parallel to $L_{t,\varepsilon}^{(i,j)}$, which we denote by $\xi_{ij}^{\star}(t) \in (0, 1)$, which satisfies:*

$$\log\Big(\frac{\xi_{ij}^{\star}(t)}{1 - \xi_{ij}^{\star}(t)}\Big) = \log\Big(\frac{\pi_j}{\pi_i}\Big) + \frac{\ell^2}{\tilde{\sigma}_t^2}\Big(\xi_{ij}^{\star}(t) - \frac{1}{2}\Big). \tag{B.8}$$

*In particular, $\pi_i = \pi_j$ implies $\xi_{ij}^{\star}(t) = \frac{1}{2}$, and $\pi_j < \pi_i$ implies $\xi_{ij}^{\star}(t) > \frac{1}{2}$ (and vice versa).*

*Moreover,*

$$F_{ij,t}'\big(\xi_{ij}^{\star}(t)\big) = \frac{\ell^2}{\tilde{\sigma}_t^2} \xi_{ij}^{\star}(t)\big(1 - \xi_{ij}^{\star}(t)\big) - 1, \tag{B.9}$$

*so $\xi_{ij}^{\star}(t)$ is hyperbolic unstable whenever $F_{ij,t}'(\xi_{ij}^{\star}(t)) > 0$.*

2. *(**Exact Parallel Dynamics**). Under the exact parallel dynamics characterized in Eq. (G.50), between the two stable equilibria near modes $\boldsymbol{\mu}^{(i)}$ and $\boldsymbol{\mu}^{(j)}$ discussed in Prop. 4.5, there exists an equilibrium $\xi_N^{\star}(t)$ of the exact parallel dynamics. Furthermore, for $\kappa$ sufficiently large, assume there is an interval $I$ containing $\xi_{ij}^{\star}(t)$ such that, for some $m > 0$:*

$$F_{ij,t}'(\xi) \ge m > 0 \qquad \forall \xi \in I, \tag{B.10}$$

*where $F_{ij,t}$ is the drift of the approximate parallel dynamics. Then, $\xi_N^{\star}(t) \in I$ satisfies*

$$\big|\xi_N^{\star}(t) - \xi_{ij}^{\star}(t)\big| \le \varepsilon/m. \tag{B.11}$$

*where $\varepsilon \in \mathcal{O}(N \exp(-\kappa))$.*

*Proof sketch*: Item (1) follows by analyzing the approximate parallel dynamics for a single pair of components: the equilibrium condition yields the stated log-odds relation, and the location and slope follow from monotonicity and differentiation of the logistic responsibility. For item (2), under two-mode dominance the exact parallel dynamics are a uniformly small perturbation of the approximate dynamics. Existence of an interior unstable equilibrium then follows by continuity and the intermediate value theorem, and a mean-value argument yields the stated displacement bound. A full proof is given in Sec. H.7.

**Discussion**: The key contribution of this result is twofold. Firstly, we show that the exact parallel dynamics admit an interior equilibria rather than assuming its existence. Second, we show that this equilibrium is robust: its location does not deviate significantly under unequal mixture weights or when passing from the approximate to the exact dynamics. Item (1) characterizes how unequal weights bias the saddle along the $(i, j)$-line toward the lower-weight component. Item (2) shows that under two-mode dominance, the exact parallel dynamics differ from the approximate dynamics by only a small perturbation, so the saddle persists and remains close to the equal weight location. Finally, since an equilibrium exists between two stable (attracting) ones (which exist and are stable for the exact parallel dynamics per Prop. 4.5), and the exact parallel dynamics are one-dimensional, intuitively this is an equilibrium which repels, i.e. it is unstable. As such, it can also admit the confinement in a radius as discussed in Prop. 4.7.

## C. Results under Inexact Score

The preceding analysis leverages the closed-form score of Gaussian mixtures. In practice, the score $\nabla_{\boldsymbol{x}} \log p_t(\boldsymbol{x})$ is approximated by a neural network $s_\theta(\boldsymbol{x}, t)$. In this section, we characterize how score estimation error affects the mechanism established in Prop. 4.7.

**Definition C.1** (Score Error Field). *Define the score error field as*

$$\psi(\boldsymbol{x}, t) := s_\theta(\boldsymbol{x}, t) - \nabla_{\boldsymbol{x}} \log p_t(\boldsymbol{x}). \tag{C.12}$$

*The learned reverse dynamics become*

$$\dot{\boldsymbol{x}}_t = -\frac{1}{2}\beta_t \left( \boldsymbol{x}_t + \nabla_{\boldsymbol{x}_t} \log p_t(\boldsymbol{x}_t) + \psi(\boldsymbol{x}_t, t) \right). \tag{C.13}$$

**Assumption C.2** (Bounded Score Error). *There exists $\varrho : [0, T] \to \mathbb{R}_{\geq 0}$ such that $\|\psi(\boldsymbol{x}, t)\|_2 \leq \varrho(t)$ for all $\boldsymbol{x} \in \text{Tube}_{t,\varepsilon}^{(i,j)}$.*

**Assumption C.3** (Lipschitz Score Error). *There exists $L_\psi : [0, T] \to \mathbb{R}_{\geq 0}$ such that for all $\boldsymbol{x}, \boldsymbol{y} \in \text{Tube}_{t,\varepsilon}^{(i,j)}$:*

$$\|\psi(\boldsymbol{x}, t) - \psi(\boldsymbol{y}, t)\|_2 \leq L_\psi(t)\|\boldsymbol{x} - \boldsymbol{y}\|_2. \tag{C.14}$$

Under these assumptions, we establish that (1) the saddle location shifts under score error, and (2) the unstable eigenvalue can decrease, amplifying the slowdown effect.

**Proposition C.4** (Saddle Perturbation). *Suppose Asm. 4.1, Asm. 4.4, Asm. C.2, and Asm. C.3 hold. Let $\xi_{ij}^*(t)$ denote the unstable equilibrium of the approximate parallel dynamics, with unstable eigenvalue $\lambda_t > 0$.*

*Then, for $\bar{\varrho}(t)$ and $L_\psi$ sufficiently small, the perturbed parallel dynamics admit an equilibrium $\xi_\theta^*(t)$ satisfying*

$$|\xi_\theta^*(t) - \xi_{ij}^*(t)| \leq \frac{2\bar{\varrho}(t)}{\ell\lambda_t} + O(\bar{\varrho}(t)^2), \tag{C.15}$$

*where $\ell = \|\boldsymbol{\mu}^{(j)} - \boldsymbol{\mu}^{(i)}\|_2$ and $\bar{\varrho}(t) = \frac{\sigma_t^2}{\sqrt{\bar{\alpha}_t}}\varrho(t)$.*

*Proof Sketch:* Under score error $\psi$, the perturbed parallel drift becomes $\tilde{F}_{ij,t}(\xi) = F_{ij,t}(\xi) + e_t(\xi)$, where $|e_t(\xi)| \leq \bar{\varrho}(t)/\ell$ by Cauchy-Schwarz. Applying the implicit function theorem at the unperturbed equilibrium and Taylor expanding yields the displacement bound. A full proof is provided in Sec. H.8.

**Remark**: The displacement bound reveals a critical sensitivity: when $\lambda_t$ is small, even small score errors induce large saddle displacements. This explains why hallucination locations can be unpredictable in trained models.

**Proposition C.5** (Eigenvalue Perturbation). *Suppose Asm. 4.1, Asm. 4.4, Asm. C.2, and Asm. C.3 hold. The perturbed parallel dynamics have eigenvalue at $\xi_\theta^*(t)$:*

$$\lambda_\theta(t) = \lambda_t + \sigma_t^2 \boldsymbol{u}^\top \nabla_{\boldsymbol{x}} \psi(\sqrt{\bar{\alpha}_t}\boldsymbol{y}(\xi_\theta^*), t)\,\boldsymbol{u} + O(\bar{\varrho}(t)). \tag{C.16}$$

*In particular, if $\sigma_t^2 \boldsymbol{u}^\top \nabla_{\boldsymbol{x}} \psi(\sqrt{\bar{\alpha}_t}\boldsymbol{y}(\xi_\theta^*), t)\,\boldsymbol{u} < -\lambda_t - C\bar{\varrho}(t)$ for some constant $C > 0$, then $\lambda_\theta(t) < 0$, and the perturbed saddle becomes instantaneously stable.*

*Proof Sketch:* Differentiating the perturbed drift $\tilde{F}_{ij,t}(\xi) = F_{ij,t}(\xi) + e_t(\xi)$ and using $e_t'(\xi) = \sigma_t^2 \boldsymbol{u}^\top \nabla_{\boldsymbol{x}} \psi(\sqrt{\bar{\alpha}_t}\boldsymbol{y}(\xi_\theta^*), t)\,\boldsymbol{u}$ yields the eigenvalue perturbation. A full proof is provided in Sec. H.9.

**Remark**: Aithal et al. (2024) argue that hallucinations arise from score smoothing due to neural network approximation. Prop. C.5 is complementary: we show that the *directional* effect of score error determines whether slowdown is amplified or mitigated. Score error can either increase or decrease hallucination propensity depending on its directional structure.

We conclude by showing that DDPM's noise mechanism provides robustness against eigenvalue perturbation.

**Proposition C.6** (DDPM Robustness to Score Error). *Consider the perturbed DDPM dynamics*

$$dA_t = b_\theta(A_t, t)\,dt + \sqrt{2\eta(t)}\,dB_t, \tag{C.17}$$

*where $b_\theta(A_t, t) = b(A_t, t) + \tilde{e}_t(A_t)$ so that $|\tilde{e}_t(a)| \le r_A(t)$ where $r_A(t) := \beta_t \varrho(t)$.*

*Define $r_{A,\max} := \sup_{t \in [0,\tau_3]} r_A(t)$. Suppose the exact drift satisfies the escape condition in Prop. 5.1 with constant $\lambda_{\mathrm{rep}} > 0$. If*

$$r_{A,\max} < \lambda_{\mathrm{rep}}\vartheta, \tag{C.18}$$

*then the perturbed drift satisfies an escape condition with constant $\tilde{\lambda}_{\mathrm{rep}} := \lambda_{\mathrm{rep}} - \frac{r_{A,\max}}{\vartheta} > 0$, and the escape bound holds with $\lambda_{\mathrm{rep}}$ replaced by $\tilde{\lambda}_{\mathrm{rep}}$ when $b_\theta$ satisfies the assumption on $K$.*

*Proof:* A full proof is provided in Sec. H.10.

**Remark**: DDIM also has an escape mechanism via the unstable eigenvalue $\lambda_t > 0$. However, by Prop. C.5, score error can reduce $\lambda_t$ or make it negative, converting the saddle into a stable equilibrium. In contrast, DDPM's escape mechanism is additive (noise-driven) rather than relying solely on eigenvalue structure, providing robustness even when score error reduces $\lambda_t$.

# D. Interpolation on Mode Diagonals

In what follows, we provide a mathematical justification for why trajectories rarely mode interpolate on diagonals across grids:

**Proposition D.1** (Diagonal Avoidance on a Grid). *Suppose Asm. 4.1 holds with the same notation as above, with $\pi_i = \pi_j$, and assume the true modes lie on an axis-aligned rectangular grid. Then, if the $(i, j)$ in Asm. 4.1 corresponds to a diagonal pair of some grid cell, then for every $t \le \tau_1$:*

$$\min\{\tilde{\gamma}_{i,t}, \tilde{\gamma}_{j,t}\} \le \exp(-\kappa), \tag{D.19}$$

*where $\tilde{\gamma}_{i,t}, \tilde{\gamma}_{j,t}$ denotes the rescaled responsibilities of modes $i, j$ per Lemma G.2.*

*Proof Sketch.* We can reduce the problem to a single rectangular cell using the implications of Lemma G.3. Suppose, for contradiction, that the dominant pair $(i, j)$ corresponds to a diagonal of this cell. Then, using the geometric identity for a rectangle together with Asm. 4.1, we obtain an upper bound on $\rho$ of order $\mathcal{O}(e^{-\kappa})$. A full proof is given in Sec. H.11.

**Discussion**: This proposition is purely geometric, and does not rely on the parallel dynamics. The responsibility of a mode is its probability density proportional to that of all other modes; in particular, if the minimum of $\tilde{\gamma}_{i,t}, \tilde{\gamma}_{j,t}$ is exponentially small, then the trajectory must be very close to the other mode. In particular, by Cor. 4.6, the trajectory *must converge to the other mode*, since it is confined to $\mathrm{Tube}_{t,\varepsilon}^{(i,j)}$. For example, if mode $i$'s responsibility is small, then the trajectory must converge to mode $j$.

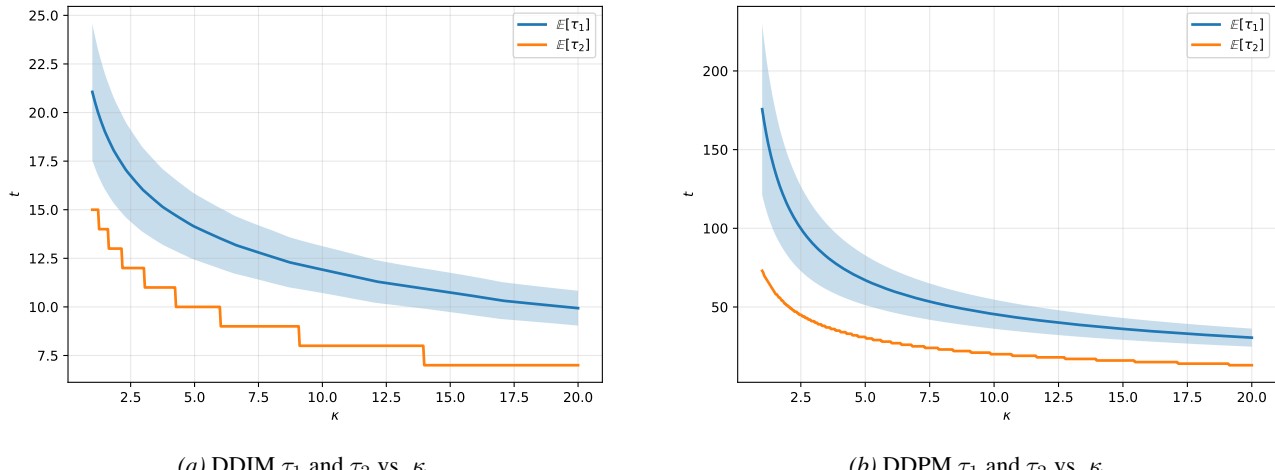

*(a)* DDIM $\tau_1$ and $\tau_2$ vs. $\kappa$.           *(b)* DDPM $\tau_1$ and $\tau_2$ vs. $\kappa$.

*Figure E.6.* For both DDIM (Figure E.6a) and DDPM (Figure E.6b), we plot how $\tau_1$ and $\tau_2$ change vs. $\kappa$ over 100,000 sampled trajectories, across 500 equidistant $\kappa$ increments ranging from 1 to 20. As expected, for large $\kappa$, Assumptions 4.1 and 4.4 hold; this yields that $\varepsilon$ is small as well. Furthermore, as expected, as $\kappa$ increases, both of these decrease. Note that $\tau_2$ is the same across all trajectories, since $\ell^2$ is the same across all mode $i, j$ pairs, so the standard deviation is 0. Furthermore, $\mathbb{E}[\tau_2]$ is much smoother for DDPM than DDIM, since DDPM has many more sampled timesteps to evaluate $\tau_2$ at; however, since $\mathbb{E}[\tau_1]$ average 100,000 different $\tau_1$, it is smooth across DDPM and DDIM.

# E. Additional Experiments

### E.1. Verifying Assumptions: Asm. 4.1 and Asm. 4.4

In Figure E.6, for the Gaussian dataset with $\varpi = 2$, we provide results demonstrating that for $\kappa >> 1$, $\tau_1$ and $\tau_2$ exist and are valid. In Sec. E.2, we demonstrate that this holds similarly for higher dimensions. Hence, Asm. 4.1 and Asm. 4.4 hold for $\kappa >> 1$ Note that, consistently, $\tau_2 < \tau_1$; in Figure 4, we show that $\tau_2$ is well before the time that the instance hits the $\varepsilon$-tube, as predicted by Theorem 4.2. Additionally, in $u$-time per Lemma G.2, Asm. 4.1 and Asm. 4.4 hold similarly.

### E.2. Higher Dimensional Gaussian Mixtures

Throughout, we use 25 modes and furthest point sampling, keeping $\ell$ roughly the same as in the $\varpi = 2$ case. Notably, however, some modes are slightly more or slightly less apart than $\ell$, hence adding uncertainty to $\tau_2$ across trajectories as well. In Figures E.7 and E.8, we validate Assumptions 4.1 and 4.4 for $\varpi = 2, 4, 8, 16, 32, 64$. We find that, as dimension increases, $\tau_1$ becomes larger (i.e., nearer to the beginning of the reverse process) and $\tau_2$ becomes smaller (i.e., nearer to the end of the reverse process) for both DDIM and DDPM, hence corresponding to different $\kappa$ (and thus different $\varepsilon$). Still, our two core assumptions still hold for large $\kappa$. In Figure E.9, we valuate Theorem 4.2 for both DDIM and DDPM across the same $\varpi$ increments. For both DDIM and DDPM, we find that exponential convergence still holds. We also find that $\varepsilon$ increases for DDIM as the dimension increases. Furthermore, $\varepsilon$ increases for DDPM as well, albeit at a much slower rate.

### E.3. DDIM Hallucinates More than DDPM For Images

In Figure E.10, we find that for a UNet denoiser pretrained on our 3 triangle dataset described in Sec. F, across various choices of training epochs, DDIM consistently hallucinates more than DDPM.

### E.4. Ablation on $\tau_3$

In Figure E.11, we find that as $\tau_3$ increases across samplers, hallucination rate drops, as expected per Prop. 4.7. However, note that a large $\tau_3$ is unrealistic; empirically, as demonstrated in Figure 4, $\tau_3$ is less than 15. Note that $\tau_3$ is written in DDIM steps systematically. In the case of DDPM, we omit the higher $\tau_3$ levels since this is very early in the reverse process in DDPM steps, resulting in divergence due to the DDPM noise term.

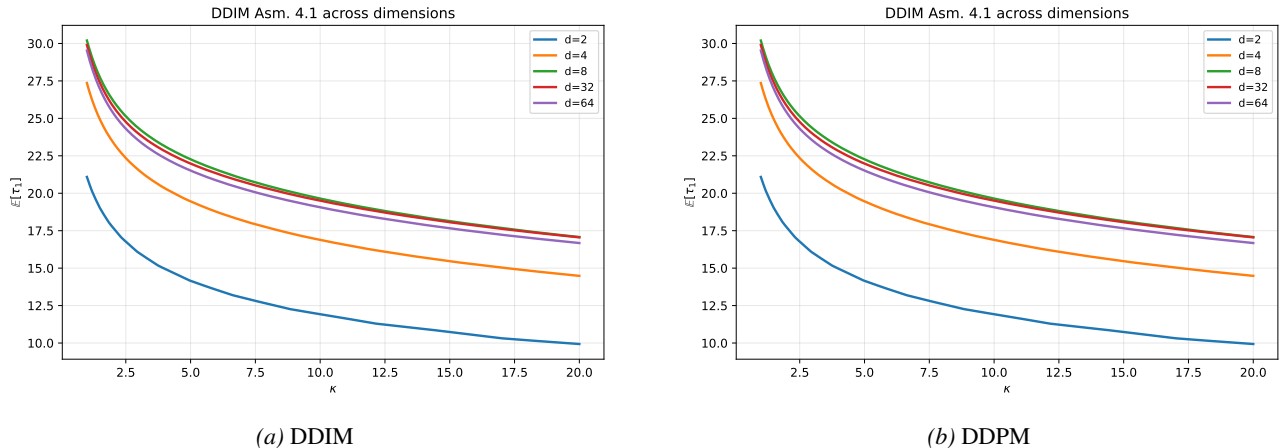

*(a)* DDIM          *(b)* DDPM

*Figure E.7.* $\tau_1$ vs. $\kappa$ for varying dimensions. We find that $\tau_1$ increases as dimension increases across both DDIM and DDPM.

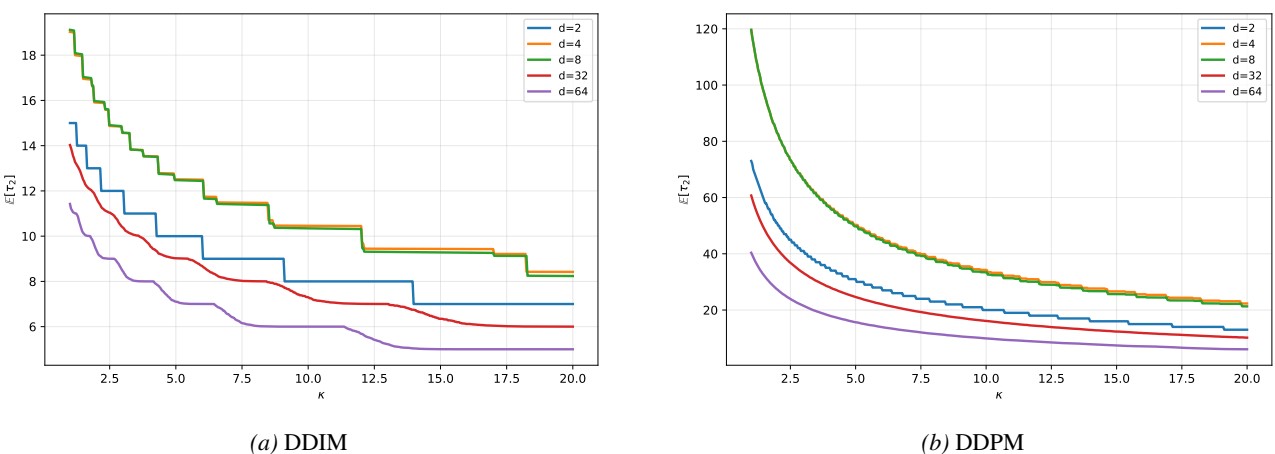

*(a)* DDIM          *(b)* DDPM

*Figure E.8.* $\tau_2$ vs. $\kappa$ for varying dimensions. We find that $\tau_2$ decreases as dimension increases across both DDIM and DDPM.

### E.5. Jacobian Eigenvalues

In Figure E.12, we show that there exists a positive ($\lambda_t$) and negative eigenvalue. These two eigenvalues are consistently positive (negative), demonstrating that the midpoint demonstrating that the midpoint is indeed a saddle. Thus, we can conclude that score error and studying the approximate parallel dynamics (rather than the exact parallel dynamics) only has a small effect on the position of the slowdown point.

### E.6. Varying Noise Level $\eta$ in DDIM

In Figure E.13, we provide a first characterization of how different noise levels $\eta$ (Song et al., 2021a) applied to the DDIM steps affect hallucination rate and escape from the midpoint region (contrasting to adding additional DDPM steps in Figure 5). In line with our results in noise being beneficial for reducing hallucinations, we find that the highest level of noise results in the lowest amount of mode interpolations. We also provide an analogue of Figure 5 for this setting, demonstrating that adding noise to DDIM helps escape the midpoint neighborhood, in line with our result in Prop. 5.1.

### E.7. Verifying Assumptions: Prop. 5.1

In Figure E.14, we verify the assumptions on $b(a, t)$ in Prop. 5.1 across 400k samples (10k per each line segment). We find that both assumptions hold, using the exact score presented in our Prop. 5.1, across all 40 line segments for a total of 400,000 samples. Details on how $b(a, t)$, $K(t)$, and $\lambda_{\rm rep}$ are computed are provided in Sec. F. Note that $\tau_3 = 9$ in DDIM steps corresponds to $\tau_3 = 21$ in DDPM steps.

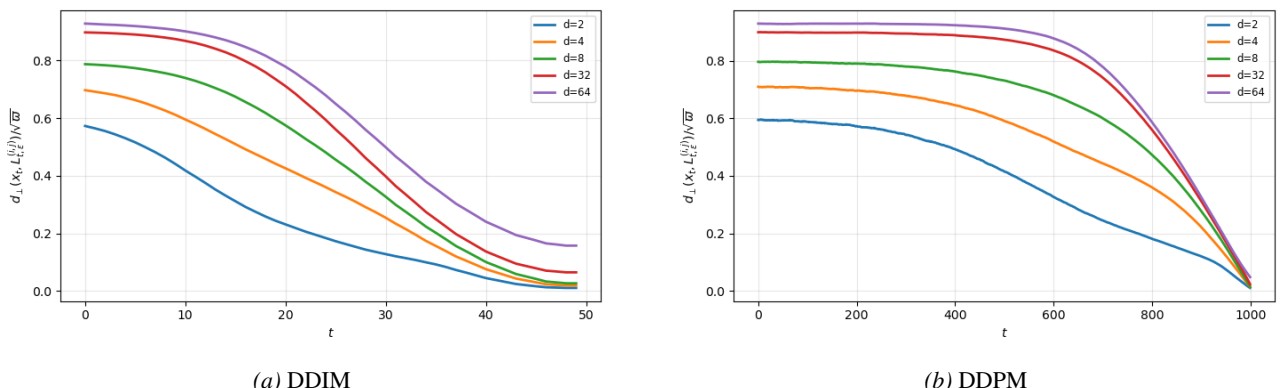

*(a)* DDIM                                                  *(b)* DDPM

*Figure E.9.* Orthogonal distance to $\varepsilon$-extended $i, j$-mode segment for DDIM. We find that, as dimension increases, $\varepsilon$ increases as well. Furthermore, the difference in final distances for DDIM and DDPM become more exacerbated as dimension increases.

### E.8. Midpoint Trapping under the Exact Score

In Figure E.15, we provide an analogue of Figure 5 in the exact score case to demonstrate that the midpoint trapping mechanism we extract theoretically in Prop. 4.7 and Prop. 5.1 holds, as expected, under the exact score. As discussed in Sec. I, this then becomes useful and predictive in the learned score case as demonstrated by Figure 5 and Tab. I.1.

### E.9. Hallucination Rate Ablations on $T$, $\sigma$, and $\ell$

In Figure E.16, we demonstrate that the gap in mode interpolation hallucination rate persists across various choices of $T$, $\sigma$, and $\ell$. In general, hallucination rates remain roughly stable. As $T$ increases, the gap increases: while discretization error does not explain the gap between DDIM and DDPM hallucinations, DDIM becomes a prohibitively coarse discretization of the PF ODE and thus has a higher hallucination rate. As $\sigma$ increases, the gap decreases. This is because regions of high probability mass begin to overlap, so feasibly classifying samples as hallucinations becomes rare: this is precisely why Assumptions 4.1 and 4.4 are required, since they ensure well-separated modes where is a useful definition of hallucinations. For similar reasons, as $\ell$ decreases, the gap decreases.

## F. Experimental Details

### F.1. Gaussian Mixture

We train our experiments 25-mode gaussian with varying dimensions. Our setup contains 25 modes with an inter-mode separation of 2 with each mode having a standard deviation $\sigma$ of 0.02. Lengths and standard deviations are then normalized by $2\sqrt{\varpi}$ during sampling. We sample $100,000$ samples uniformly across all modes and train a 3-layer MLP with a hidden size of 64 using a batch size of $10,000$ for $10,000$ epochs. We train with Adam (Kingma & Ba, 2015) with a learning rate of $1e{-}4$ which decays linearly to $1e{-}5$. For higher dimensions, modes are sampled from a lattice of $5^{\varphi}$. In higher dimensions ($\varphi > 2$) sampling of the 25 modes is done using farthest point sampling. This is done on set of candidates given by $\min(M, 5^{\varphi})$, with a value of $M = 500,000$ as the maximum number of candidate points. Additionally, in higher dimensions, we use a stronger training pipeline of Fourier features, embedding the log signal-to-noise ratio, and varying depth to ensure approximation of the score.

During sampling, we choose $100,000$ sampled freshly from a standard normal distribution $\mathcal{N}(0, 1)$ and run inference on those samples in the reverse process. Furthermore, in $2D$, we eliminate invalid samples by removing those trajectories for which the final distance to the line joining two modes is greater than $5\sigma$. These are those trajectories for which the outcome is neither a mode interpolation nor a convergence to one of the true modes. In the 2D Gaussian, this amounts to $< 0.01\%$ of the total number of samples used. Mode interpolations in $2D$ are thus samples which are within $5\sigma$ of a nearby line but further than $5\sigma$ from any mode. A lower threshold would result in false true samples with probability $\approx 1$ across 100,000 samples. For higher dimensional Gaussian datasets ($> 10$), to handle concentration of measure, we use a threshold of $\sigma(\sqrt{\varpi} + 4)$ instead. Additionally, for the analysis of exponential convergence, we find that this threshold is hard to tune, since initial distances scale with $\sqrt{\varpi}$; thus, any instance which is initially further than $\sqrt{\varpi}$ at the beginning of the reverse

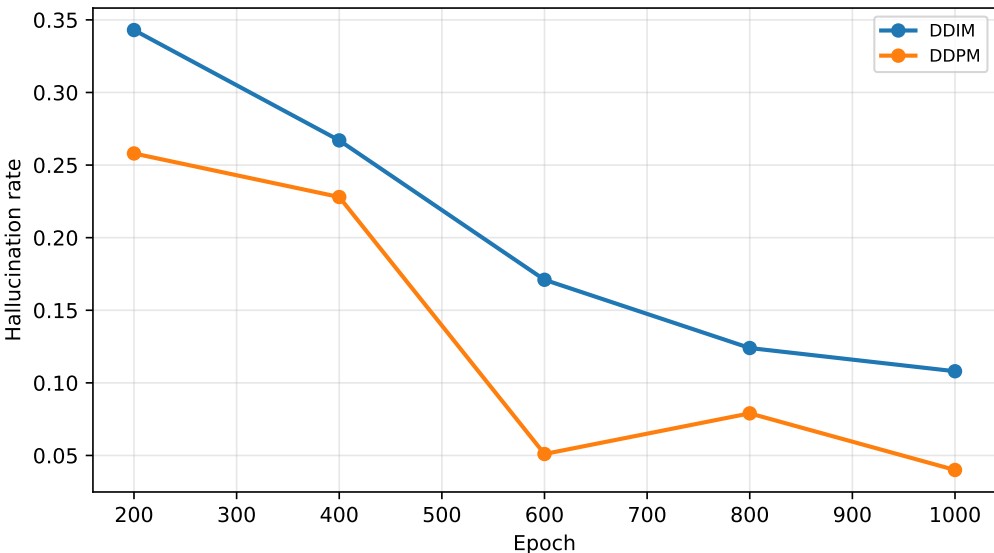

*Figure E.10.* We compare the image hallucination rate of DDIM and DDPM for 1000 epochs of training using a UNet architecture on the 3 triangle dataset. The evaluation is conducted on 1,000 samples uniformly sampled from $\mathcal{N}(0, 1)$ Note that the hallucination rate of DDPM is consistently lower than DDIM in this case.

process is removed. This also amounts to a negligible amount of samples.

### F.2. Counting Triangles

We create a dataset of 10,000 images of size $64 \times 64$ with 3 triangles placed randomly in the image. The locations are chosen such that the images are non-overlapping. The goal is to faithfully generate images that maintain the number of triangles in the generated image with no constraint on the position of the triangles in the images generated. Example images are provided in Figure F.17.

### F.3. Computing Prop. 5.1 Objects

Note that we can write $\boldsymbol{x}_t = \boldsymbol{m} + A_t \boldsymbol{u} + \boldsymbol{z}_t$ where $\langle \boldsymbol{z}_t, \boldsymbol{u} \rangle = 0$ and $||\boldsymbol{z}_t||_2 \leq \varepsilon$, since the trajectory is restricted to the $\varepsilon$-tube. This thus allows us to compute $b(A_t, t)$ with respect to $A_t$. Note that these require $b(a, t)$ to be negative (positive) for $a \geq \vartheta$ ($a \leq -\vartheta$), since $b(a, t)$ is moving in reverse time in the statement of Prop. 5.1. Next, we check the two assumptions of Prop. 5.1 across all line segments and across 50 intervals of $a$ on each side of the line segment until the endpoints of $a$. We also sample $\boldsymbol{z}_t$ uniformly in $[-\varepsilon, \varepsilon]$ at each evaluation of $a$. For the assumption with respect to $K(t)$, the endpoints are $[-2\vartheta, 2\vartheta]$. For the assumption with respect to $\lambda_{\text{rep}}$, from the proof of Prop. 5.1, note that the assumption with $K(t)$ bounds the probability that the trajectory stays in a $2\vartheta$-neighborhood and the $\lambda_{\text{rep}}$ assumption bounds the probability that the trajectory returns after exit. Thus, in Sec. E.7, we verify the assumption with $\lambda_{\text{rep}}$ for $|a| \in [\vartheta, a^*]$, where $a^*$ is when return after exit happens $0\%$ of the time and DDPM trajectories interpolate with rate $< 0.001$ with $\tau_3$ steps remaining. $[\vartheta, a^*]$ is thus the region where return after exit becomes infeasible, which is precisely where the assumption on $\lambda_{\text{rep}}$ must hold. In Sec. E.7, $a^* = 0.280$ where $\vartheta = 0.15$ (nondimensionalized).

### F.4. Additional Figures

In Figure F.18, we illustrate the reduction to the DDPM dynamics along the bisector normal which is integral for our theory in Sec. 5.

## G. Helpful Lemmas

**Lemma G.1.** *Given the diffusion marginals as defined in Eq.* (2), *we can rewrite the score as:*

$$\nabla_{\boldsymbol{x}_t} \log p_t(\boldsymbol{x}_t) = -\frac{1}{\sigma_t^2} \left( \boldsymbol{x}_t - \sum_{k=1}^{N} \gamma_k(\boldsymbol{x}_t) \boldsymbol{\mu}_t^{(k)} \right) \tag{G.20}$$

*for the responsibilities* $\gamma_k(\boldsymbol{x}_t) = \frac{\pi_k \mathcal{N}(\boldsymbol{x}_t; \boldsymbol{\mu}_t^{(k)}, \sigma_t^2 \boldsymbol{I})}{\sum_{k=1}^{N} \pi_k \mathcal{N}(\boldsymbol{x}_t; \boldsymbol{\mu}_t^{(k)}, \sigma_t^2 \boldsymbol{I})}$, *where* $\sigma_t^2 = \sigma^2 \bar{\alpha}_t + (1 - \bar{\alpha}_t)$.

*Proof.* Firstly, we compute the gradient of the diffusion marginal as:

$$\nabla_{\boldsymbol{x}_t} p_t(\boldsymbol{x}_t) = \sum_{k=1}^{N} \pi_k \nabla_{\boldsymbol{x}_t} \mathcal{N}(\boldsymbol{x}_t; \boldsymbol{\mu}_t^{(k)}, \sigma_t^2 \boldsymbol{I}) \tag{G.21}$$

$$= \sum_{k=1}^{N} \pi_k \nabla_{\boldsymbol{x}_t} \mathcal{N}(\boldsymbol{x}_t; \boldsymbol{\mu}_t^{(k)}, \sigma_t^2 \boldsymbol{I}) \nabla_{\boldsymbol{x}_t} \left( -\frac{||\boldsymbol{x}_t - \boldsymbol{\mu}_t^{(k)}||_2^2}{2\sigma_t^2} \right) \tag{G.22}$$

$$= \sum_{k=1}^{N} \pi_k \nabla_{\boldsymbol{x}_t} \mathcal{N}(\boldsymbol{x}_t; \boldsymbol{\mu}_t^{(k)}, \sigma_t^2 \boldsymbol{I}) \left( -\frac{\boldsymbol{x}_t - \boldsymbol{\mu}_t^{(k)}}{\sigma_t^2} \right) \tag{G.23}$$

Then, incorporating Eq. (G.23) into the definition of the score yields:

$$\nabla_{\boldsymbol{x}_t} \log p_t(\boldsymbol{x}_t) = \frac{\nabla_{\boldsymbol{x}_t} p_t(\boldsymbol{x}_t)}{p_t(\boldsymbol{x}_t)} \tag{G.24}$$

$$= -\frac{1}{\sigma_t^2} \sum_{k=1}^{N} \frac{\pi_k \mathcal{N}(\boldsymbol{x}_t; \boldsymbol{\mu}_t^{(k)}, \sigma_t^2 \boldsymbol{I})}{p_t(\boldsymbol{x}_t)} (\boldsymbol{x}_t - \boldsymbol{\mu}_t^{(k)}) \tag{G.25}$$

$$= -\frac{1}{\sigma_t^2} \sum_{k=1}^{N} \gamma_k(\boldsymbol{x}_t)(\boldsymbol{x}_t - \boldsymbol{\mu}_t^{(k)}), \text{ definition of } \gamma_k \tag{G.26}$$

$$= -\frac{1}{\sigma_t^2} \left( \sum_{k=1}^{N} \gamma_k(\boldsymbol{x}_t)(\boldsymbol{x}_t) - \sum_{k=1}^{N} \gamma_k(\boldsymbol{x}_t) \boldsymbol{\mu}_t^{(k)} \right) \tag{G.27}$$

$$= -\frac{1}{\sigma_t^2} \left( \boldsymbol{x}_t - \sum_{k=1}^{N} \gamma_k(\boldsymbol{x}_t) \boldsymbol{\mu}_t^{(k)} \right), \text{ since } \sum_{k=1}^{N} \gamma_k(\boldsymbol{x}_t) = 1 \tag{G.28}$$

as desired. $\qquad\square$

**Lemma G.2.** *Let* $\boldsymbol{y}_t = \frac{\boldsymbol{x}_t}{\sqrt{\bar{\alpha}_t}}$, *where* $\boldsymbol{x}_t$ *is governed by Eq.* (3), *and let* $\hat{\boldsymbol{\mu}}(\boldsymbol{y}_t) = \sum_{i=1}^{N} \tilde{\gamma}_i(\boldsymbol{y}_t) \boldsymbol{\mu}^{(i)}$ *with new responsibilities:*

$$\tilde{\gamma}_k(\boldsymbol{y}_t) = \frac{\pi_k \mathcal{N}(\boldsymbol{y}_t; \boldsymbol{\mu}^{(k)}, \tilde{\sigma}_t^2 \boldsymbol{I})}{\sum_{m=1}^{N} \pi_m \mathcal{N}(\boldsymbol{y}_t; \boldsymbol{\mu}^{(m)}, \tilde{\sigma}_t^2 \boldsymbol{I})} \tag{G.29}$$

*and* $\tilde{\sigma}_t^2 = \frac{\sigma_t^2}{\bar{\alpha}_t}$.

*Next, define the time-rescaling* $u$ *as:*

$$u(t) = \int_t^T \frac{\beta_s}{2\sigma_s^2} ds \tag{G.30}$$

*Then, letting* $t = u$ *WLOG, we have the rescaled dynamics:*

$$\dot{\boldsymbol{y}}_t = \hat{\boldsymbol{\mu}}(\boldsymbol{y}_t) - \boldsymbol{y}_t \tag{G.31}$$

*Proof.* Firstly, let $I(t) = -\int_0^t \beta_s ds$, i.e. $\bar{\alpha}_t = e^{I(t)}$. This yields that:

$$\dot{\bar{\alpha}}_t = (e^{\dot{I}(t)}) \tag{G.32}$$

$$= e^{I(t)} \dot{I}(t) \tag{G.33}$$

$$= e^{I(t)}(-\beta_t) \tag{G.34}$$

$$= -\beta_t \bar{\alpha}_t \tag{G.35}$$

Next, by definition of $\boldsymbol{y}_t$, we have that:

$$\dot{\boldsymbol{y}}_t = \dot{\boldsymbol{x}}_t \bar{\alpha}_t^{-1/2} + \boldsymbol{x}_t \frac{d}{dt} \bar{\alpha}_t^{-1/2} \tag{G.36}$$

$$= \dot{\boldsymbol{x}}_t \bar{\alpha}_t^{-1/2} + \boldsymbol{x}_t \left( \frac{\beta_t}{2\sqrt{\bar{\alpha}_t}} \right) \tag{G.37}$$

$$= \frac{1}{\sqrt{\bar{\alpha}_t}} \left( -\frac{\beta_t}{2} (\boldsymbol{x}_t + \nabla_{\boldsymbol{x}_t} \log p_t(\boldsymbol{x}_t)) \right) + \frac{\beta_t}{2\sqrt{\bar{\alpha}_t}} \boldsymbol{x}_t \tag{G.38}$$

$$= \frac{-\beta_t}{2\sqrt{\bar{\alpha}_t}} \boldsymbol{x}_t - \frac{\beta_t}{2\sqrt{\bar{\alpha}_t}} \nabla_{\boldsymbol{x}_t} \log p_t(\boldsymbol{x}_t) + \frac{\beta_t}{2\sqrt{\bar{\alpha}_t}} \boldsymbol{x}_t \tag{G.39}$$

$$= -\frac{\beta_t}{2\sqrt{\bar{\alpha}_t}} \nabla_{\boldsymbol{x}_t} \log p_t(\boldsymbol{x}_t) \tag{G.40}$$

$$= -\frac{\beta_t}{2\sqrt{\bar{\alpha}_t}} \left( -\frac{1}{\sigma_t^2} (\boldsymbol{x}_t - \sum_{k=1}^{N} \gamma_k(\boldsymbol{x}_t) \boldsymbol{\mu}_t^{(k)}) \right), \text{ Lemma G.1} \tag{G.41}$$

$$= -\frac{\beta_t}{2\sigma_t^2 \sqrt{\bar{\alpha}_t}} \left( \sum_{k=1}^{N} \gamma_k(\boldsymbol{x}_t) \boldsymbol{\mu}_t^{(k)} - \boldsymbol{x}_t \right) \tag{G.42}$$

$$= -\frac{\beta_t}{2\sigma_t^2 \sqrt{\bar{\alpha}_t}} \left( \sqrt{\bar{\alpha}_t} \sum_{k=1}^{N} \gamma_k(\boldsymbol{x}_t) \boldsymbol{\mu}^{(k)} - \sqrt{\bar{\alpha}_t} \boldsymbol{y}_t \right) \tag{G.43}$$

$$= -\frac{\beta_t}{2\sigma_t^2} (\hat{\boldsymbol{\mu}}(\boldsymbol{y}_t) - \boldsymbol{y}_t) \tag{G.44}$$

The chain rule yields the time rescaling:

$$\frac{d\boldsymbol{y}_t}{du} = \frac{d\boldsymbol{y}_t}{dt} \frac{dt}{du} \tag{G.45}$$

$$= \left( -\frac{\beta_t}{2\sigma_t^2} (\hat{\boldsymbol{\mu}}(\boldsymbol{y}_t) - \boldsymbol{y}_t) \right) \left( -\frac{2\sigma_t^2}{\beta_t} \right) \tag{G.46}$$

$$= \hat{\boldsymbol{\mu}}(\boldsymbol{y}_t) - \boldsymbol{y}_t \tag{G.47}$$

as desired. □

**Remark**. This lemma transforms the nonautonomous dynamics in Eq. (3), where the means evolve with the original diffusion time, into dynamics with respect to the *static means*. Here $t \in [0, T]$ denotes the original diffusion time, and the reverse trajectory is traversed from $t = T$ to $t = 0$. The variable $u$ defined above parameterizes this same reverse trajectory in the forward direction: $u$ increases as the original diffusion time $t$ decreases. Thus, stability statements in $u$-time are stability statements along the reverse trajectory, matching the direction in which the sampler evolves. For notational simplicity, after this change of variables we often relabel $u$ as $t$. Note that Asm. 4.1 and Asm. 4.4 hold in $u$ time as well, as discussed in Sec. E.1.

**Lemma G.3.** *Assume Asm. 4.1 holds. Consider $\tau_1$ from Asm. 4.1.. We decompose $\boldsymbol{y}_t$ into parallel and orthogonal components i.e.:*

$$\boldsymbol{y}_t = \boldsymbol{\mu}^{(i)} + \xi_t \ell \boldsymbol{u} + \boldsymbol{w}_t, \tag{G.48}$$

where $\ell = ||\boldsymbol{\mu}^{(j)} - \boldsymbol{\mu}^{(i)}||_2$, $\boldsymbol{u} = \frac{\boldsymbol{\mu}^{(j)} - \boldsymbol{\mu}^{(i)}}{\ell}$, and $\boldsymbol{w}_t$ satisfies $\langle \boldsymbol{w}_t, \boldsymbol{u} \rangle = 0$.

*Then, for $t \leq \tau_1$, we have that:*

$$\dot{\boldsymbol{y}} = \hat{\boldsymbol{\mu}}(\boldsymbol{y}_t) - \boldsymbol{y}_t \tag{G.49}$$

*is equivalent to the dynamical system:*

$$\dot{\xi}_t = \tilde{\gamma}_j(\boldsymbol{y}_t) - \xi_t + \delta_{\xi_t} \tag{G.50}$$
$$\dot{\boldsymbol{w}}_t = -\boldsymbol{w}_t + \boldsymbol{\delta}_{\boldsymbol{w}_t} \tag{G.51}$$

*where:*

$$|\delta_\xi|, ||\boldsymbol{\delta}_{\boldsymbol{w}}||_2 \leq \mathcal{O}\left(N \exp(-\kappa)\right) \tag{G.52}$$

*Proof.* Firstly, we have that:

$$\dot{\boldsymbol{y}}_t = \sum_{k=1}^{N} \tilde{\gamma}_k(\boldsymbol{y}_t)\boldsymbol{\mu}^{(k)} - \boldsymbol{y}_t \tag{G.53}$$

if and only if, expanding with Eq. (G.48):

$$\ell\dot{\xi}_t\boldsymbol{u} + \dot{\boldsymbol{w}}_t = \sum_{k=1}^{N} \tilde{\gamma}_k(\boldsymbol{y}_t)\boldsymbol{\mu}^{(k)} - (\boldsymbol{\mu}^{(i)} + \xi_t\ell\boldsymbol{u} + \boldsymbol{w}_t). \tag{G.54}$$

Next, we have that $\tilde{\gamma}_i(\boldsymbol{y}_t)(\boldsymbol{\mu}^{(i)} - \boldsymbol{\mu}^{(i)}) = 0$; $\tilde{\gamma}_j(\boldsymbol{y}_t)(\boldsymbol{\mu}^{(j)} - \boldsymbol{\mu}^{(i)}) = \tilde{\gamma}_j(\boldsymbol{y}_t)\ell\boldsymbol{u}$; and $\boldsymbol{E}_t := \sum_{k \notin \{i,j\}} \tilde{\gamma}_k(\boldsymbol{y}_t)(\boldsymbol{\mu}^{(k)} - \boldsymbol{\mu}^{(i)})$, which when incorporated into Eq. (G.54) yields:

$$\ell\dot{\xi}_t\boldsymbol{u} + \dot{\boldsymbol{w}}_t = \tilde{\gamma}_j(\boldsymbol{y}_t)\ell\boldsymbol{u} + \boldsymbol{E}_t - \xi_t\ell\boldsymbol{u} - \boldsymbol{w}_t \tag{G.55}$$
$$= (\tilde{\gamma}_j(\boldsymbol{y}_t) - \xi_t)\ell\boldsymbol{u} - \boldsymbol{w}_t + \boldsymbol{E}_t. \tag{G.56}$$

Next, noting that $\langle \boldsymbol{u}, \boldsymbol{u} \rangle = 1$, it follows from Eq. (G.56) that:

$$\langle \ell\dot{\xi}_t\boldsymbol{u}, \boldsymbol{u} \rangle = \langle (\tilde{\gamma}_j(\boldsymbol{y}_t) - \xi_t)\ell\boldsymbol{u} - \boldsymbol{w}_t + \boldsymbol{E}_t, \boldsymbol{u} \rangle \iff \tag{G.57}$$
$$\ell\dot{\xi}_t = (\tilde{\gamma}_j(\boldsymbol{y}_t) - \xi_t)\ell + \langle \boldsymbol{E}_t, \boldsymbol{u} \rangle \iff \tag{G.58}$$
$$\dot{\xi}_t = \tilde{\gamma}_j(\boldsymbol{y}_t) - \xi_t + \delta_{\xi_t}, \quad \delta_{\xi_t} := \frac{1}{\ell}\langle \boldsymbol{E}_t, \boldsymbol{u} \rangle \tag{G.59}$$

yielding the first equation in Eq. (G.51), the parallel dynamics along $L_{t,\varepsilon}^{(i,j)}$. Furthermore, by Eq. (G.58), we have that:

$$\ell\dot{\xi}_t\boldsymbol{u} = (\tilde{\gamma}_j(\boldsymbol{y}_t) - \xi_t)\ell\boldsymbol{u} + \langle \boldsymbol{E}_t, \boldsymbol{u} \rangle\boldsymbol{u} \tag{G.60}$$

Thus, subtracting $\ell\dot{\xi}_t\boldsymbol{u}$ from the RHS of Eq. (G.54) and then plugging in Eq. (G.60) yields:

$$\dot{\boldsymbol{w}}_t = (\tilde{\gamma}_j(\boldsymbol{y}_t) - \xi_t)\ell\boldsymbol{u} - \boldsymbol{w}_t + \boldsymbol{E}_t - ((\tilde{\gamma}_j(\boldsymbol{y}_t) - \xi_t)\ell\boldsymbol{u} + \langle \boldsymbol{E}_t, \boldsymbol{u} \rangle\boldsymbol{u}) \tag{G.61}$$
$$= -\boldsymbol{w}_t + \boldsymbol{E}_t - \langle \boldsymbol{E}_t, \boldsymbol{u} \rangle\boldsymbol{u} \tag{G.62}$$

Note that $\langle \boldsymbol{E}_t, \boldsymbol{u} \rangle \boldsymbol{u} = \mathrm{Proj}_{(\boldsymbol{u})}(\boldsymbol{E}_t) =: \mathrm{Proj}_{\|}(\boldsymbol{E}_t)$, yielding that $\boldsymbol{E}_t - \langle \boldsymbol{E}_t, \boldsymbol{u} \rangle \boldsymbol{u} =: \mathrm{Proj}_{\perp}(\boldsymbol{E}_t)$. Incorporating this into Eq. (G.62) yields:

$$\dot{\boldsymbol{w}}_t = -\boldsymbol{w}_t + \boldsymbol{\delta}_{\boldsymbol{w}_t}, \quad \boldsymbol{\delta}_{\boldsymbol{w}_t} := \mathrm{Proj}_{\perp}(\boldsymbol{E}_t). \tag{G.63}$$

This yields the second equation in Eq. (G.51), the orthogonal dynamics with respect to $L_{t,\varepsilon}^{(i,j)}$. To prove the lemma, it thus suffices to bound $|\delta_{\xi_t}|$ and $||\boldsymbol{\delta}_{\boldsymbol{w}_t}||_2$. To do so, we begin by bounding the responsibilities $\tilde{\gamma}_k(\boldsymbol{y}_t)$ for $k \notin \{i, j\}$:

$$\tilde{\gamma}_k(\boldsymbol{y}_t) = \frac{\pi_k \mathcal{N}(\boldsymbol{y}_t; \boldsymbol{\mu}^{(k)}, \tilde{\sigma}_t^2 \boldsymbol{I})}{\sum_{m=1}^{N} \pi_m \mathcal{N}(\boldsymbol{y}_t; \boldsymbol{\mu}^{(m)}, \tilde{\sigma}_t^2 \boldsymbol{I})} \tag{G.64}$$

$$\leq \frac{\pi_k \mathcal{N}(\boldsymbol{y}_t; \boldsymbol{\mu}^{(k)}, \tilde{\sigma}_t^2 \boldsymbol{I})}{\pi_i \mathcal{N}(\boldsymbol{y}_t; \boldsymbol{\mu}^{(i)}, \tilde{\sigma}_t^2 \boldsymbol{I}) + \pi_j \mathcal{N}(\boldsymbol{y}_t; \boldsymbol{\mu}^{(j)}, \tilde{\sigma}_t^2 \boldsymbol{I})}. \tag{G.65}$$

Let $d^* := \arg\min_{d \in \{i,j\}} ||\boldsymbol{y}_t - \boldsymbol{\mu}^{(d)}||_2$. Note that:

$$\pi_i \mathcal{N}(\boldsymbol{y}_t; \boldsymbol{\mu}^{(i)}, \tilde{\sigma}_t^2 \boldsymbol{I}) + \pi_j \mathcal{N}(\boldsymbol{y}_t; \boldsymbol{\mu}^{(j)}, \tilde{\sigma}_t^2 \boldsymbol{I}) \geq \max\{\pi_i \mathcal{N}(\boldsymbol{y}_t; \boldsymbol{\mu}^{(i)}, \tilde{\sigma}_t^2 \boldsymbol{I}), \pi_j \mathcal{N}(\boldsymbol{y}_t; \boldsymbol{\mu}^{(j)}, \tilde{\sigma}_t^2 \boldsymbol{I})\} \tag{G.66}$$

$$\geq \pi_{d^*} \mathcal{N}(\boldsymbol{y}_t; \boldsymbol{\mu}^{(d^*)}, \tilde{\sigma}_t^2 \boldsymbol{I}) \tag{G.67}$$

Incorporating Eq. (G.67) into Eq. (G.65) yields:

$$\tilde{\gamma}_k(\boldsymbol{y}_t) \leq \frac{\pi_k \mathcal{N}(\boldsymbol{y}_t; \boldsymbol{\mu}^{(k)}, \tilde{\sigma}_t^2 \boldsymbol{I})}{\pi_{d^*} \mathcal{N}(\boldsymbol{y}_t; \boldsymbol{\mu}^{(d^*)}, \tilde{\sigma}_t^2 \boldsymbol{I})} \tag{G.68}$$

$$= \frac{\pi_k}{\pi_{d^*}} \frac{\exp(-\frac{||\boldsymbol{y}_t - \boldsymbol{\mu}^{(k)}||_2^2}{2\tilde{\sigma}_t^2})}{\exp(-\frac{||\boldsymbol{y}_t - \boldsymbol{\mu}^{(d^*)}||_2^2}{2\tilde{\sigma}_t^2})} \tag{G.69}$$

$$= \frac{\pi_k}{\pi_{d^*}} \exp\left(-\frac{1}{2\tilde{\sigma}_t^2}(||\boldsymbol{y}_t - \boldsymbol{\mu}^{(k)}||_2^2 - ||\boldsymbol{y}_t - \boldsymbol{\mu}^{(d^*)}||_2^2)\right). \tag{G.70}$$

By Asm. 4.1, we have that $||\boldsymbol{y}_t - \boldsymbol{\mu}^{(k)}||_2^2 \geq ||\boldsymbol{y}_t - \boldsymbol{\mu}^{(d^*)}||_2^2 + \frac{\Delta_t}{\bar{\alpha}_t}$, which holds if and only if $||\boldsymbol{y}_t - \boldsymbol{\mu}^{(k)}||_2^2 - ||\boldsymbol{y}_t - \boldsymbol{\mu}^{(d^*)}||_2^2 \geq \frac{\Delta_t}{\bar{\alpha}_t}$. Incorporating this into Eq. (G.70) yields:

$$\tilde{\gamma}_k(\boldsymbol{y}_t) \leq \frac{\pi_k}{\pi_{d^*}} \exp(-\frac{\Delta_t}{2\bar{\alpha}_t \tilde{\sigma}_t^2}) \tag{G.71}$$

$$\leq \frac{\pi_k}{\pi_{d^*}} \exp(-\kappa), \quad \text{Asm. 4.1} \tag{G.72}$$

Next, we have that:

$$|\delta_{\xi_t}| = |\frac{1}{\ell}\langle \boldsymbol{E}_t, \boldsymbol{u} \rangle| \tag{G.73}$$

$$\leq \frac{1}{\ell}||\boldsymbol{E}_t||_2 ||\boldsymbol{u}||_2, \quad \text{Cauchy-Schwartz} \tag{G.74}$$

$$= \frac{1}{\ell}||\boldsymbol{E}_t||_2 \tag{G.75}$$

$$= \frac{1}{\ell}||\sum_{k \notin \{i,j\}} \tilde{\gamma}_k(\boldsymbol{y}_t)(\boldsymbol{\mu}^{(k)} - \boldsymbol{\mu}^{(i)})||_2 \tag{G.76}$$

$$\leq \frac{\max_{k \notin \{i,j\}} ||\boldsymbol{\mu}^{(k)} - \boldsymbol{\mu}^{(i)}||_2}{\ell} \sum_{k \notin \{i,j\}} \tilde{\gamma}_k(\boldsymbol{y}_t) \tag{G.77}$$

$$\leq \frac{\max_{k \notin \{i,j\}} ||\boldsymbol{\mu}^{(k)} - \boldsymbol{\mu}^{(i)}||_2}{\ell}(N-2)\frac{\pi_k}{\pi_{d^*}} \exp(-\kappa), \quad \text{Eq. (G.72)} \tag{G.78}$$

$$\leq \mathcal{O}(N \exp(-\kappa)) \tag{G.79}$$

Similarly, we have that:

$$||\boldsymbol{\delta}_{\boldsymbol{w}_t}||_2 \leq ||\text{Proj}_{\perp}(\boldsymbol{E})||_2 \tag{G.80}$$

$$\leq ||\boldsymbol{E}||_2, \text{ property of orth. projection} \tag{G.81}$$

$$\leq \sum_{k \notin \{i,j\}} \tilde{\gamma}_k(\boldsymbol{y}_t)||\boldsymbol{\mu}^{(k)} - \boldsymbol{\mu}^{(i)}||_2 \tag{G.82}$$

$$\leq (N-2) \max_{k \notin \{i,j\}} ||\boldsymbol{\mu}^{(k)} - \boldsymbol{\mu}^{(i)}||_2 \frac{\pi_i}{\pi_d} \exp(-\kappa), \text{ Eq. (G.72)} \tag{G.83}$$

$$\leq \mathcal{O}(N \exp(-\kappa)) \tag{G.84}$$

as desired.

$\square$

**Lemma G.4.** *Assume Asm. 4.1 holds. Then, there exists a time $\tau \leq \tau_1$ such that $\xi_t$, as defined by Eq. (G.51), lies in $[-\epsilon, 1 + \epsilon]$, where $\epsilon \in O(N \exp(-\kappa))$, for all time $t \leq \tau$.*

*Proof.* Recall the definition of an $\varepsilon$-extended line segment in Sec. 4. By enlarging the constant in $\varepsilon \in O(N \exp(-\kappa))$, we may assume that $|\delta_{\xi_t}| \leq \varepsilon/2$ by Lemma G.3. Since $\tilde{\gamma}_j(\boldsymbol{y}_t) \in (0, 1)$, if $\xi_t > 1 + \varepsilon$ and $r_t := \xi_t - (1 + \varepsilon)$, then $\dot{r}_t = \dot{\xi}_t \leq -r_t - \varepsilon/2$; similarly, if $\xi_t < -\varepsilon$ and $r_t := -\varepsilon - \xi_t$, then $\dot{r}_t = -\dot{\xi}_t \leq -r_t - \varepsilon/2$. Thus, for $t$ sufficiently large, $r_t$ vanishes and $\xi_t$ is trapped in $[-\varepsilon, 1 + \varepsilon]$. $\square$

**Lemma G.5.** *For any $\phi \in (0, 1)$, with probability at least $1 - 2 \exp(-(\varpi)\phi^2/8)$, we have that $||\boldsymbol{w}_T||_2$, where $\boldsymbol{w}_t$ is defined as in Eq. (G.51), satisfies:*

$$||\boldsymbol{w}_T||_2 \leq \sqrt{(\varpi)(1 + \phi)} \tag{G.85}$$

*Proof.* Apply standard tail bound for $\boldsymbol{x}_T \sim \mathcal{N}(0, \boldsymbol{I})$, since $||\boldsymbol{w}_T||_2^2 \sim \chi_{\varpi}^2$ (Lemma 1 in Laurent & Massart (2000)). $\square$

**Lemma G.6** (Brownian confinement in an interval). *Let $(B_t)_{t \geq 0}$ be standard one-dimensional Brownian motion. Then for any $V > 0$ and $a > 0$,*

$$\mathbb{P}\Big( \sup_{0 \leq t \leq V} |B_t| \leq a \Big) \leq \frac{4}{\pi} \exp\Big( -\frac{\pi^2 V}{8a^2} \Big). \tag{G.86}$$

*Proof.* See Eq 7.15 in (Mörters & Peres, 2010) and evaluate at the midpoint keeping the first eigenvalue term. $\square$

## H. Proofs

### H.1. Proof of Theorem 4.2

*Proof.* Consider $\tau \leq \tau_1$ such that Lemma G.4 holds, and then fix $t \leq \tau$. Consider the dynamics in Lemma G.2, which are equivalent to those in Eq. (3) and are defined with respect to the rescaled time:

$$u(t) := \int_t^T \frac{\beta_s}{2\sigma_s^2} \, ds. \tag{H.87}$$

Furthermore, consider the parallel and orthogonal dynamics as defined in Lemma G.3, except parameterized by the $u(t)$ as they were defined originally per Lemma G.3. Note that $u(t)$ is monotonically *decreasing* for forward $t$, since:

$$\frac{d}{dt} u(t) = -\frac{\beta_t}{2\sigma_t^2} < 0, \tag{H.88}$$

by definition of $\beta_t$ and $\sigma_t^2$; in particular, it is monotonically *increasing* at our studied reverse $t$.

Accordingly, the dynamics in Lemma G.2 are:

$$\frac{d}{du}\xi_{u(t)} = \gamma_j(\boldsymbol{y}_{u(t)}) - \xi_{u(t)} + \delta_{\xi_{u(t)}}, \tag{H.89}$$

$$\frac{d}{du}\boldsymbol{w}_{u(t)} = -\boldsymbol{w}_{u(t)} + \boldsymbol{\delta}_{\boldsymbol{w}_{u(t)}}, \tag{H.90}$$

where $|\delta_{\xi_u}|, \|\boldsymbol{\delta}_{\boldsymbol{w}_u}\|_2 \leq \varepsilon$.

Then, by definition of the orthogonal distance (and since, by Lemma G.2, in $\boldsymbol{y}$-space the $(i,j)$-mode segment is static, i.e. $L_{u(t)} = L_0$ for all $t$:),

$$d_\perp(\boldsymbol{y}_{u(t)}, L_{0,\varepsilon}^{(i,j)}) = \|\boldsymbol{w}_{u(t)}\|_2, \tag{H.91}$$

since the closest point on the extended line segment is the orthogonal projection. Note that the following inequality follows vacuously if $\|\boldsymbol{w}_{u(t)}\|_2 = 0$, hence we consider $\|\boldsymbol{w}_{u(t)}\|_2 \neq 0$. Define $g(u) := \|\boldsymbol{w}_u\|_2$, where $u := u(t)$.

In the first case that $g(u) \neq 0$, we have

$$g(u)^2 = \boldsymbol{w}_u^\top \boldsymbol{w}_u \quad \implies \quad \frac{d}{du}g(u)^2 = \frac{d}{du}\left(\boldsymbol{w}_u^\top \boldsymbol{w}_u\right) = 2\,\boldsymbol{w}_u^\top \frac{d\boldsymbol{w}_u}{du}. \tag{H.92}$$

Using the orthogonal dynamics $\frac{d\boldsymbol{w}_u}{du} = -\boldsymbol{w}_u + \boldsymbol{\delta}_{\boldsymbol{w}_u}$ and $\|\boldsymbol{\delta}_{\boldsymbol{w}_u}\|_2 \leq \varepsilon$, we obtain

$$\frac{d}{du}g(u)^2 = 2\,\boldsymbol{w}_u^\top(-\boldsymbol{w}_u + \boldsymbol{\delta}_{\boldsymbol{w}_u}) = -2\|\boldsymbol{w}_u\|_2^2 + 2\,\boldsymbol{w}_u^\top \boldsymbol{\delta}_{\boldsymbol{w}_u} \tag{H.93}$$

$$\leq -2g(u)^2 + 2\|\boldsymbol{w}_u\|_2\|\boldsymbol{\delta}_{\boldsymbol{w}_u}\|_2 \leq -2g(u)^2 + 2\varepsilon g(u). \tag{H.94}$$

Since $\frac{d}{du}g(u)^2 = 2g(u)g'(u)$ for $g(u) \neq 0$, dividing by $2g(u)$ yields

$$g'(u) \leq -g(u) + \varepsilon. \tag{H.95}$$

In the other case that $g(u) = 0$, the inequality holds trivially. Multiplying by $e^u$ then yields:

$$\frac{d}{du}(e^u g(u)) \leq \varepsilon e^u. \tag{H.96}$$

Integrating from $0$ to $u$ then gives:

$$e^u g(u) - g(0) \leq \varepsilon(e^u - 1), \tag{H.97}$$

hence:

$$g(u) \leq e^{-u}g(0) + \varepsilon(1 - e^{-u}) \leq e^{-u}g(0) + \varepsilon. \tag{H.98}$$

In particular, denoting $z : u(z) = 0$ to mean the time $z \in [0, T]$ s.t. $u(z) = 0$, we have that:

$$\|\boldsymbol{w}_{u(t)}\|_2 \leq e^{-u(t)}\|\boldsymbol{w}_{z:u(z)=0}\|_2 + \varepsilon(1 - e^{-u(t)}) \leq e^{-u(t)}\|\boldsymbol{w}_T\|_2 + \varepsilon. \tag{H.99}$$

In particular, by definition of $u$, $z = T$, yielding:

$$\|\boldsymbol{w}_{u(t)}\|_2 \leq e^{-u(t)}\|\boldsymbol{w}_T\|_2 + \varepsilon(1 - e^{-u(t)}) \leq e^{-u(t)}\|\boldsymbol{w}_T\|_2 + \varepsilon. \tag{H.100}$$

By definition of $\boldsymbol{y}_{u(t)}$ in Lemma G.2, we have that $d_\perp(\boldsymbol{x}_t, L_{t,\varepsilon}^{(i,j)}) = \sqrt{\bar{\alpha}_t}\, d_\perp(\boldsymbol{y}_{u(t)}, L_{0,\varepsilon}^{(i,j)})$. Applying this to Eq. (H.100),

we obtain that:

$$d_\perp(\boldsymbol{x}_t, L_{t,\varepsilon}^{(i,j)}) = \sqrt{\bar{\alpha}_t}\, d_\perp(\boldsymbol{y}_{u(t)}, L_{0,\varepsilon}^{(i,j)}) \tag{H.101}$$

$$= \sqrt{\bar{\alpha}_t}\, \|\boldsymbol{w}_{u(t)}\|_2 \tag{H.102}$$

$$\leq \sqrt{\bar{\alpha}_t}\left(e^{-u(t)}\|\boldsymbol{w}_T\|_2 + \varepsilon\right) \tag{H.103}$$

$$= \sqrt{\bar{\alpha}_t}\exp(-u(t))d_\perp(\boldsymbol{y}_T, L_{T,\varepsilon}^{(i,j)}) + \sqrt{\bar{\alpha}_t}\varepsilon \tag{H.104}$$

$$= \frac{\sqrt{\bar{\alpha}_t}}{\sqrt{\bar{\alpha}_T}}\exp(-u(t))d_\perp(\boldsymbol{x}_T, L_{T,\varepsilon}^{(i,j)}) + \sqrt{\bar{\alpha}_t}\varepsilon \tag{H.105}$$

$$= \sqrt{\frac{\bar{\alpha}_t}{\gamma}}\exp(-u(t))d_\perp(\boldsymbol{x}_T, L_{T,\varepsilon}^{(i,j)}) + \sqrt{\bar{\alpha}_t}\varepsilon, \tag{H.106}$$

$$\leq \mathcal{O}(\exp(-u(t))d_\perp(\boldsymbol{x}_T, L_{T,\varepsilon}^{(i,j)}) + \varepsilon) \tag{H.107}$$

by definition of $\bar{\alpha}_T$ and $u(t)$. To obtain the result without the initial distance, note that $\|\boldsymbol{w}_T\|_2 \leq \|\boldsymbol{y}_T\|_2 \leq \frac{\sqrt{\varpi(1+\phi)}}{\sqrt{\bar{\alpha}_T}} = \frac{\sqrt{\varpi(1+\phi)}}{\sqrt{\gamma}}$ since $\boldsymbol{w}_T$ is a component of $\boldsymbol{y}_T$, and by applying Lemma G.5; note that the extra $\sqrt{\gamma}$ in the denominator is again absorbed by the $\mathcal{O}$. Finally, since $t$ was arbitrary, this holds for all $t \leq \tau$ as desired.

$\square$

## H.2. Proof of Corollary 4.3

*Proof.* From the proof of Theorem 4.2, we have that the dynamics of $\|\boldsymbol{w}_u\|_2$ satisfy the differential inequality:

$$\frac{d}{du}\|\boldsymbol{w}_u\|_2 \leq -\|\boldsymbol{w}_u\|_2 + \varepsilon \tag{H.108}$$

At the boundary of the tube, we thus have:

$$\frac{d}{du}\|\boldsymbol{w}_u\|_2\bigg|_{\|\boldsymbol{w}_u\|_2=\varepsilon} \leq -\varepsilon + \varepsilon = 0. \tag{H.109}$$

In particular, if $\|\boldsymbol{w}_u\|_2 \leq \varepsilon$, it cannot cross the boundary to the outside, because doing so would require a strictly positive derivative. Note that since $\sqrt{\bar{\alpha}_u}$ and its reciprocal are uniformly bounded on $[T - \tau_1, T]$, the rescaling only changes the tube radius by a multiplicative constant which we absorb into $\varepsilon$. Thus, together with Lemma G.4, the $\varepsilon$-tube around the line segment in Theorem 4.2 is invariant along the reverse trajectory; if a trajectory enters this set during the interval $t \leq \tau$, it stays there. $\square$

## H.3. Proof of Proposition 4.5

*Proof.* We work with the rescaled reverse-time dynamics from Lemma G.2 and Lemma G.3; as in Lemma G.2, we relabel the time variable $u$ as $t$. We begin by showing existence of an instantaneously stable equilibrium $\xi_t^{(i),*}$ in the $\varepsilon$-ball around $\boldsymbol{\mu}^{(i)}$, without loss of generality. The proof is symmetric for $\boldsymbol{\mu}^{(j)}$.

By Lemma G.3, we have the parallel dynamics:

$$\dot{\xi}_t = \tilde{\gamma}_j(\boldsymbol{y}_t) - \xi_t + \delta_{\xi_t}, \tag{H.110}$$

where:

$$\boldsymbol{y}_t = \boldsymbol{\mu}^{(i)} + \xi_t \ell \boldsymbol{u} + \boldsymbol{w}_t, \tag{H.111}$$

where $\ell = \|\boldsymbol{\mu}^{(j)} - \boldsymbol{\mu}^{(i)}\|_2$, $\boldsymbol{u} = \frac{\boldsymbol{\mu}^{(j)} - \boldsymbol{\mu}^{(i)}}{\ell}$, and $\boldsymbol{w}_t$ satisfies $\langle \boldsymbol{w}_t, \boldsymbol{u} \rangle = 0$. Furthermore, since the trajectory is in the $\varepsilon$-tube around $L_t^{(i,j)}$, we have that $\|\boldsymbol{w}_t\|_2 \leq \varepsilon$.

Fix $\xi := \xi_t$ for $t \leq \hat{\tau}$ arbitrary. Then, we have that:

$$\|\boldsymbol{y}_t - \boldsymbol{\mu}^{(i)}\|_2^2 = \|\boldsymbol{\mu}^{(i)} - \boldsymbol{\mu}^{(i)} + \xi\ell\boldsymbol{u} + \boldsymbol{w}_t\|_2^2 \tag{H.112}$$

$$= \|\xi\ell\boldsymbol{u} + \boldsymbol{w}_t\|_2^2 \tag{H.113}$$

$$= \|\xi\ell\boldsymbol{u}\|_2^2 + \|\boldsymbol{w}_t\|_2^2 + 2\langle\xi\ell\boldsymbol{u}, \boldsymbol{w}_t\rangle \tag{H.114}$$

$$= \xi^2\ell^2\|\boldsymbol{u}\|_2^2 + \|\boldsymbol{w}_t\|_2^2, \ \langle\boldsymbol{u}, \boldsymbol{w}_t\rangle = 0 \tag{H.115}$$

$$= \xi^2\ell^2\|\boldsymbol{u}\|_2^2 + \|\boldsymbol{w}_t\|_2^2, \ \|\boldsymbol{u}\|_2 = 1. \tag{H.116}$$

Next, we have that:

$$\|\boldsymbol{y}_t - \boldsymbol{\mu}^{(j)}\|_2^2 = \|\boldsymbol{\mu}^{(i)} - \boldsymbol{\mu}^{(j)} + \xi\ell\boldsymbol{u} + \boldsymbol{w}_t\|_2^2 \tag{H.117}$$

$$= \|\boldsymbol{\mu}^{(i)} - (\boldsymbol{\mu}^{(i)} + \ell\boldsymbol{u}) + \xi\ell\boldsymbol{u} + \boldsymbol{w}_t\|_2^2 \tag{H.118}$$

$$= \|(\xi - 1)\ell\boldsymbol{u} + \boldsymbol{w}_t\|_2^2 \tag{H.119}$$

$$= \|(\xi - 1)\ell\|_2^2 + \|\boldsymbol{w}_t\|_2^2 + 2\langle(\xi - 1)\ell\boldsymbol{u}, \boldsymbol{w}_t\rangle \tag{H.120}$$

$$= (1 - \xi)^2\ell^2 + \|\boldsymbol{w}_t\|_2^2, \tag{H.121}$$

by the same token, since $\|\boldsymbol{u}\|_2^2 = 1$ and $\langle\boldsymbol{u}, \boldsymbol{w}_t\rangle = 0$.

Incorporating Eq. (H.116) and Eq. (H.121), we have that:

$$\|\boldsymbol{y}_t - \boldsymbol{\mu}^{(j)}\|_2^2 - \|\boldsymbol{y}_t - \boldsymbol{\mu}^{(i)}\|_2^2 = (1 - \xi)^2\ell^2 + \|\boldsymbol{w}_t\|_2^2 - (\xi^2\ell^2 + \|\boldsymbol{w}_t\|_2^2) \tag{H.122}$$

$$= (1 - \xi)^2\ell^2 - \xi^2\ell^2 \tag{H.123}$$

$$= ((1 - \xi)^2 - \xi^2)\ell^2 \tag{H.124}$$

$$= (1 - 2\xi)\ell^2. \tag{H.125}$$

We then bound the responsibility of mode $j$ as:

$$\tilde{\gamma}_j(\boldsymbol{y}_t) = \frac{\pi_j\mathcal{N}(\boldsymbol{y}_t; \boldsymbol{\mu}^{(j)}, \tilde{\sigma}_t^2\boldsymbol{I})}{\sum_{m=1}^N \pi_m\mathcal{N}(\boldsymbol{y}_t; \boldsymbol{\mu}^{(m)}, \tilde{\sigma}_t^2\boldsymbol{I})} \tag{H.126}$$

$$\leq \frac{\pi_j\mathcal{N}(\boldsymbol{y}_t; \boldsymbol{\mu}^{(j)}, \tilde{\sigma}_t^2\boldsymbol{I})}{\pi_i\mathcal{N}(\boldsymbol{y}_t; \boldsymbol{\mu}^{(i)}, \tilde{\sigma}_t^2\boldsymbol{I}) + \pi_j\mathcal{N}(\boldsymbol{y}_t; \boldsymbol{\mu}^{(j)}, \tilde{\sigma}_t^2\boldsymbol{I})} \tag{H.127}$$

$$= \frac{\pi_j\exp\left(-\frac{\|\boldsymbol{y}_t - \boldsymbol{\mu}^{(j)}\|_2^2}{2\tilde{\sigma}_t^2}\right)}{\pi_i\exp\left(-\frac{\|\boldsymbol{y}_t - \boldsymbol{\mu}^{(i)}\|_2^2}{2\tilde{\sigma}_t^2}\right) + \pi_j\exp\left(-\frac{\|\boldsymbol{y}_t - \boldsymbol{\mu}^{(j)}\|_2^2}{2\tilde{\sigma}_t^2}\right)} \tag{H.128}$$

$$= \frac{1}{1 + \frac{\pi_i}{\pi_j}(\exp\left(\frac{\|\boldsymbol{y}_t - \boldsymbol{\mu}^{(j)}\|_2^2 - \|\boldsymbol{y}_t - \boldsymbol{\mu}^{(i)}\|_2^2}{2\tilde{\sigma}_t^2}\right)}, \ \text{dividing numerator and denominator} \tag{H.129}$$

$$\leq \frac{\pi_j}{\pi_i}\exp\left(-\frac{\|\boldsymbol{y}_t - \boldsymbol{\mu}^{(j)}\|_2^2 - \|\boldsymbol{y}_t - \boldsymbol{\mu}^{(i)}\|_2^2}{2\tilde{\sigma}_t^2}\right) \tag{H.130}$$

$$\leq \frac{\pi_j}{\pi_i}\exp\left(-\frac{(1 - 2\xi)\ell^2}{2\tilde{\sigma}_t^2}\right) \tag{H.131}$$

Next, suppose that $\xi \in [-a, a]$ for $a \in \mathcal{O}(N\exp(-\kappa))$ satisfying $a \leq \frac{1}{4}$, to be chosen later. Then, $1 - 2a \geq \frac{1}{2}$, and thus:

$$\varepsilon_{\tilde{\gamma}} = \sup_{\xi \in [-a,a]} \tilde{\gamma}_j(\boldsymbol{y}_t) \tag{H.132}$$

$$\leq \frac{\pi_j}{\pi_i} \exp\left(-\frac{(1-2a)\ell^2}{2\tilde{\sigma}_t^2}\right) \tag{H.133}$$

$$\leq \frac{\pi_j}{\pi_i} \exp\left(-\frac{(1-2a)\ell^2}{2\tilde{\sigma}_t^2}\right) \tag{H.134}$$

$$\leq \frac{\pi_j}{\pi_i} \exp\left(-(1-2a)2\kappa\right), \ \text{Asm. 4.4} \tag{H.135}$$

$$\in \mathcal{O}(\exp(-\kappa)), \ 1-2a \geq \frac{1}{2}. \tag{H.136}$$

Then, by the above, $\varepsilon_{\tilde{\gamma}} \in \mathcal{O}(N\exp(-\kappa))$. Furthermore, by Lemma G.3, we have that $\varepsilon_{\delta_\xi} := \sup_{\xi \in [-a,a]} |\delta_\xi| \leq \sup_{\xi \in [-a,1+a]} |\delta_\xi| \in \mathcal{O}(N\exp(-\kappa))$. Thus, let $a := 2(\varepsilon_{\delta_\xi} + \varepsilon_{\tilde{\gamma}})$, which satisfies $a \in \mathcal{O}(N\exp(-\kappa))$. Since $a \in \mathcal{O}(N\exp(-\kappa))$ there exists a $C_a > 0$ s.t. $a \leq C_a N\exp(-\kappa)$ for $N$ sufficiently large; choosing $\kappa \geq \log N + \log(4C_a)$ ensures that $a \leq \frac{1}{4}$.

We then evaluate the behavior of $\xi$ at the endpoints. At $\xi = -a$, we have that:

$$F_t(-a) = \tilde{\gamma}_j(\boldsymbol{y}_t(-a)) + a + \delta_{-a} \tag{H.137}$$

$$\geq a + \delta_{-a}, \ \tilde{\gamma}_j \geq 0 \tag{H.138}$$

$$\geq a - |\delta_a|, \ x \geq -|x| \text{ for } x \in \mathbb{R} \tag{H.139}$$

$$\geq a - \varepsilon_{\delta_a}, \delta_a \leq \varepsilon_{\delta_a} \tag{H.140}$$

$$= 2\varepsilon_{\delta_a} + 2\varepsilon_{\tilde{\gamma}} - \varepsilon_{\delta_a} \tag{H.141}$$

$$\geq \varepsilon_{\delta_a} + \varepsilon_{\tilde{\gamma}} \tag{H.142}$$

$$> 0. \tag{H.143}$$

Next, at $\xi = a$, we have that:

$$F_t(a) = \tilde{\gamma}_j(\boldsymbol{y}_t(a)) - a + \delta_a \tag{H.144}$$

$$\leq \varepsilon_{\tilde{\gamma}} - a + \varepsilon_{\delta_a}, \ \tilde{\gamma}_j \leq \varepsilon_{\tilde{\gamma}} \text{ and } \delta_a \leq \varepsilon_{\delta_a} \tag{H.145}$$

$$= \varepsilon_{\tilde{\gamma}} - 2\varepsilon_{\delta_a} - 2\varepsilon_{\tilde{\gamma}} + \varepsilon_{\delta_a} \tag{H.146}$$

$$= -(\varepsilon_{\delta_a} + \varepsilon_{\tilde{\gamma}}) \tag{H.147}$$

$$< 0. \tag{H.148}$$

Since $F_t$ is continuous in $\xi$, since it is a composition of continuous functions (responsibilities are compositions of continuous Gaussian p.d.f.s, and hence continuous), by the intermediate value theorem there exists a $\xi_t^{*,i} \in (-a, a)$ s.t. $F_t(\xi_t^{*,i}) = 0$. Furthermore:

$$||\boldsymbol{y}_t(\xi_t^{*,i}) - \boldsymbol{\mu}^{(i)}||_2 \leq |\xi_t^{*,i}|\ell + ||\boldsymbol{w}||_2, \ \text{triangle inequality} \tag{H.149}$$

$$\leq a\ell + \varepsilon \tag{H.150}$$

$$\in \mathcal{O}(N\exp(-\kappa)), \tag{H.151}$$

i.e. there exists an instantaneous equilibria of Eq. (G.50) in a $\varepsilon$-ball of $\boldsymbol{\mu}^{(i)}$, where $\varepsilon \in \mathcal{O}(N\exp(-\kappa))$. Since $\xi$ was arbitrary, this holds for all $t \leq \hat{\tau}$.

Next, we show the (uniform) instantaneous stability of $\xi_t^{*,i}$, without loss of generality; the proof is symmetric for $\xi_t^{*,j}$. Fix $t \leq \hat{\tau}$. Then, we have that:

$$\frac{d}{d\xi_t} F_t(\xi_t^{*,i}) = \left( \frac{d}{d\xi_t} \tilde{\gamma}_j(\boldsymbol{y}_t) + \frac{d}{d\xi_t} \delta_{\xi_t} - 1 \right) |_{\xi_t = \xi_t^{*,i}}. \tag{H.152}$$

We demonstrate that $F_t'(\xi_t^{*,i}) < 0$. Fix time $t \leq \hat{\tau}$. We have that:

$$\nabla_{\boldsymbol{y}_t} \tilde{\gamma}_j(\boldsymbol{y}_t) = \frac{\tilde{\gamma}_j(\boldsymbol{y}_t)}{\tilde{\sigma}_t^2}(\boldsymbol{\mu}^{(j)} - \hat{\boldsymbol{\mu}}) \tag{H.153}$$

where $\hat{\boldsymbol{\mu}}$ is the same as that in Lemma G.2. Hence, we have that:

$$\frac{d}{d\xi_t} \tilde{\gamma}_j(\boldsymbol{y}_t) = \langle \nabla_{\boldsymbol{y}_t} \tilde{\gamma}_j(\boldsymbol{y}_t), \frac{d\boldsymbol{y}_t}{d\xi_t} \rangle, \text{ chain rule} \tag{H.154}$$

$$= \frac{\ell \tilde{\gamma}_j(\boldsymbol{y}_t)}{\tilde{\sigma}_t^2} \langle \boldsymbol{\mu}^{(j)} - \hat{\boldsymbol{\mu}}(\boldsymbol{y})), \boldsymbol{u} \rangle \text{ Eq. (H.153)} \tag{H.155}$$

To bound the inner product, note that, by linearity of the inner product:

$$\langle \boldsymbol{\mu}^{(j)} - \hat{\boldsymbol{\mu}}(\boldsymbol{y}_t), \boldsymbol{u} \rangle = \langle \boldsymbol{\mu}^{(j)} - \boldsymbol{\mu}^{(i)}, \boldsymbol{u} \rangle + \langle \boldsymbol{\mu}^{(i)} - \hat{\boldsymbol{\mu}}(\boldsymbol{y}_t), \boldsymbol{u} \rangle \tag{H.156}$$

This implies:

$$|\langle \boldsymbol{\mu}^{(j)} - \hat{\boldsymbol{\mu}}(\boldsymbol{y}_t), \boldsymbol{u} \rangle| \leq |\langle \boldsymbol{\mu}^{(j)} - \boldsymbol{\mu}^{(i)}, \boldsymbol{u} \rangle| + |\langle \boldsymbol{\mu}^{(i)} - \hat{\boldsymbol{\mu}}(\boldsymbol{y}_t), \boldsymbol{u} \rangle|, \text{ triangle inequality} \tag{H.157}$$

$$\leq ||\boldsymbol{\mu}^{(j)} - \boldsymbol{\mu}^{(i)}||_2 ||\boldsymbol{u}||_2 + ||\boldsymbol{\mu}^{(i)} - \hat{\boldsymbol{\mu}}(\boldsymbol{y}_t)||_2 ||\boldsymbol{u}||_2, \text{ Cauchy-Schwartz} \tag{H.158}$$

$$\leq \ell + || \sum_{k \in [N]} \tilde{\gamma}_k(\boldsymbol{y}_t)(\boldsymbol{\mu}^{(k)} - \boldsymbol{\mu}^{(i)})||_2, \ ||\boldsymbol{u}||_2 = 1 \tag{H.159}$$

$$\leq \ell + \tilde{\gamma}_j(\boldsymbol{y}_t)||\boldsymbol{\mu}^{(j)} - \boldsymbol{\mu}^{(i)}||_2 + || \sum_{k \in [N]} \tilde{\gamma}_k(\boldsymbol{y}_t)(\boldsymbol{\mu}^{(k)} - \boldsymbol{\mu}^{(i)})||_2, \text{ triangle inequality} \tag{H.160}$$

$$\leq \ell + \tilde{\gamma}_j \ell + \max_{k \notin \{i,j\}} ||\boldsymbol{\mu}^{(k)} - \boldsymbol{\mu}^{(i)}||_2 \sum_{k \notin \{i,j\}} \tilde{\gamma}_k(\boldsymbol{y}_t) \tag{H.161}$$

$$\leq 2\ell + \max_{k \notin \{i,j\}} ||\boldsymbol{\mu}^{(k)} - \boldsymbol{\mu}^{(i)}||_2 \sum_{k \notin \{i,j\}} \tilde{\gamma}_k(\boldsymbol{y}_t) \tag{H.162}$$

Incorporating Eq. (H.162) into Eq. (H.155) yields:

$$|\frac{d}{d\xi_t} \tilde{\gamma}_j(\boldsymbol{y}_t)| \leq \frac{2\ell^2}{\tilde{\sigma}_t^2} \tilde{\gamma}_j(\boldsymbol{y}_t) + \frac{max_{k \notin \{i,j\}} ||\boldsymbol{\mu}^{(k)} - \boldsymbol{\mu}^{(i)}||_2 \ell}{\tilde{\sigma}_t^2} \tilde{\gamma}_j(\boldsymbol{y}_t) \sum_{k \notin \{i,j\}} \tilde{\gamma}_k(\boldsymbol{y}_t) \tag{H.163}$$

$$\leq \frac{2\ell^2}{\tilde{\sigma}_t^2} \tilde{\gamma}_j(\boldsymbol{y}_t) + \frac{max_{k \notin \{i,j\}} ||\boldsymbol{\mu}^{(k)} - \boldsymbol{\mu}^{(i)}||_2 \ell^2}{\tilde{\sigma}_t^2} \tilde{\gamma}_j(\boldsymbol{y}_t) \sum_{k \notin \{i,j\}} \tilde{\gamma}_k(\boldsymbol{y}_t). \tag{H.164}$$

Recall that $\xi_t^{*,i} \in (-a, a)$ where $a \leq \frac{1}{4}$, and thus we have $(1 - 2a) \geq \frac{1}{2}$. Let $x := \frac{\ell^2}{2\tilde{\sigma}_t^2}$. We then have, by Eq. (H.135), that:

$$\tilde{\gamma}_j(\boldsymbol{y}_t) \leq \frac{\pi_j}{\pi_i}(\exp(-(1 - 2a)x) \leq \frac{\pi_j}{\pi_i} \exp(-x/2). \tag{H.165}$$

Thus:

$$\frac{2\ell^2}{\tilde{\sigma}_t^2} \tilde{\gamma}_j(\boldsymbol{y}_t) = 4x\tilde{\gamma}_j(\boldsymbol{y}_t) \leq 4\frac{\pi_j}{\pi_i} x \exp(-x/2). \tag{H.166}$$

Furthermore, by Asm. 4.4, we have that $\frac{\ell^2}{2\tilde{\sigma}_t^2} \geq 2\kappa$, i.e. $x \geq 2\kappa$. Therefore, we have that $x \exp(-x/2) \leq \sup_{x \geq 2\kappa} x \exp(-x/2) = (2\kappa) \exp(-2\kappa/2) = 2\kappa \exp(-\kappa)$. Then, incorporating this and Eq. (H.166) into Eq. (H.164), we have that:

$$\sup_{\xi \in [-a,a]} |\frac{d}{d\xi_t}\tilde{\gamma}_j(\boldsymbol{y}_t)| \leq \sup_{\xi \in [-a,a]} \left( \frac{2\ell^2}{\tilde{\sigma}_t^2}\tilde{\gamma}_j(\boldsymbol{y}_t) + \frac{max_{k \notin \{i,j\}}||\boldsymbol{\mu}^{(k)} - \boldsymbol{\mu}^{(i)}||_2 \ell^2}{\tilde{\sigma}_t^2}\tilde{\gamma}_j(\boldsymbol{y}_t)\sum_{k \notin \{i,j\}}\tilde{\gamma}_k(\boldsymbol{y}_t) \right) \tag{H.167}$$

$$\leq 2\sup_{\xi \in [-a,a]}\sup_{\frac{\ell^2}{2\tilde{\sigma}_t^2} \geq \kappa}\frac{\ell^2}{\tilde{\sigma}_t^2}\tilde{\gamma}_j(\boldsymbol{y}_t) + \max_{k \notin \{i,j\}}||\boldsymbol{\mu}^{(k)} - \boldsymbol{\mu}^{(i)}||_2 \sup_{\xi \in [-a,a]}\sup_{\frac{\ell^2}{2\tilde{\sigma}_t^2} \geq \kappa}\frac{\ell^2}{\tilde{\sigma}_t^2}\tilde{\gamma}_j(\boldsymbol{y}_t)\sum_{k \notin \{i,j\}}\tilde{\gamma}_k(\boldsymbol{y}_t) \tag{H.168}$$

$$\leq 8\frac{\pi_j}{\pi_i}\kappa \exp(-\kappa) + \max_{k \notin \{i,j\}}2||\boldsymbol{\mu}^{(k)} - \boldsymbol{\mu}^{(i)}||_2\kappa \exp(-\kappa)\sum_{k \notin \{i,j\}}\tilde{\gamma}_k(\boldsymbol{y}_t) \tag{H.169}$$

$$\leq 8\frac{\pi_j}{\pi_i}\kappa \exp(-\kappa) + \max_{k \notin \{i,j\}}2||\boldsymbol{\mu}^{(k)} - \boldsymbol{\mu}^{(i)}||_2\kappa \exp(-\kappa)(N-2)\max_{k \notin \{i,j\}}\frac{\pi_k}{\pi_{d^*}}\exp(-\kappa), \text{ Eq. (G.72)}. \tag{H.170}$$

We next bound the derivative of $\delta_{\xi_t}$ for $\xi_t \in (-a, a)$ arbitrary.

We have that:

$$\frac{d}{d\xi_t}\delta_{\xi_t} = \frac{1}{\ell}\sum_{k \notin \{i,j\}}\frac{d}{d\xi_t}\tilde{\gamma}_k(\boldsymbol{y}_t)\langle \boldsymbol{\mu}^{(k)} - \boldsymbol{\mu}^{(i)}, \boldsymbol{u}\rangle \tag{H.171}$$

which yields that:

$$|\frac{d}{d\xi_t}\delta_{\xi_t}| \leq \frac{1}{\ell}\sum_{k \notin \{i,j\}}|\frac{d}{d\xi_t}\tilde{\gamma}_k(\boldsymbol{y}_t)||\langle \boldsymbol{\mu}^{(k)} - \boldsymbol{\mu}^{(i)}, \boldsymbol{u}\rangle|, \text{ triangle inequality} \tag{H.172}$$

$$\leq \frac{1}{\ell}\sum_{k \notin \{i,j\}}|\frac{d}{d\xi_t}\tilde{\gamma}_k(\boldsymbol{y}_t)|||\boldsymbol{\mu}^{(k)} - \boldsymbol{\mu}^{(i)}||_2||\boldsymbol{u}||_2 \tag{H.173}$$

$$\leq \frac{max_{k \notin \{i,j\}}||\boldsymbol{\mu}^{(k)} - \boldsymbol{\mu}^{(i)}||_2}{\ell}(N-2)\max_{k \notin \{i,j\}}|\frac{d}{d\xi_t}\tilde{\gamma}_k(\boldsymbol{y}_t)|. \tag{H.174}$$

We then have that:

$$|\frac{d}{d\xi_t}\tilde{\gamma}_k(\boldsymbol{y}_t)| = |\langle \nabla_{\boldsymbol{y}_t}\tilde{\gamma}_k(\boldsymbol{y}_t), \frac{d\boldsymbol{y}_t}{d\xi_t}\rangle|, \text{ chain rule} \tag{H.175}$$

$$= \frac{\ell\tilde{\gamma}_k(\boldsymbol{y}_t)}{\tilde{\sigma}_t^2}|\langle \boldsymbol{\mu}^{(k)} - \hat{\boldsymbol{\mu}}(\boldsymbol{y}_t), \boldsymbol{u}\rangle|, \tag{H.176}$$

$$\leq \frac{\ell\tilde{\gamma}_k(\boldsymbol{y}_t)}{\tilde{\sigma}_t^2}||\boldsymbol{\mu}^{(k)} - \hat{\boldsymbol{\mu}}(\boldsymbol{y}_t)||_2, \text{ Cauchy-Schwartz, } ||\boldsymbol{u}||_2 = 1. \tag{H.177}$$

$$\leq \frac{\ell}{\tilde{\sigma}_t^2}\frac{\pi_k}{\pi_{d^*}}\exp(-\kappa)||\boldsymbol{\mu}^{(k)} - \hat{\boldsymbol{\mu}}(\boldsymbol{y}_t)||_2, \text{ Eq. (G.72)} \tag{H.178}$$

Then, we have that:

$$||\boldsymbol{\mu}^{(k)} - \hat{\boldsymbol{\mu}}(\boldsymbol{y}_t)||_2 = ||\boldsymbol{\mu}^{(k)} - \hat{\boldsymbol{\mu}}(\boldsymbol{y}_t) - \boldsymbol{\mu}^{(i)} + \boldsymbol{\mu}^{(i)}||_2 \tag{H.179}$$

$$\leq ||\boldsymbol{\mu}^{(k)} - \boldsymbol{\mu}^{(i)}||_2 + ||\hat{\boldsymbol{\mu}}(\boldsymbol{y}_t) - \boldsymbol{\mu}^{(i)}||_2 \tag{H.180}$$

$$\leq \max_{k \notin \{i,j\}} ||\boldsymbol{\mu}^{(k)} - \boldsymbol{\mu}^{(i)}||_2 + ||\sum_{m \in [N]} \tilde{\gamma}_m(\boldsymbol{y}_t)\boldsymbol{\mu}^{(m)} - \boldsymbol{\mu}^{(i)}||_2 \tag{H.181}$$

$$\leq \max_{k \notin \{i,j\}} ||\boldsymbol{\mu}^{(k)} - \boldsymbol{\mu}^{(i)}||_2 + \tilde{\gamma}_j(\boldsymbol{y}_t)||\boldsymbol{\mu}^{(j)} - \boldsymbol{\mu}^{(i)}||_2 + ||\sum_{m \notin \{i,j\}} \tilde{\gamma}_m(\boldsymbol{y}_t)\boldsymbol{\mu}^{(m)} - \boldsymbol{\mu}^{(i)}||_2 \tag{H.182}$$

$$\leq \max_{k \notin \{i,j\}} ||\boldsymbol{\mu}^{(k)} - \boldsymbol{\mu}^{(i)}||_2 + \frac{\pi_j}{\pi_i} \exp(-\frac{(1-2\xi)\ell^2}{2\tilde{\sigma}_t^2})||\boldsymbol{\mu}^{(j)} - \boldsymbol{\mu}^{(i)}||_2 + ||\sum_{m \notin \{i,j\}} \tilde{\gamma}_m(\boldsymbol{y}_t)\boldsymbol{\mu}^{(m)} - \boldsymbol{\mu}^{(i)}||_2, \text{ Eq. (H.131)} \tag{H.183}$$

$$\leq \max_{k \notin \{i,j\}} ||\boldsymbol{\mu}^{(k)} - \boldsymbol{\mu}^{(i)}||_2 + \frac{\pi_j}{\pi_i} \ell \exp(-\kappa) + ||\sum_{m \notin \{i,j\}} \tilde{\gamma}_m(\boldsymbol{y}_t)\boldsymbol{\mu}^{(m)} - \boldsymbol{\mu}^{(i)}||_2, \ \xi_t \in (-a,a), 1 - 2a \geq \frac{1}{2} \tag{H.184}$$

$$\leq \max_{k \notin \{i,j\}} ||\boldsymbol{\mu}^{(k)} - \boldsymbol{\mu}^{(i)}||_2 + \frac{\pi_j}{\pi_i} \ell \exp(-\kappa) + \max_{m \notin \{i,j\}} ||\boldsymbol{\mu}^{(m)} - \boldsymbol{\mu}^{(i)}||_2 (N-2) \exp(-\kappa), \text{ Eq. (G.72)}. \tag{H.185}$$

Incorporating Eq. (H.185) into Eq. (H.178) yields:

$$|\frac{d}{d\xi_t} \tilde{\gamma}_k(\boldsymbol{y}_t)| \leq |\sup_{\xi_t \in (-a,a)} \frac{d}{d\xi_t} \tilde{\gamma}_k(\boldsymbol{y}_t)| \tag{H.186}$$

$$\leq |\sup_{\xi_t \in (-a,a)} \frac{\ell}{\tilde{\sigma}_t^2} \frac{\pi_k}{\pi_{d^*}} (\max_{k \notin \{i,j\}} ||\boldsymbol{\mu}^{(k)} - \boldsymbol{\mu}^{(i)}||_2)| \tag{H.187}$$

$$\leq \frac{\ell}{\tilde{\sigma}_t^2} \frac{\pi_k}{\pi_{d^*}} \exp(-\kappa)(\max_{k \notin \{i,j\}} ||\boldsymbol{\mu}^{(k)} - \boldsymbol{\mu}^{(i)}||_2 + \frac{\pi_j}{\pi_i} \ell \exp(-\kappa) \tag{H.188}$$

$$+ \max_{m \notin \{i,j\}} ||\boldsymbol{\mu}^{(m)} - \boldsymbol{\mu}^{(i)}||_2 (N-2) \exp(-\kappa)) \tag{H.189}$$

Then, incorporating Eq. (H.189) into Eq. (H.174) yields:

$$|\frac{d}{d\xi_t} \delta_{\xi_t}| \leq \frac{\max_{k \notin \{i,j\}} ||\boldsymbol{\mu}^{(k)} - \boldsymbol{\mu}^{(i)}||_2}{\ell} (N-2) \max_{k \notin \{i,j\}} |\frac{d}{d\xi_t} \tilde{\gamma}_k(\boldsymbol{y}_t)| \tag{H.190}$$

$$\leq \frac{\max_{k \notin \{i,j\}} ||\boldsymbol{\mu}^{(k)} - \boldsymbol{\mu}^{(i)}||_2}{\ell} (N-2) \frac{\ell}{\tilde{\sigma}_t^2} \max_{k \notin \{i,j\}} \frac{\pi_k}{\pi_{d^*}} \exp(-\kappa)(\max_{k \notin \{i,j\}} ||\boldsymbol{\mu}^{(k)} - \boldsymbol{\mu}^{(i)}||_2 + \frac{\pi_j}{\pi_i} \ell \exp(-\kappa) \tag{H.191}$$

$$+ \max_{m \notin \{i,j\}} ||\boldsymbol{\mu}^{(m)} - \boldsymbol{\mu}^{(i)}||_2 (N-2) \exp(-\kappa)) \tag{H.192}$$

Incorporating Eq. (H.170) and Eq. (H.192) into Eq. (H.152) yields:

$$\frac{d}{d\xi_t}F_t(\xi_t^{*,i}) \le |\frac{d}{d\xi_t}F_t(\xi_t^{*,i})| \tag{H.193}$$

$$= \left|\left(\frac{d}{d\xi_t}\tilde{\gamma}_j(\boldsymbol{y}_t) + \frac{d}{d\xi_t}\delta_{\xi_t} - 1\right)|_{\xi_t=\xi_t^{*,i}}\right| \tag{H.194}$$

$$\le \sup_{\xi_t \in (-a,a)} \left|\left(\frac{d}{d\xi_t}\tilde{\gamma}_j(\boldsymbol{y}_t) + \frac{d}{d\xi_t}\delta_{\xi_t} - 1\right)\right| \tag{H.195}$$

$$\le \sup_{\xi_t \in (-a,a)} |\frac{d}{d\xi_t}\tilde{\gamma}_j(\boldsymbol{y}_t)| + \sup_{\xi_t \in (-a,a)} |\frac{d}{d\xi_t}\delta_{\xi_t}| - 1 \tag{H.196}$$

$$\le 8\frac{\pi_j}{\pi_i}\kappa\exp(-\kappa) + \max_{k\notin\{i,j\}} 2||\boldsymbol{\mu}^{(k)} - \boldsymbol{\mu}^{(i)}||_2\kappa\exp(-\kappa)(N-2)\max_{k\notin\{i,j\}}\frac{\pi_k}{\pi_{d^*}}\exp(-\kappa) \tag{H.197}$$

$$+ \frac{\max_{k\notin\{i,j\}}||\boldsymbol{\mu}^{(k)} - \boldsymbol{\mu}^{(i)}||_2}{\ell}(N-2)\frac{\ell}{\sigma^2}\max_{k\notin\{i,j\}}\frac{\pi_k}{\pi_{d^*}}\exp(-\kappa)(\max_{k\notin\{i,j\}}||\boldsymbol{\mu}^{(k)} - \boldsymbol{\mu}^{(i)}||_2 + \frac{\pi_j}{\pi_i}\ell\exp(-\kappa) \tag{H.198}$$

$$+ \max_{m\notin\{i,j\}}||\boldsymbol{\mu}^{(m)} - \boldsymbol{\mu}^{(i)}||_2(N-2)\exp(-\kappa)) - 1 \tag{H.199}$$

since $\frac{1}{\tilde{\sigma}_t^2} \le \frac{1}{\min_{t\in[t,\hat{\tau}]}\tilde{\sigma}_t^2} = \frac{1}{\sigma^2}$.

Define $W_{i,j} := \frac{\pi_j}{\pi_i}$, $M := \max_{k\notin\{i,j\}}||\boldsymbol{\mu}^{(k)} - \boldsymbol{\mu}^{(i)}||_2$, and $R := \max_{k\notin\{i,j\}} W_{k,d^*}$. We then must choose $\kappa$ s.t.

$$8W_{i,j}\kappa\exp(-\kappa) + 2M\kappa\exp(-\kappa)(N-2)R\exp(-\kappa) \tag{H.200}$$

$$+ \frac{M(N-2)R}{\sigma^2}(M\exp(-\kappa) + W_{i,j}\ell\exp(-2\kappa) + M(N-2)\exp(-2\kappa)) < 1 \tag{H.201}$$

Denote:

$$T_1(\kappa) = 8W_{i,j}\kappa\exp(-\kappa), \tag{H.202}$$

$$T_2(\kappa) = 2M\kappa\exp(-\kappa)(N-2)R\exp(-\kappa), \tag{H.203}$$

and,

$$T_3(\kappa) = \frac{M(N-2)R}{\sigma^2}(M\exp(-\kappa) + W_{i,j}\ell\exp(-2\kappa) + M(N-2)\exp(-2\kappa)) \tag{H.204}$$

We first bound $T_1$:

$$T_1(\kappa) = 8W_{i,j}\kappa\exp(-\kappa) \tag{H.205}$$

$$\le 8W_{i,j}\exp(-\kappa/2) < \frac{1}{3} \tag{H.206}$$

$$\iff \exp(-\kappa/2) < \frac{1}{24W_{i,j}} \tag{H.207}$$

$$\iff \kappa > 2\log(24W_{i,j}) \tag{H.208}$$

$$\in 2\log N + \Theta(1) \tag{H.209}$$

Next, we bound $T_2$. Since $\kappa > 1$, we have that $\kappa \exp(-\kappa) \le \exp(-1)$, hence $\kappa \exp(-2\kappa) = \exp(-\kappa)(\kappa \exp(-\kappa)) \le \exp(-1) \exp(-\kappa)$. Thus:

$$T_2(\kappa) = 2M\kappa \exp(-2\kappa)(N-2)R \tag{H.210}$$

$$\le 2M \exp(-1) \exp(-\kappa)(N-2)R < \frac{1}{3} \tag{H.211}$$

$$\iff \exp(-\kappa) < \frac{\exp(1)}{6MR(N-2)} \tag{H.212}$$

$$\iff \kappa > \log(6 \exp(-1)MR(N-2)) \tag{H.213}$$

$$\in \log N + \Theta(1) \tag{H.214}$$

Finally, we bound $T_3$. Since the inverse exponential is monotonically decreasing, $\exp(-2\kappa) \le \exp(-\kappa)$. Hence, $M \exp(-\kappa) + W_{i,j}\ell(\exp(-2\kappa) + M(n_2) \exp(-2\kappa) \le (M + W_{i,j}\ell + M(N-2)) \exp(-\kappa)$. Define the constant:

$$A := \frac{M(N-2)R}{\sigma^2}(M + W_{i,j}\ell + M(N-2)) \tag{H.215}$$

Then:

$$T_3(\kappa) = \frac{M(N-2)R}{\sigma^2}(M \exp(-\kappa) + W_{i,j}\ell \exp(-2\kappa) + M(N-2) \exp(-2\kappa)) \tag{H.216}$$

$$\le A \exp(-\kappa) < \frac{1}{3} \tag{H.217}$$

$$\iff \exp(-\kappa) < \frac{1}{3A} \tag{H.218}$$

$$\iff \kappa > \log(3A) \tag{H.219}$$

$$\iff \kappa > \log(3\frac{M(N-2)R}{\sigma^2}(M + W_{i,j}\ell + M(N-2))) \tag{H.220}$$

$$\in 2 \log N + \Theta(1) \tag{H.221}$$

Furthermore, $\kappa \ge \log N + \log(4C_a) \in 2 \log N + \Theta(1)$. Thus, the maximum over all these constraints is also in $2 \log N + \Theta(1)$, as desired. Finally, note that $t \in [0, \hat{\tau})$ was arbitrary, and the bound of the derivative of $F_t(\xi_t)$ evaluated at the equilibria does not depend on time, so $\xi_t^{*,i}$ is uniformly instantaneously stable for all $t \in [0, \hat{\tau})$, as desired. $\qquad\square$

### H.4. Proof of Corollary 4.6

*Proof.* Fix $\hat{\tau} = \min\{\tau, \tau_2\}$ as in Prop. 4.7. By Prop. 4.5, there exists an instantaneously stable equilibrium $\xi_t^{*,i}$ of Eq. (G.50) within a $\varepsilon$-ball of mode $\mu^{(i)}$, for all $t < \hat{\tau}$. Moreover, the proof of Prop. 4.5 shows that the stability is *uniform* over $t \in [0, \hat{\tau}]$, in the sense that there exists a constant $m_0 > 0$ (independent of $t$) such that

$$F_t'(\xi_t^{*,i}) \le -m_0 \qquad \forall t \in [0, \hat{\tau}]. \tag{H.222}$$

Since $\tilde{\gamma}_j$ is a smooth composition of Gaussian p.d.f.s and $\Delta_{\xi_t}$ is differentiable in $\xi$ (as used throughout the Prop. 4.5 proof), we have that $F_t$ is $C^1$ in $\xi$ for each fixed $t$. Since $F_t(\xi_t^{*,i}) = 0$ and $\partial_\xi F_t(\xi_t^{*,i}) \le -m_0 < 0$, the implicit function theorem implies that the equilibria can be chosen as a continuous branch $t \mapsto \xi_t^{*,i}$ on $[0, \hat{\tau}]$.

Hence, by continuity of $F_t'$ in $\xi$ for each fixed $t \in [0, \hat{\tau}]$ there exists a radius $\varkappa_t > 0$ such that

$$F_t'(\xi) \le -\frac{m_0}{2} \qquad \forall \xi \in [\xi_t^{*,i} - \varkappa_t, \ \xi_t^{*,i} + \varkappa_t]. \tag{H.223}$$

By joint continuity in $(t, \xi)$ and compactness of $[0, \hat{\tau}]$, the radii may be chosen so that $\varkappa := \inf_{t \in [0, \hat{\tau}]} \varkappa_t > 0$. Recall from the proof of Prop. 4.5 that, for $a := 2(\varepsilon_{\delta_\xi} + \varepsilon_{\tilde{\gamma}})$, we have $F_t(-a) > 0, F_t(a) < 0, |\xi_t^{*,i}| \le a, \forall t \in [0, \hat{\tau}]$. Since

$a \in O(N \exp(-\kappa))$, there exists $C > 0$ such that $a \leq CN \exp(-\kappa)$. Thus, to ensure $2a \leq \varkappa$, it suffices to take $\kappa \geq \log\left(\frac{2CN}{\varkappa}\right)$ ,, which is satisfied by the condition $\kappa \geq 2\log N + \Theta(1)$. Therefore, $[-a, a] \subseteq [\xi_t^{*,i} - \varkappa_t, \xi_t^{*,i} + \varkappa_t]$, $\forall t \in [0, \hat{\tau}]$.

Let $\xi_t^{(1)}$ and $\xi_t^{(2)}$ be two solutions of Eq. (G.50) with $\xi_{t_0}^{(1)}, \xi_{t_0}^{(2)} \in [-a, a]$. Since $F_t(-a) > 0$ and $F_t(a) < 0$ uniformly in $t$, the interval $[-a, a]$ is invariant. Hence, for every $t < t_0$,

$$\xi_t^{(1)}, \; \xi_t^{(2)} \in [-a, a] \subseteq [\xi_t^{*,i} - \varkappa_t, \; \xi_t^{*,i} + \varkappa_t]. \tag{H.224}$$

Define $\Delta_t := \xi_t^{(1)} - \xi_t^{(2)}$. Then $\Delta_t$ is $C^1$ in $t$, and:

$$\dot{\Delta}_t \;=\; F_t(\xi_t^{(1)}) - F_t(\xi_t^{(2)}). \tag{H.225}$$

Now, fix any $t < t_0 \leq \hat{\tau}$. By the mean value theorem applied to the $C^1$ function $F_t(\cdot)$, there exists $\theta_t$ lying between $\xi_t^{(1)}$ and $\xi_t^{(2)}$ such that:

$$F_t(\xi_t^{(1)}) - F_t(\xi_t^{(2)}) \;=\; F_t'(\theta_t)\,(\xi_t^{(1)} - \xi_t^{(2)}) \;=\; F_t'(\theta_t)\,\Delta_t. \tag{H.226}$$

By Eq. (H.224), we have $\theta_t \in [\xi_t^{*,i} - \varkappa_t, \xi_t^{*,i} + \varkappa_t]$, so Eq. (H.223) yields $F_t'(\theta_t) \leq -m_0/2$. Incorporating this into Eq. (H.225) and Eq. (H.226) yields:

$$\dot{\Delta}_t \;=\; F_t'(\theta_t)\,\Delta_t, \qquad \text{with } F_t'(\theta_t) \leq -\frac{m_0}{2}. \tag{H.227}$$

Next, consider the absolutely continuous function $t \mapsto |\Delta_t|$. For $t$ such that $\Delta_t \neq 0$,

$$\frac{d}{dt}|\Delta_t| \;=\; \text{sign}(\Delta_t)\,\dot{\Delta}_t \;=\; \text{sign}(\Delta_t)\,F_t'(\theta_t)\,\Delta_t \;=\; F_t'(\theta_t)\,|\Delta_t| \;\leq\; -\frac{m_0}{2}\,|\Delta_t|. \tag{H.228}$$

Integrating Eq. (H.228) and applying Grönwall's inequality yields for all $t < t_0$:

$$|\xi_t^{(1)} - \xi_t^{(2)}| \;=\; |\Delta_t| \;\leq\; \exp\left(-\frac{m_0}{2}\big(u(t) - u(t_0)\big)\right)|\Delta_{t_0}| \;=\; \exp\left(-\frac{m_0}{2}\big(u(t) - u(t_0)\big)\right)|\xi_{t_0}^{(1)} - \xi_{t_0}^{(2)}|. \tag{H.229}$$

Taking $t_0 = \hat{\tau}$ and $\xi^{(2)} = \bar{\xi}$, where $\bar{\xi}_{\hat{\tau}} = 0$, gives a solution $\bar{\xi}_t \in [-a, a]$ s.t. for all $t \in [0, \hat{\tau}]$, every trajectory in $[-a, a]$ contracts exponentially fast to $\bar{\xi}_t$ by Eq. (H.229). Note that this follows symmetrically in a neighborhood of $\xi_t^{*,j}$.

$\square$

## H.5. Proof of Proposition 4.7

Before stating the proof, we firstly recall the parallel dynamics from Lemma G.3:

$$\dot{\xi}_t = \tilde{\gamma}_j(\boldsymbol{y}_t) - \xi_t + \delta_{\xi_t}, \tag{H.230}$$

where:

$$|\delta_\xi|, \|\boldsymbol{\delta_w}\|_2 \leq \varepsilon \in \mathcal{O}\left(N \exp(-\kappa)\right), \tag{H.231}$$

for:

$$\boldsymbol{y}_t = \boldsymbol{\mu}^{(i)} + \xi_t \ell \boldsymbol{u} + \boldsymbol{w}_t, \tag{H.232}$$

where $\ell = \|\boldsymbol{\mu}^{(j)} - \boldsymbol{\mu}^{(i)}\|_2$, $\boldsymbol{u} = \frac{\boldsymbol{\mu}^{(j)} - \boldsymbol{\mu}^{(i)}}{\ell}$, and $\boldsymbol{w}_t$ satisfies $\langle \boldsymbol{w}_t, \boldsymbol{u} \rangle = 0$, and we also consider the responsibility of mode $j$:

$$\tilde{\gamma}_j(\boldsymbol{y}_t) = \frac{\pi_j \mathcal{N}(\boldsymbol{y}_t; \boldsymbol{\mu}^{(j)}, \tilde{\sigma}_t^2 \boldsymbol{I})}{\sum_{m \in [N]} \pi_m\, \mathcal{N}(\boldsymbol{y}_t; \boldsymbol{\mu}^{(m)}, \tilde{\sigma}_t^2 \boldsymbol{I})}. \tag{H.233}$$

We simplify the parallel dynamics by dropping the term bounded by $\varepsilon$, and additionally consider only the dynamics with respect to the two-mode responsibility, i.e. we consider:

$$\dot{\xi}_t = \tilde{\gamma}_j^{(i,j)}(\boldsymbol{y}_t) - \xi_t, \tag{H.234}$$

where:

$$\tilde{\gamma}_j^{(i,j)}(\boldsymbol{y}_t) := \frac{\pi_j \mathcal{N}(\boldsymbol{y}_t; \boldsymbol{\mu}^{(j)}, \tilde{\sigma}_t^2 \boldsymbol{I})}{\pi_i \mathcal{N}(\boldsymbol{y}_t; \boldsymbol{\mu}^{(i)}, \tilde{\sigma}_t^2 \boldsymbol{I}) + \pi_j \mathcal{N}(\boldsymbol{y}_t; \boldsymbol{\mu}^{(j)}, \tilde{\sigma}_t^2 \boldsymbol{I})}. \tag{H.235}$$

Throughout the statement and proof, we work in the rescaled reverse-time parameter $u$ defined in Lemma G.2. This parameter increases as the original diffusion time $t$ decreases from $T$ to $0$. For notational simplicity, we relabel $u$ as $t$. Thus, $\xi_{\tau_3}$ denotes the initial value of the approximate parallel dynamics over a remaining reverse-time interval of length $\tau_3$, while $\xi_0$ denotes the terminal value. Then, we prove the result:

*Proof.* **(1)**: Since the instance is within the $\varepsilon$-tube around $L_t^{(i,j)}$, we have that $||\boldsymbol{w}_t|| \leq \varepsilon$, just as we do Prop. 4.5. Thus, by the same token as in the proof of Prop. 4.5, we have that:

$$||\boldsymbol{y}_t - \boldsymbol{\mu}^{(i)}||_2^2 = \xi_t^2 \ell^2 + ||\boldsymbol{w}_t||_2^2, \tag{H.236}$$

and:

$$||\boldsymbol{y}_t - \boldsymbol{\mu}^{(j)}||_2^2 = ||(\xi_t - 1)\ell\boldsymbol{u} + \boldsymbol{w}_t||_2^2 = (1 - \xi_t)^2 \ell^2 + ||\boldsymbol{w}_t||_2^2, \tag{H.237}$$

yielding:

$$||\boldsymbol{y}_t - \boldsymbol{\mu}^{(j)}||_2^2 - ||\boldsymbol{y}_t - \boldsymbol{\mu}^{(i)}||_2^2 = (1 - 2\xi_t)\ell^2. \tag{H.238}$$

From Eq. (H.235), we have that:

$$\frac{\tilde{\gamma}_j^{(i,j)}(\boldsymbol{y}_t)}{1 - \tilde{\gamma}_j^{(i,j)}(\boldsymbol{y}_t)} = \frac{\pi_j \mathcal{N}(\boldsymbol{y}_t; \boldsymbol{\mu}^{(j)}, \tilde{\sigma}_t^2 \boldsymbol{I})}{\pi_i \mathcal{N}(\boldsymbol{y}_t; \boldsymbol{\mu}^{(i)}, \tilde{\sigma}_t^2 \boldsymbol{I})} \tag{H.239}$$

$$= \frac{\pi_j}{\pi_i} \exp(-\frac{||\boldsymbol{y}_t - \boldsymbol{\mu}^{(j)}||_2^2 - ||\boldsymbol{y}_t - \boldsymbol{\mu}^{(i)}||_2^2}{2\tilde{\sigma}_t^2}) \tag{H.240}$$

$$= \frac{\pi_j}{\pi_i} \exp(\frac{\ell^2}{\tilde{\sigma}_t^2}(\xi - \frac{1}{2})), \text{ Eq. (H.238)} \tag{H.241}$$

Since $\pi_i = \pi_j$, at $\xi = \frac{1}{2}$, Eq. (H.241) evaluates to:

$$\frac{\tilde{\gamma}_j^{(i,j)}(\boldsymbol{y}_t)}{1 - \tilde{\gamma}_j^{(i,j)}(\boldsymbol{y}_t)}\Big|_{\xi_t = 1/2} = 1 \tag{H.242}$$

yielding:

$$\tilde{\gamma}_j^{(i,j)}(\boldsymbol{y}_t)_{\xi_t = 1/2} = 1/2 \tag{H.243}$$

Define $F_t(\xi_t) = \tilde{\gamma}_j^{(i,j)}(\boldsymbol{y}_t) - \xi_t$. Then, $F_t(1/2) = 0$, so $\xi^* = \frac{1}{2}$ is an (instantaneous) equilibrium of Eq. (H.234).

The corresponding point in $\mathbb{R}^d$ is thus:

$$\boldsymbol{y}_t^* := \boldsymbol{y}_t\big|_{\xi=1/2} \tag{H.244}$$

$$= \boldsymbol{\mu}^{(i)} + \frac{1}{2}\ell\boldsymbol{u} + \boldsymbol{w}_t \tag{H.245}$$

$$= \frac{\boldsymbol{\mu}^{(i)} + \boldsymbol{\mu}^{(j)}}{2} + \boldsymbol{w}_t. \tag{H.246}$$

Next, we demonstrate that this is a hyperbolic unstable (instantaneous) equilibria. We have that:

$$\frac{d}{d\xi_t} F_t(\xi_t) = \frac{d}{d\xi_t}\tilde{\gamma}_j^{(i,j)}(\boldsymbol{y}_t) - 1 \tag{H.247}$$

$$= \frac{\ell^2}{\tilde{\sigma}_t^2}\tilde{\gamma}_j^{(i,j)}(\boldsymbol{y}_t)(1 - \tilde{\gamma}_j^{(i,j)}(\boldsymbol{y}_t)) - 1, \tag{H.248}$$

We thus have that:

$$\frac{d}{d\xi_t} F_t(\xi_t)\big|_{\xi_t=1/2} = \frac{\ell^2}{\tilde{\sigma}_t^2}\frac{1}{2}\frac{1}{2} - 1 = \frac{\ell^2}{4\tilde{\sigma}_t^2} - 1. \tag{H.249}$$

By Asm. 4.4, we thus have that:

$$\lambda_t := \frac{\ell^2}{4\tilde{\sigma}_t^2} - 1 \geq \kappa - 1 > 0 \tag{H.250}$$

since $\kappa > 1$. Since $\lambda_t$ has no complex components, $\xi_t = 1/2$ thus corresponds to a hyperbolic unstable (instantaneous) equilibria.

**(2)**: Consider $t \in [0, \tau_3]$ and recall the approximate parallel dynamics in Eq. (H.234). Note that $\underline{\lambda}$ and $\overline{\lambda}$ exist by the extreme value theorem, since they are defined on a compact domain with respect to continuous $\lambda_t$ (since $\tilde{\sigma}_t^2$ is continuous). Within $\text{Tube}_{\varepsilon,t}^{(i,j)}$, by Eq. (H.241), we have the identity:

$$\frac{\tilde{\gamma}_j^{(i,j)}(\xi_t)}{1 - \tilde{\gamma}_j^{(i,j)}(\xi_t)} = \frac{\pi_j}{\pi_i}\exp\left(\frac{\ell^2}{\tilde{\sigma}_t^2}\left(\xi_t - \frac{1}{2}\right)\right). \tag{H.251}$$

Noting that $\pi_i = \pi_j$, denoting $a_t := \frac{\ell^2}{\tilde{\sigma}_t^2}$ and solving the log-odds equation obtains the logistic form:

$$\tilde{\gamma}_j^{(i,j)} = \frac{1}{1 + \exp(-a_t(\xi_t - \frac{1}{2}))} =: \varsigma\left(a_t\left(\xi_t - \frac{1}{2}\right)\right), \tag{H.252}$$

where $\varsigma(z) = \frac{1}{1+e^{-z}}$ is the sigmoid function. Since $\varsigma'(z) = \varsigma(z)(1 - \varsigma(z))$ and $\varsigma''(z) = \varsigma'(z)(1 - 2\varsigma(z))$, and since $0 \leq \varsigma \leq 1$ implies $0 \leq \varsigma' \leq \frac{1}{4}$ and $|1 - 2\varsigma| \leq 1$, we have $|\varsigma''(z)| \leq \frac{1}{4}$. Hence,

$$|F_t''(\xi_t)| = \big|\tilde{\gamma}_j^{(i,j)\prime\prime}(\xi_t)\big| \leq \frac{a_t^2}{4}. \tag{H.253}$$

Next, for an $a$ to be chosen, consider $0 < |a| \leq \vartheta$. Since $F_t \in C^2$, Taylor's theorem with Lagrange remainder at $\xi = \frac{1}{2}$ gives:

$$F_t\left(\frac{1}{2} + a\right) = F_t\left(\frac{1}{2}\right) + F_t'\left(\frac{1}{2}\right)a + \frac{1}{2}F_t''\left(\frac{1}{2} + ra\right)a^2 \tag{H.254}$$

for some $r \in (0, 1)$. Using $F_t(\frac{1}{2}) = 0$ and $F_t'(\frac{1}{2}) = \lambda_t$, we obtain

$$F_t\left(\frac{1}{2} + a\right) = \lambda_t a + \frac{1}{2} F_t''\left(\frac{1}{2} + ra\right) a^2. \tag{H.255}$$

By Eq. (H.253),

$$\left| F_t\left(\frac{1}{2} + a\right) - \lambda_t a \right| \le \frac{1}{2} \cdot \frac{a_t^2}{4} |a|^2 = \frac{a_t^2}{8} |a|^2. \tag{H.256}$$

Dividing by $|a|$ yields

$$\left| \frac{F_t(\frac{1}{2} + a)}{a} - \lambda_t \right| \le \frac{a_t^2}{8} |a| \le \frac{a_t^2}{8} \vartheta, \tag{H.257}$$

and thus

$$\lambda_t - \frac{a_t^2}{8} \vartheta \le \frac{F_t(\frac{1}{2} + a)}{a} \le \lambda_t + \frac{a_t^2}{8} \vartheta. \tag{H.258}$$

Since $\lambda_t = \frac{\ell^2}{4\bar{\sigma}_t^2} - 1$, we have $a_t = \frac{\ell^2}{\bar{\sigma}_t^2} = 4(1 + \lambda_t)$, hence $\frac{a_t^2}{8} = 2(1 + \lambda_t)^2$. Plugging into Eq. (H.258) gives

$$\frac{F_t(\frac{1}{2} + a)}{a} \le \lambda_t + 2(1 + \lambda_t)^2 \vartheta \le \bar{\lambda} + 2(1 + \bar{\lambda})^2 \vartheta =: \Lambda_+, \tag{H.259}$$

and similarly

$$\frac{F_t(\frac{1}{2} + a)}{a} \ge \lambda_t - 2(1 + \lambda_t)^2 \vartheta \ge \underline{\lambda} - 2(1 + \bar{\lambda})^2 \vartheta. \tag{H.260}$$

By our choice of radius $\vartheta$ in the statement of Prop. 4.7, we have $\underline{\lambda} - 2(1 + \bar{\lambda})^2 \vartheta > 0$, hence $\frac{F_t(\frac{1}{2} + a)}{a} > 0$, so

$$\mathrm{sign}\left(F_t\left(\frac{1}{2} + a\right)\right) = \mathrm{sign}(a) \qquad \text{whenever } 0 < |a| \le \vartheta,$$

uniformly for all $t \in [0, \tau_3]$.

Next, letting $t$ now be the original diffusion time, let $s := u(t) - u(\tau_3)$, so that $t(s) = u^{-1}(u(\tau_3) + s)$. Define

$$\zeta_s := \xi_{t(s)}, \qquad \widetilde{F}_s := F_{t(s)}, \qquad A_s := \zeta_s - \frac{1}{2}. \tag{H.261}$$

This yields that

$$\zeta_0 = \xi_{\tau_3}, \qquad \zeta_{u(0) - u(\tau_3)} = \xi_0, \qquad \dot{\zeta}_s = \widetilde{F}_s(\zeta_s). \tag{H.262}$$

As discussed at the beginning of Sec. H.5, we relabel $\tau_3$ as this remaining interval $u(0) - u(\tau_3)$, yielding $\zeta_0 = \xi_{\tau_3}$ and $\zeta_{\tau_3} = \xi_0$.

Now, for $0 < |A_s| \le \vartheta$, by the preceding sign condition,

$$\frac{d}{ds} |A_s| = \mathrm{sign}(A_s) \dot{A}_s \tag{H.263}$$

$$= \mathrm{sign}(A_s) \widetilde{F}_s\left(\frac{1}{2} + A_s\right) \tag{H.264}$$

$$= \left| \widetilde{F}_s\left(\frac{1}{2} + A_s\right) \right| \tag{H.265}$$

$$\le \Lambda_+ |A_s|. \tag{H.266}$$

Next, suppose that

$$0 < |A_0| = \left| \xi_{\tau_3} - \frac{1}{2} \right| < \vartheta \exp(-\Lambda_+ \tau_3). \tag{H.267}$$

Let $\varrho := \inf\{s \in [0, \tau_3] : |A_s| \geq \vartheta\}$ with $\inf \emptyset = +\infty$. Suppose $\varrho \leq \tau_3$. Then $|A_s| < \vartheta$ $\forall s \in [0, \varrho)$ and $|A_\varrho| = \vartheta$. By Grönwall's inequality, for $s \in [0, \varrho)$,

$$|A_s| \leq |A_0| \exp(\Lambda_+ s). \tag{H.268}$$

Taking $s \to \varrho$,

$$|A_\varrho| \leq |A_0| \exp(\Lambda_+ \varrho) \tag{H.269}$$

$$\leq |A_0| \exp(\Lambda_+ \tau_3) \tag{H.270}$$

$$= \left|\xi_{\tau_3} - \frac{1}{2}\right| \exp(\Lambda_+ \tau_3) \tag{H.271}$$

$$< \vartheta, \tag{H.272}$$

contradicting $|A_\varrho| = \vartheta$. Hence $\varrho = +\infty$, so $|A_s| < \vartheta$ $\forall s \in [0, \tau_3]$. Equivalently,

$$\left|\xi_t - \frac{1}{2}\right| < \vartheta \qquad \forall t \in [0, \tau_3]. \tag{H.273}$$

By contrapositive, if exit occurs by terminal time $t = 0$, then

$$\left|\xi_{\tau_3} - \frac{1}{2}\right| \geq \vartheta \exp(-\Lambda_+ \tau_3), \tag{H.274}$$

so a necessary condition for exit is

$$\tau_3 \geq \frac{1}{\Lambda_+} \log\left(\frac{\vartheta}{|\xi_{\tau_3} - \frac{1}{2}|}\right). \tag{H.275}$$

$\square$

## H.6. Proof of Proposition 5.1

*Proof.* Set $T := \tau_3$. Under the time relabeling $\theta := T - t$, the proposition's event $\{|A_0| \leq \vartheta\}$ corresponds to $\{|A_T| \leq \vartheta\}$; $A_T = 0$ in Eq. (4) corresponds to $A_0 = 0$; and the assumption with $\lambda_{\text{rep}}$ is negated in the resulting forward-time process, which we analyze below.

Define the continuous martingale

$$M_t := \int_0^t \sqrt{2\eta(s)}\, dB_s, \qquad t \in [0, T]. \tag{H.276}$$

Integrating Eq. (4) yields the decomposition

$$A_t = \int_0^t b(A_s, s)\, ds + M_t, \qquad t \in [0, T]. \tag{H.277}$$

We first bound the confinement event

$$E_{2\vartheta} := \left\{\sup_{0 \leq t \leq T} |A_t| \leq 2\vartheta\right\}. \tag{H.278}$$

On $E_{2\vartheta}$ we have $|A_s| \leq 2\vartheta$ for all $s \in [0, T]$, hence by the local drift bound,

$$|b(A_s, s)| \leq K(s)\, |A_s| \leq 2K(s)\vartheta, \qquad s \in [0, T]. \tag{H.279}$$

Therefore, for each $t \in [0, T]$ and on $E_{2\vartheta}$,

$$\left|\int_0^t b(A_s, s)\, ds\right| \leq \int_0^t |b(A_s, s)|\, ds \leq 2\vartheta \int_0^T K(s)\, ds. \tag{H.280}$$

Combining Eqs. (H.277) and (H.280), for all $t \in [0, T]$ and on $E_{2\vartheta}$,

$$|M_t| = \left|A_t - \int_0^t b(A_s, s)\, ds\right| \leq |A_t| + \left|\int_0^t b(A_s, s)\, ds\right| \leq 2\vartheta\left(1 + \int_0^T K(s)\, ds\right). \tag{H.281}$$

Hence,

$$E_{2\vartheta} \subseteq \left\{ \sup_{0 \leq t \leq T} |M_t| \leq 2\left(1 + \int_0^T K(s)\,ds\right)\vartheta \right\}. \tag{H.282}$$

We next identify the quadratic variation of $M$. Fix $t \in [0, T]$ and let $\pi = \{0 = t_0 < t_1 < \cdots < t_n = t\}$ be a partition of $[0, t]$ with mesh

$$|\pi| := \max_{0 \leq k \leq n-1} (t_{k+1} - t_k). \tag{H.283}$$

Write increments

$$\Delta M_k := M_{t_{k+1}} - M_{t_k} = \int_{t_k}^{t_{k+1}} \sqrt{2\eta(s)}\,dB_s. \tag{H.284}$$

Define

$$Q(\pi) := \sum_{k=0}^{n-1} (\Delta M_k)^2. \tag{H.285}$$

By Itô isometry applied to Eq. (H.284),

$$\mathbb{E}\big[(\Delta M_k)^2\big] = \int_{t_k}^{t_{k+1}} 2\eta(s)\,ds =: v_k. \tag{H.286}$$

Since $\Delta M_k$ is Gaussian with mean 0 and variance $v_k$,

$$\mathbb{E}\big[(\Delta M_k)^4\big] = 3v_k^2, \qquad \mathrm{Var}\big((\Delta M_k)^2\big) = 2v_k^2. \tag{H.287}$$

Using independence of Brownian increments,

$$\mathbb{E}[Q(\pi)] = \sum_{k=0}^{n-1} v_k = \int_0^t 2\eta(s)\,ds, \tag{H.288}$$

and

$$\mathrm{Var}(Q(\pi)) = 2\sum_{k=0}^{n-1} v_k^2 \leq 2\left(\max_{0 \leq k \leq n-1} v_k\right)\sum_{k=0}^{n-1} v_k. \tag{H.289}$$

Since $\eta \in L^1([0, T])$, absolute continuity of the integral implies $\max_k v_k \to 0$ as $|\pi| \to 0$. Hence,

$$\mathbb{E}\left[\left(Q(\pi) - \int_0^t 2\eta(s)\,ds\right)^2\right] = \mathrm{Var}(Q(\pi)) \to 0 \qquad \text{as } |\pi| \to 0. \tag{H.290}$$

Therefore $Q(\pi) \to \int_0^t 2\eta(s)\,ds$ in probability. In particular, the quadratic variation

$$[M]_t := \lim_{|\pi| \to 0} Q(\pi) \tag{H.291}$$

satisfies

$$[M]_t = \int_0^t 2\eta(s)\,ds, \qquad t \in [0, T]. \tag{H.292}$$

Next, by Dambis–Dubins–Schwarz (Dubins & Schwarz, 1965), there exists a standard Brownian motion $(W_r)_{r \geq 0}$ such that

$$M_t = W_{[M]_t}, \qquad t \in [0, T]. \tag{H.293}$$

Since $[M]_t$ is nondecreasing,

$$\sup_{0 \leq t \leq T} |M_t| = \sup_{0 \leq r \leq [M]_T} |W_r|, \qquad [M]_T = 2\int_0^T \eta(s)\,ds. \tag{H.294}$$

Combining Eqs. (H.282) and (H.294) gives

$$\mathbb{P}(E_{2\vartheta}) \leq \mathbb{P}\Big(\sup_{0 \leq r \leq [M]_T} |W_r| \leq 2\Big(1 + \int_0^T K(s)\,ds\Big)\vartheta\Big). \tag{H.295}$$

Applying Lemma G.6 with

$$V = [M]_T = 2\int_0^T \eta(s)\,ds, \qquad a = 2\Big(1 + \int_0^T K(s)\,ds\Big)\vartheta, \tag{H.296}$$

yields the confinement bound used in Eq. (7).

We now bound the terminal event. Let $(\mathcal{F}_t)_{t \in [0,T]}$ be the natural filtration of $(A_t)$. Define

$$H_\vartheta := \{|A_T| \leq \vartheta\}.$$

Then

$$\mathbb{P}(H_\vartheta) = \mathbb{P}(H_\vartheta \cap E_{2\vartheta}) + \mathbb{P}(H_\vartheta \cap E_{2\vartheta}^c) \leq \mathbb{P}(E_{2\vartheta}) + \mathbb{P}(H_\vartheta \cap E_{2\vartheta}^c). \tag{H.297}$$

Next, let

$$\tau := \inf\{t \in [0,T] : |A_t| \geq 2\vartheta\}. \tag{H.298}$$

By convention, $\tau = +\infty$ if no such time exists. By continuity,

$$E_{2\vartheta} = \{\tau > T\}, \qquad E_{2\vartheta}^c = \{\tau \leq T\}.$$

Hence

$$\mathbb{P}(H_\vartheta \cap E_{2\vartheta}^c) = \mathbb{P}(|A_T| \leq \vartheta,\ \tau \leq T). \tag{H.299}$$

Using conditional expectation at $\tau$ and the strong Markov property,

$$\begin{aligned}
\mathbb{P}(|A_T| \leq \vartheta,\ \tau \leq T) &= \mathbb{E}\Big[\mathbf{1}_{\{\tau \leq T\}}\, \mathbb{P}_{\tau, A_\tau}(|A_T| \leq \vartheta)\Big] \\
&\leq \sup_{s \in [0,T]}\ \sup_{a \in \{\pm 2\vartheta\}} \mathbb{P}_{s,a}(|A_T| \leq \vartheta).
\end{aligned} \tag{H.300}$$

Fix $s \in [0,T]$ and consider $A_s = 2\vartheta$. Define the first return time

$$\sigma := \inf\{t \in [s,T] : A_t = \vartheta\}. \tag{H.301}$$

By continuity,

$$\mathbb{P}_{s,2\vartheta}(|A_T| \leq \vartheta) \leq \mathbb{P}_{s,2\vartheta}(\sigma \leq T). \tag{H.302}$$

By Eq. (6), for all $t \in [0,T]$,

$$b(a,t) \geq \lambda_{\mathrm{rep}}\, a \qquad \forall a \geq \vartheta, \tag{H.303}$$

and recall $\eta(t) \leq \eta_{\max}$ on $[0,T]$. For $r > 0$ define $Z_t := e^{-rA_t}$. Applying Itô's formula on $[s, \sigma \wedge T]$ gives

$$dZ_t = Z_t\big(-r\,b(A_t,t) + r^2\eta(t)\big)\,dt - rZ_t\sqrt{2\eta(t)}\,dB_t. \tag{H.304}$$

On $[s,\sigma)$, using (H.303) and $\eta(t) \leq \eta_{\max}$,

$$-r\,b(A_t,t) + r^2\eta(t) \leq -r(\lambda_{\mathrm{rep}}\vartheta) + r^2\eta_{\max}. \tag{H.305}$$

Choose

$$r := \frac{\lambda_{\mathrm{rep}}\vartheta}{2\eta_{\max}}. \tag{H.306}$$

Then

$$-r(\lambda_{\text{rep}}\vartheta) + r^2\eta_{\max} = -\frac{\lambda_{\text{rep}}^2\vartheta^2}{4\eta_{\max}} < 0, \tag{H.307}$$

so $(Z_{t\wedge\sigma})_{t\in[s,T]}$ is a supermartingale. By optional stopping,

$$\mathbb{E}[Z_{\sigma\wedge T}] \leq Z_s = e^{-r\cdot 2\vartheta}. \tag{H.308}$$

On $\{\sigma \leq T\}$ we have $Z_{\sigma\wedge T} = e^{-r\vartheta}$, hence

$$\mathbb{P}_{s,2\vartheta}(\sigma \leq T) \leq e^{-r\vartheta} = \exp\left(-\frac{\lambda_{\text{rep}}\vartheta^2}{2\eta_{\max}}\right). \tag{H.309}$$

An identical argument with $\widetilde{Z}_t := e^{rA_t}$ yields the same bound starting from $A_s = -2\vartheta$. Therefore,

$$\sup_{s\in[0,T]} \sup_{a\in\{\pm 2\vartheta\}} \mathbb{P}_{s,a}(|A_T| \leq \vartheta) \leq 2\exp\left(-\frac{\lambda_{\text{rep}}\vartheta^2}{2\eta_{\max}}\right). \tag{H.310}$$

Plugging this into (H.300) and then into (H.297) yields

$$\mathbb{P}(H_\vartheta) \leq \mathbb{P}(E_{2\vartheta}) + 2\exp\left(-\frac{\lambda_{\text{rep}}\vartheta^2}{2\eta_{\max}}\right). \tag{H.311}$$

Finally, substitute the bound on $\mathbb{P}(E_{2\vartheta})$ from Eq. (H.295) (via Lemma G.6) into Eq. (H.311) to obtain Eq. (7). $\qquad\square$

### H.7. Proof of Proposition B.1

*Proof.* **Unequal Weights**

We consider Eq. (H.234) and $t \leq \hat{\tau}$, where $\hat{\tau}$ is defined in Asm. 4.4. By Eq. (H.241), we have that:

$$\frac{\tilde{\gamma}_{j,t}^{(ij)}(\xi)}{1 - \tilde{\gamma}_{j,t}^{(ij)}(\xi)} = \frac{\pi_j}{\pi_i}\exp\left(-\frac{\|\boldsymbol{y}(\xi) - \boldsymbol{\mu}^{(j)}\|_2^2 - \|\boldsymbol{y}(\xi) - \boldsymbol{\mu}^{(i)}\|_2^2}{2\tilde{\sigma}_t^2}\right)$$
$$= \frac{\pi_j}{\pi_i}\exp\left(\frac{\ell^2}{\tilde{\sigma}_t^2}\left(\xi - \frac{1}{2}\right)\right), \tag{H.312}$$

Define the pairwise parallel drift

$$F_{ij,t}(\xi) := \tilde{\gamma}_{j,t}^{(ij)}(\xi) - \xi. \tag{H.313}$$

Let $\xi_{ij}^\star(t) \in (0,1)$ be an interior equilibrium, i.e. $F_{ij,t}(\xi_{ij}^\star(t)) = 0$, so $\tilde{\gamma}_{j,t}^{(ij)}(\xi_{ij}^\star(t)) = \xi_{ij}^\star(t)$. Taking log-odds in Eq. (H.312) and substituting $\tilde{\gamma}_{j,t}^{(ij)}(\xi_{ij}^\star(t)) = \xi_{ij}^\star(t)$ yields

$$\log\left(\frac{\xi_{ij}^\star(t)}{1 - \xi_{ij}^\star(t)}\right) = \log\left(\frac{\pi_j}{\pi_i}\right) + \frac{\ell^2}{\tilde{\sigma}_t^2}\left(\xi_{ij}^\star(t) - \frac{1}{2}\right), \tag{H.314}$$

which is Eq. (B.8). In particular, if $\pi_i = \pi_j$ then $\xi_{ij}^\star(t) = \frac{1}{2}$. If $\pi_j < \pi_i$, then $\log(\pi_j/\pi_i) < 0$ and Eq. (H.314) implies $\xi_{ij}^\star(t) > \frac{1}{2}$ (and similarly $\pi_j > \pi_i$ implies $\xi_{ij}^\star(t) < \frac{1}{2}$).

Next, differentiating Eq. (H.312) yields:

$$\tilde{\gamma}_{j,t}^{(ij)\,\prime}(\xi) = \frac{\ell^2}{\tilde{\sigma}_t^2}\tilde{\gamma}_{j,t}^{(ij)}(\xi)\left(1 - \tilde{\gamma}_{j,t}^{(ij)}(\xi)\right), \tag{H.315}$$

so

$$F_{ij,t}'(\xi) = \tilde{\gamma}_{j,t}^{(ij)\,\prime}(\xi) - 1. \tag{H.316}$$

Evaluating at $\xi_{ij}^\star(t)$ (where $\tilde{\gamma}_{j,t}^{(ij)}(\xi_{ij}^\star(t)) = \xi_{ij}^\star(t)$) yields

$$F_{ij,t}'(\xi_{ij}^\star(t)) = \frac{\ell^2}{\tilde{\sigma}_t^2} \, \xi_{ij}^\star(t)\big(1 - \xi_{ij}^\star(t)\big) - 1, \qquad\qquad\qquad \text{(H.317)}$$

which is Eq. (B.9). Thus $\xi_{ij}^\star(t)$ is hyperbolic unstable whenever $F_{ij,t}'(\xi_{ij}^\star(t)) > 0$.

**Exact Parallel Dynamics** By Lemma G.3, the exact parallel dynamics in Eq. (G.50) differs from Eq. (H.234) by an error term $\delta_{\xi_t}$. By Lemma G.4, we can thus write:

$$F_{N,t}(\xi) = F_{ij,t}(\xi) + e_t(\xi), \qquad \sup_{\xi \in [-\varepsilon, 1+\varepsilon]} |e_t(\xi)| \leq \varepsilon, \qquad \varepsilon \in \mathcal{O}(N \exp(-\kappa)). \qquad \text{(H.318)}$$

We first show existence of an interior equilibrium for $F_{N,t}$, the exact parallel dynamics in Eq. (G.50). We denote $F_{ij,t}$ as the drift of the approximate parallel dynamics in Eq. (H.234). By Prop. 4.5, the exact parallel dynamics has instantaneously stable equilibria near $\xi = 0$ and $\xi = 1$, denoted $\xi_t^{*,i}$ and $\xi_t^{*,j}$, respectively. Instantaneous stability implies that the drift points toward each equilibrium. Hence we can pick points $\xi_L, \xi_R$ with

$$\xi_t^{*,i} < \xi_t^{(L)} < \xi_t^{(R)} < \xi_t^{*,j}, \qquad F_{N,t}(\xi_L) < 0, \qquad F_{N,t}(\xi_R) > 0. \qquad \text{(H.319)}$$

(For example, take $\xi_L$ slightly to the right of $\xi_t^{*,i}$ and $\xi_R$ slightly to the left of $\xi_t^{*,j}$.) Since $F_{N,t}$ is continuous in $\xi$, the intermediate value theorem applied to Eq. (H.319) yields an interior root $\xi_N^\star(t) \in (\xi_t^{(L)}, \xi_t^{(R)})$ with $F_{N,t}(\xi_N^\star(t)) = 0$.

Assume there exists an interval $I$ containing $\xi_{ij}^\star(t)$ such that

$$F_{ij,t}'(\xi) \geq m > 0 \qquad \forall \xi \in I, \qquad\qquad\qquad \text{(H.320)}$$

which is Eq. (B.10). Define

$$\rho := \sup\left\{r > 0 : \; [\xi_{ij}^\star(t) - r, \, \xi_{ij}^\star(t) + r] \subset I\right\}. \qquad\qquad \text{(H.321)}$$

Since $\rho > 0$, recalling that $\varepsilon \leq C_1 \exp(-\kappa)$, we choose $\kappa$ sufficiently large such that:

$$\frac{\varepsilon}{m} < \rho. \qquad\qquad\qquad\qquad \text{(H.322)}$$

Now set $\xi_\pm := \xi_{ij}^\star(t) \pm \varepsilon/m$, so that Eq. (H.322) implies $[\xi_-, \xi_+] \subset I$. Since $F_{ij,t}(\xi_{ij}^\star(t)) = 0$ and $F_{ij,t}' \geq m$ on $[\xi_-, \xi_+] \subset I$, the mean value theorem gives

$$F_{ij,t}(\xi_+) \geq \varepsilon, \qquad F_{ij,t}(\xi_-) \leq -\varepsilon, \qquad\qquad \text{(H.323)}$$

Combining Eqs. (H.318) and (H.323) gives

$$F_{N,t}(\xi_+) \geq \varepsilon - \varepsilon \geq 0, \qquad F_{N,t}(\xi_-) \leq -\varepsilon + \varepsilon \leq 0.$$

By continuity, there exists a root $\xi_N^\star(t) \in [\xi_-, \xi_+]$, and therefore

$$\left|\xi_N^\star(t) - \xi_{ij}^\star(t)\right| \leq \frac{\varepsilon}{m}, \qquad\qquad\qquad \text{(H.324)}$$

which is Eq. (B.11).

This completes the proof. $\qquad\qquad\qquad\qquad\qquad\qquad\qquad\qquad\qquad\qquad\qquad$ $\square$

## H.8. Proof of Prop. C.4

*Proof.* The approximate parallel drift along $L_t^{(i,j)}$ is $F_{ij,t}(\xi) = \tilde{\gamma}_j^{(ij)}(\boldsymbol{y}(\xi)) - \xi$, where $\boldsymbol{y}(\xi) = \boldsymbol{\mu}^{(i)} + \xi\ell\boldsymbol{u}$ and $\boldsymbol{u} = (\boldsymbol{\mu}^{(j)} - \boldsymbol{\mu}^{(i)})/\ell$. Under score error $\psi$, the perturbed PF ODE (C.13) induces perturbed parallel dynamics. Projecting onto $\boldsymbol{u}$, the drift becomes

$$\tilde{F}_{ij,t}(\xi) = F_{ij,t}(\xi) + e_t(\xi), \quad \text{where} \quad e_t(\xi) := \frac{\sigma_t^2}{\sqrt{\bar{\alpha}_t}\ell} \langle \psi(\sqrt{\bar{\alpha}_t}\boldsymbol{y}(\xi), t), \boldsymbol{u}\rangle. \qquad \text{(H.325)}$$

By Cauchy-Schwarz and Asm. C.2,

$$|e_t(\xi)| \leq \frac{\sigma_t^2 \|\psi(\sqrt{\bar{\alpha}_t}\boldsymbol{y}(\xi), t)\|_2 \cdot \|\boldsymbol{u}\|_2}{\sqrt{\bar{\alpha}_t}\ell} \leq \frac{\bar{\varrho}(t)}{\ell}. \tag{H.326}$$

Define $G : [0, T] \times [0, 1] \to \mathbb{R}$ by $G(t, \xi) := \tilde{F}_{ij,t}(\xi)$. At the unperturbed equilibrium with $\varrho = 0$, we have $G(t, \xi_{ij}^*(t))|_{\varrho=0} = F_{ij,t}(\xi_{ij}^*(t)) = 0$. The partial derivative satisfies

$$\left.\frac{\partial G}{\partial \xi}\right|_{\xi=\xi_{ij}^*(t)} = F'_{ij,t}(\xi_{ij}^*(t)) + e'_t(\xi_{ij}^*(t)) = \lambda_t + O(\sigma_t^2 L_\psi(t)), \tag{H.327}$$

where $|e'_t(\xi)| = |\sigma_t^2 \boldsymbol{u}^\top \nabla_{\boldsymbol{x}}\psi(\sqrt{\bar{\alpha}_t}\boldsymbol{y}(\xi), t)\,\boldsymbol{u}| \leq \sigma_t^2 L_\psi(t)$ by Asm. C.3. For $L_\psi(t) \leq \frac{\lambda_t}{2\sigma_t^2}$, we have $\partial G/\partial \xi \geq \lambda_t/2 > 0$, so by the implicit function theorem there exists a unique $\xi_\theta^*(t)$ near $\xi_{ij}^*(t)$ with $G(t, \xi_\theta^*(t)) = 0$.

Taylor expanding $G(t, \xi)$ around $\xi_{ij}^*(t)$:

$$0 = G(t, \xi_\theta^*(t)) = \underbrace{F_{ij,t}(\xi_{ij}^*(t))}_{=0} + e_t(\xi_{ij}^*(t)) + (\lambda_t + O(\sigma_t^2 L_\psi(t)))(\xi_\theta^* - \xi_{ij}^*) + O(|\xi_\theta^* - \xi_{ij}^*|^2). \tag{H.328}$$

Rearranging and using (H.326):

$$|\xi_\theta^*(t) - \xi_{ij}^*(t)| \leq \frac{2|e_t(\xi_{ij}^*(t))|}{\lambda_t} + O(\bar{\varrho}(t)^2) \leq \frac{2\bar{\varrho}(t)}{\ell\lambda_t} + O(\bar{\varrho}(t)^2). \tag{H.329}$$

$\square$

## H.9. Proof of Prop. C.5

*Proof.* The perturbed parallel drift is $\tilde{F}_{ij,t}(\xi) = F_{ij,t}(\xi) + e_t(\xi)$, with derivative

$$\tilde{F}'_{ij,t}(\xi) = F'_{ij,t}(\xi) + e'_t(\xi). \tag{H.330}$$

Since $e_t(\xi) = \frac{\sigma_t}{\sqrt{\bar{\alpha}_t}\ell}\langle\psi(\sqrt{\bar{\alpha}_t}\boldsymbol{y}(\xi), t), \boldsymbol{u}\rangle$ and $\frac{d\boldsymbol{y}}{d\xi} = \ell\boldsymbol{u}$,

$$e'_t(\xi) = \sigma_t^2 \boldsymbol{u}^\top \nabla_{\boldsymbol{x}}\psi(\sqrt{\bar{\alpha}_t}\boldsymbol{y}(\xi), t)\,\boldsymbol{u} \tag{H.331}$$

At the perturbed equilibrium $\xi_\theta^*(t)$:

$$\lambda_\theta(t) := \tilde{F}'_{ij,t}(\xi_\theta^*(t)) = F'_{ij,t}(\xi_\theta^*(t)) + \sigma_t^2 \boldsymbol{u}^\top \nabla_{\boldsymbol{x}}\psi(\sqrt{\bar{\alpha}_t}\boldsymbol{y}(\xi_\theta^*), t)\,\boldsymbol{u} \tag{H.332}$$

$$= F'_{ij,t}(\xi_{ij}^*(t)) + O(|\xi_\theta^* - \xi_{ij}^*|) + \sigma_t^2 \boldsymbol{u}^\top \nabla_{\boldsymbol{x}}\psi(\sqrt{\bar{\alpha}_t}\boldsymbol{y}(\xi_\theta^*), t)\,\boldsymbol{u} \tag{H.333}$$

$$= \lambda_t + \sigma_t^2 \boldsymbol{u}^\top \nabla_{\boldsymbol{x}}\psi(\sqrt{\bar{\alpha}_t}\boldsymbol{y}(\xi_\theta^*), t)\,\boldsymbol{u} + O(\bar{\varrho}(t)), \tag{H.334}$$

where the second line uses Lipschitz continuity of $F'_{ij,t}(\xi)$: by the mean value theorem,

$$|F'_{ij,t}(\xi_\theta^*) - F'_{ij,t}(\xi_{ij}^*)| \leq \sup_{\xi\in[0,1]} |F''_{ij,t}(\xi)| \cdot |\xi_\theta^* - \xi_{ij}^*| = O(|\xi_\theta^* - \xi_{ij}^*|), \tag{H.335}$$

and the last line uses Prop. C.4. The two items follow from the sign of $\sigma_t^2 \boldsymbol{u}^\top \nabla_{\boldsymbol{x}}\psi(\sqrt{\bar{\alpha}_t}\boldsymbol{y}(\xi_\theta^*), t)\,\boldsymbol{u}$. $\square$

## H.10. Proof of Prop. C.6

*Proof.* We use the same relabeling as the proof of Prop. 5.1. For $a \geq \vartheta$, the drift satisfies $b(a, t) \geq \lambda_{\text{rep}}a \geq \lambda_{\text{rep}}\vartheta$. The perturbed drift satisfies

$$b_\theta(a, t) = b(a, t) + \tilde{e}_t(a) \geq \lambda_{\text{rep}}a - |\tilde{e}_t(a)| \geq \lambda_{\text{rep}}a - r_A(t) \geq \lambda_{\text{rep}}a - r_{A,\max}. \tag{H.336}$$

Under the condition $r_{A,\max} < \lambda_{\text{rep}}\vartheta$, since $a \geq \vartheta$:

$$b_\theta(a, t) \geq \left(\lambda_{\text{rep}} - \frac{r_{A,\max}}{\vartheta}\right)a = \tilde{\lambda}_{\text{rep}}\,a, \tag{H.337}$$

where $\tilde{\lambda}_{\text{rep}} > 0$ by assumption. The symmetric argument holds for $a \leq -\vartheta$, so the escape mechanism of Prop. 5.1 applies with $\lambda_{\text{rep}}$ replaced by $\tilde{\lambda}_{\text{rep}}$. $\square$

### H.11. Proof of Proposition D.1

*Proof.* Fix any $t \leq \tau_1$ and write $\boldsymbol{p} := \boldsymbol{y}_t$ for the state at time $t$. We first prove the claim for a single rectangular cell.

Consider a single grid cell with corners (true modes) $\boldsymbol{\mu}^{(1)}, \boldsymbol{\mu}^{(2)}, \boldsymbol{\mu}^{(3)}, \boldsymbol{\mu}^{(4)}$, where the diagonal pairs are $(1,4)$ and $(2,3)$. Define $d_a^2 := \|\boldsymbol{\mu}^{(a)} - \boldsymbol{p}\|_2^2$. For any rectangle, one has the identity

$$d_1^2 + d_4^2 = d_2^2 + d_3^2. \tag{H.338}$$

(For example, place the rectangle at $(0,0), (L,0), (0,H), (L,H)$ and expand the four squared distances to verify the equality for all $\boldsymbol{p}$.)

Assume the dominant pair $(i,j)$ from Asm. 4.1 corresponds to the diagonal $(1,4)$ of this cell. Let

$$m := \min\{d_1^2, d_4^2\}, \qquad \delta := |d_1^2 - d_4^2|.$$

Then (wlog, say $d_1^2 = m$ and $d_4^2 = m + \delta$) we have

$$\frac{d_1^2 + d_4^2}{2} = m + \frac{\delta}{2}. \tag{H.339}$$

Also, using $\min\{a,b\} \leq (a+b)/2$ and Eq. (H.338),

$$\min\{d_2^2, d_3^2\} \leq \frac{d_2^2 + d_3^2}{2} = \frac{d_1^2 + d_4^2}{2} = m + \frac{\delta}{2}. \tag{H.340}$$

By Asm. 4.1 for the pair $(1,4)$, every non-dominant mode $k \notin \{1,4\}$ satisfies

$$d_k^2 \geq m + \Delta_t.$$

In particular,

$$d_2^2 \geq m + \Delta_t, \qquad d_3^2 \geq m + \Delta_t. \tag{H.341}$$

Combining Eq. (H.341) with Eq. (H.340) yields

$$m + \Delta_t \leq m + \frac{\delta}{2} \implies \Delta_t \leq \frac{\delta}{2}. \tag{H.342}$$

Let $\tilde{\gamma}_{a,t}$ denote the (rescaled) responsibilities as in Lemma G.2; for brevity write $\gamma_a := \tilde{\gamma}_{a,t}$. By Eq. (H.241), for any $a, b$, we have that:

$$\frac{\gamma_a}{\gamma_b} = \frac{\pi_a}{\pi_b} \exp\left(\frac{d_b^2 - d_a^2}{2\tilde{\sigma}_t^2}\right).$$

Applying this with $(a,b) = (1,4)$ and taking logs gives

$$d_4^2 - d_1^2 = 2\tilde{\sigma}_t^2 \left(\log \frac{\gamma_1}{\gamma_4} - \log \frac{\pi_1}{\pi_4}\right),$$

hence

$$\delta = |d_4^2 - d_1^2| \leq 2\tilde{\sigma}_t^2 \left(\left|\log \frac{\gamma_1}{\gamma_4}\right| + \left|\log \frac{\pi_1}{\pi_4}\right|\right). \tag{H.343}$$

Plug Eq. (H.343) into Eq. (H.342):

$$\Delta_t \leq \tilde{\sigma}_t^2 \left(\left|\log \frac{\gamma_1}{\gamma_4}\right| + \left|\log \frac{\pi_1}{\pi_4}\right|\right). \tag{H.344}$$

By Asm. 4.1, $\Delta_t \geq 2\tilde{\sigma}_t^2 \kappa$, so

$$2\kappa \leq \left|\log \frac{\gamma_1}{\gamma_4}\right| + \left|\log \frac{\pi_1}{\pi_4}\right|.$$

Exponentiating and using $\exp(|\log x|) = \max\{x, 1/x\}$ yields

$$e^{2\kappa} \leq \max\left\{\frac{\gamma_1}{\gamma_4}, \frac{\gamma_4}{\gamma_1}\right\} \cdot \max\left\{\frac{\pi_1}{\pi_4}, \frac{\pi_4}{\pi_1}\right\}. \tag{H.345}$$

Define $\rho_t := \min\{\gamma_1, \gamma_4\}$. Since $\gamma_a \in (0, 1)$ and $\gamma_1 + \gamma_4 \leq 1$, we have $\max\{\gamma_1, \gamma_4\} \leq 1 - \rho_t$, hence

$$\max\left\{\frac{\gamma_1}{\gamma_4}, \frac{\gamma_4}{\gamma_1}\right\} = \frac{\max\{\gamma_1, \gamma_4\}}{\min\{\gamma_1, \gamma_4\}} \leq \frac{1 - \rho_t}{\rho_t}. \tag{H.346}$$

Let $C_\pi := \max\{\pi_1/\pi_4, \pi_4/\pi_1\}$. Combining Eqs. (H.345) and (H.346) gives

$$e^{2\kappa} \leq C_\pi \frac{1 - \rho_t}{\rho_t} \quad \implies \quad \rho_t \leq \frac{C_\pi}{C_\pi + e^{2\kappa}} \leq C_\pi e^{-2\kappa}.$$

Since $\rho_t = \min\{\gamma_1, \gamma_4\}$ and $\gamma_a = \tilde{\gamma}_{a,t}$, we have shown

$$\min\{\tilde{\gamma}_{1,t}, \tilde{\gamma}_{4,t}\} \leq C_\pi e^{-2\kappa} \qquad \forall\, t \leq \tau_1.$$

This proves the desired $\mathcal{O}(e^{-2\kappa})$ bound for a single diagonal pair.

Next, we extend this to the rest of the cell. If $(i, j)$ is a diagonal pair of some grid cell, let $(k, \ell)$ be the other two corners of that cell. Then the rectangle identity Eq. (H.338) holds for the quadruple $(i, k, \ell, j)$ for the same point $\boldsymbol{p} = \boldsymbol{y}_t$, and Asm. 4.1 for $(i, j)$ implies $d_k^2, d_\ell^2 \geq m + \Delta_t$ just as in Eq. (H.341). Repeating the above argument with the relabeled indices gives $\min\{\tilde{\gamma}_{i,t}, \tilde{\gamma}_{j,t}\} \leq C_{\pi,ij} e^{-2\kappa}$ where $C_{\pi,ij} := \max\{\pi_i/\pi_j, \pi_j/\pi_i\}$. Lastly, since the cell chosen was arbitrary, this holds for all (i,j) in the grid, and we are done. $\qquad\square$

# I. Use of the Exact Score

In what follows, we detail why we use the exact score in our theoretical analysis and how this connects to the realistic learned score setting which we aim to study. Since the marginals under the exact score agree for the ODE and SDE as discussed in Sec. 2, the hallucination rate for the ODE and SDE agree under the exact score (in particular, it is 0). However, we can still leverage the ODE/SDE under the exact score to obtain a *mechanism* which becomes useful and predictive in the learned score setting. In particular, assuming the exact score as regularity, we show that $\boldsymbol{x}_t$ under the ODE can get stuck (Prop. 4.7) while $\boldsymbol{x}_t$ under the SDE can escape (Prop. 5.1). We study the exact score scenario in our main paper because it provides the cleanest presentation of this vulnerability. Our experiments then demonstrate that the trapping region in Prop. 4.7 characterizes hallucination differences between DDIM and DDPM in the realistic learned score scenario (e.g., in Figure 5).

To make this mathematically precise, we have $p_t^{\text{DDIM,exact}} = p_t^{\text{DDPM,exact}}$. Let $H := \{\text{mode interpolation occurs}\}$. Then, we have $\Pr^{\text{DDIM,exact}}(H) = \Pr^{\text{DDPM,exact}}(H)$. However, we are interested in explaining the observed differences in hallucination rates in the realistic, practical learned score setting where $\Pr^{\text{DDIM,learned}}(H) \gg \Pr^{\text{DDPM,learned}}(H)$ (Figure 1). To bridge this gap, we must look beyond unconditional marginals $p_t$ and examine the *conditional trajectory dynamics* $p(\boldsymbol{x}_0 \mid \boldsymbol{x}_{\tau_3})$, which differ entirely for DDIM and DDPM. We find that we can extract a difference in DDIM (Prop. 4.7) and DDPM (Prop. 5.1) under the exact score which uncovers the trapping mechanism driving $\Pr^{\text{DDIM,learned}}(H) \gg \Pr^{\text{DDPM,learned}}(H)$. Concretely, let $M_{\vartheta,\tau_3} := \{\text{trajectory enters } \vartheta\text{-midpoint neighborhood during final } \tau_3 \text{ steps}\}$. Then:

$$\Pr(H) = \Pr(H \mid M_{\vartheta,\tau_3})\Pr(M_{\vartheta,\tau_3}) + \Pr(H \mid M_{\vartheta,\tau_3}^c)\Pr(M_{\vartheta,\tau_3}^c) \tag{I.347}$$

The contributions of our exact score theory in Prop. 4.7 and Prop. 5.1 are twofold:

1. The midpoint neighborhood and $M_{\vartheta,\tau_3}$ are driving the differences in hallucination rates between DDIM and DDPM, i.e. $\Pr(H \mid M_{\vartheta,\tau_3}^c)$ is small and the first term dominates so that $\Pr(H) \approx \Pr(H \mid M_{\vartheta,\tau_3})\Pr(M_{\vartheta,\tau_3})$. This is done in Prop. 4.7.

2. Demonstrate that $\Pr^{\text{DDIM,exact}}(H \mid M_{\vartheta,\tau_3}) \gg \Pr^{\text{DDPM,exact}}(H \mid M_{\vartheta,\tau_3})$. This is done in Prop. 5.1.

| sampler | $P(H)$ | $P(H \mid M)$ | $P(H \mid M^c)$ | $P(M)$ | $P(H \mid M)P(M)$ | $P(H \mid M^c)P(M^c)$ |
|---------|--------|----------------|------------------|--------|--------------------|------------------------|
| DDIM | 0.078 | 0.735 | 0.006 | 0.099 | 0.073 | 0.005 |
| DDPM | 0.005 | 0.103 | 0.001 | 0.042 | 0.004 | 0.001 |

*Table I.1.* Hallucination probabilities in the learned score case, averaged across 100k trajectories, with $\tau_3 = 11$ and $\vartheta = 0.35\ell_t$. Here, $H$ denotes the event that the final generated sample is a mode interpolation, and $M := M_{\vartheta,\tau_3}$ denotes the event that the trajectory enters the $\vartheta$-midpoint neighborhood of the nearest $i, j$-mode segment during the final $\tau_3$ steps. We observe that $\mathrm{Pr}^{\mathrm{DDIM}}(H \mid M) \gg \mathrm{Pr}^{\mathrm{DDPM}}(H \mid M)$, while $\mathrm{Pr}(H \mid M^c)$ remains small for both samplers. Thus, the dominant contribution comes from the first decomposition term $\mathrm{Pr}(H \mid M)\,\mathrm{Pr}(M)$, supporting the conclusion that midpoint-neighborhood entry is the key event relevant for hallucinations in the learned score case.

These results arise due to the differences in (conditional) dynamics, even though the marginals are the same under the exact score. The exact score assumption in our theory allows us to demonstrate this cleanly. Still, $\mathrm{Pr}^{\mathrm{DDIM,exact}}(M_{\vartheta,\tau_3}) = \mathrm{Pr}^{\mathrm{DDPM,exact}}(M_{\vartheta,\tau_3}) = 0$.

But, in the practical learned score setting where $\mathrm{Pr}^{\mathrm{learned}}(M_{\vartheta,\tau_3}) \neq 0$ for both samplers, the fundamental difference $\mathrm{Pr}^{\mathrm{DDIM,learned}}(H \mid M_{\vartheta,\tau_3}) \gg \mathrm{Pr}^{\mathrm{DDPM,learned}}(H \mid M_{\vartheta,\tau_3})$, which persists in the learned score case but is uncovered via our exact score analysis, helps explain $\mathrm{Pr}^{\mathrm{DDIM,learned}}(H) \gg \mathrm{Pr}^{\mathrm{DDPM,learned}}(H)$. Empirically, in Tab. I.1, we find that the first term in Eq. (I.347) remains dominant while $\mathrm{Pr}^{\mathrm{DDIM,learned}}(H \mid M_{\vartheta,\tau_3}) \gg \mathrm{Pr}^{\mathrm{DDPM,learned}}(H \mid M_{\vartheta,\tau_3})$.

# J. Symbol Table

| Symbols | |
|---|---|
| $\varpi$ | Dimension of all vectors throughout. |
| $p_{\text{base}}$ | The distribution which the diffusion sampling process starts from. Throughout, we take this to be $\mathcal{N}(0, \boldsymbol{I})$, the standard Gaussian. |
| $p_{\text{data}}$ | The distribution which the diffusion process ends at and aims to sample from. Throughout, we take this to be a $N$-component Gaussian mixture. |
| $\boldsymbol{x}_t$ | A trajectory satisfying the PF ODE given in Eq. (3) for DDIM or Eq. (2) for DDPM, given in reverse time from $t = T$ to $t = 0$ unless otherwise specified. |
| $\beta(t)$ | Diffusion noise schedule. |
| $\boldsymbol{W}_t$ | Standard $\varpi$-dimensional Brownian motion |
| $\bar{\alpha}_t$ | Amount of signal from data distribution retained in forward or reverse process. Given by $\exp(-\int_0^t \beta(s)\,ds)$ |
| $p_t$ | Marginal distribution of diffusion process instance $\boldsymbol{x}_t$ at time $t$; equivalent in forward and reverse process. |
| $\nabla_{\boldsymbol{x}_t} \log p_t(\boldsymbol{x}_t)$ | Exact score function of marginal $p_t$. |
| $\boldsymbol{I}$ | Identity matrix. |
| $\mathcal{N}(\boldsymbol{x}; \boldsymbol{\mu}, \sigma^2 \boldsymbol{I})$ | Probability density function of Gaussian distribution centered at $\boldsymbol{\mu}$ with isotropic covariance matrix $\sigma^2 \boldsymbol{I}$. |
| $\bar{\boldsymbol{W}}_t$ | Standard Brownian motion in reverse time. |
| $s_{\boldsymbol{\theta}}(\boldsymbol{x}_t)$ | Learned score function. |
| $\pi_i$ | Coefficient of Gaussian mixture. |
| $N$ | Number of components in Gaussian mixture. |
| $\boldsymbol{\mu}^{(i)}$ | $i$th mode of Gaussian mixture, $i \in [N]$. |
| $\sigma^2$ | Variance of Gaussian mixture, i.e. each component has covariance $\sigma^2 \boldsymbol{I}$. |
| $\boldsymbol{\mu}_t^{(i)}$ | Time-dependent modes of $p_t$ given Gaussian mixture target. |
| $\sigma_t^2$ | Time-dependent variance of $p_t$ given Gaussian target. |
| $\tilde{\sigma}_t^2$ | Rescaled time-dependent variance of $p_t$. |
| $L^{(i,j)}$ | Line segment joining mode $i$ and mode $j$. |
| $L_t^{(i,j)}$ | Line segment joining time-dependent mode $i$ and time-dependent mode $j$. |
| $L_{t,\varepsilon}^{(i,j)}$ | $\varepsilon$-extended line segment joining time-dependent mode $i$ and time-dependent mode $j$. |
| $d_\perp$ | Orthogonal distance between trajectory and (nearby) line segment. |

## Symbols

| | |
|---|---|
| $\text{Tube}_{t,\zeta}^{(i,j)}$ | Tube around $L_t^{(i,j)}$ of radius $\zeta$. Throughout, $\zeta = \varepsilon$, unless specified otherwise. |
| $\Delta_t$ | Time-dependent margin between modes. Must be sufficiently large after time in reverse process w.r.t. $\tilde{\sigma}_t^2$. |
| $\kappa$ | Margin coefficient, defining gap between $\tilde{\sigma}_t^2$ and $\Delta_t$. Also used for margin w.r.t. $\ell^2$. |
| $\tau_1$ | Critical time for Asm. 4.1 to hold. |
| $\tau$ | Time which is less than $\tau_1$; critical time for exponential convergence in Theorem 4.2 to hold. |
| $\tau_2$ | Critical time for Asm. 4.4 to hold. Empirically strictly less than $\tau_1$. |
| $\hat{\tau}$ | Minimum of $\tau$ and $\tau_2$. Critical time for Prop. 4.5 to hold. |
| $\tau_3$ | Critical time less than $\hat{\tau}$ for Prop. 4.7 to hold. |
| $\phi$ | Used to define probabilities in concentration inequalities throughout. |
| $\varepsilon$ | Size by which nearby line segment is extended; radius of containment tube; and radius of ball around modes of mixture where stable equilibria exist. |
| $\xi_t$ | Used to denote parallel dynamics to $L_t^{(i,j)}$. $\xi^*$ is an equilibrium of some kind, stated specifically in each proposition in which it appears. |
| $\boldsymbol{w}_t$ | Used to denote orthogonal dynamics to $L_t^{(i,j)}$. |
| $\boldsymbol{y}_t^*$ | Unstable equilibria at midpoint along $L_t^{(i,j)}$ (w.r.t. approximate dynamics) |
| $\boldsymbol{y}_t$ | Used for time-rescaled dynamics in Lemma G.2, where modes become fixed and effective variance $\sigma_t^2$ is rescaled to $\tilde{\sigma}_t^2$. |
| $\lambda_t$ | Unstable eigenvalue on midpoint, or unstable equilibria on $L_t^{(i,j)}$ more broadly. |
| $\vartheta$ | Radius from midpoint for which, given $\tau_3$ time remaining. |
| $\tilde{\lambda}_t$ | Integral of $\lambda_t$ from time $t_{\nu_1}$ to time $t_{\nu_2}$. |
| $\ell^2$ | Squared distance between (fixed, time-independent) modes. |
| $H^{(i,j)}$ | Bisector hyperplane to $L^{(i,j)}$, i.e. the plane orthogonal to the line which passes through the midpoint of the line. |
| $\boldsymbol{u}$ | The coordinate normal to $H^{(i,j)}$. |
| $A_t$ | One dimensional dynamics of distance from midpoint along $\boldsymbol{u}$, with respect to DDPM. |
| $b(A_t, t)$ | Deterministic dynamics (drift) of $A_t$. |
| $\pi$ | Without a subscript, defines the usual scalar $\pi$. |

| | Symbols |
|---|---|
| $K(t)$ | Controls how fast $b(A_t, t)$ is. |
| $\eta(t)$ | $1/2\beta(t)$. |
| $\delta$ | In the main paper, used to define the terminal confinement events. Used to define residual terms due to $N - 2$ components in appendix proofs. |
| $H_\delta$ | Event that at the end of the reverse process, trajectory stays near the bisector hyperplane. |
| $E_{2\delta}$ | Event that at throughout the reverse process, one enters the midpoint neighborhood. |
| $\lambda_{\text{rep}}$ | Coefficient which controls speed of drift away from bisector hyperplane. |
| $\gamma_k$ | Gaussian responsibilities under Eq. (3). |
| $\tilde{\gamma}_k$ | Gaussian responsibilities under time-rescaled dynamics in Lemma G.2. |
| $\hat{\boldsymbol{\mu}}^{(i)}$ | Convex combination of static modes w.r.t. time-rescaped Gaussian responsibilities. |
| $I(t)$ | Integral of $\beta(s)$ without exponential |
| $\delta_\xi$ | Effect of $N-2$ modes on parallel dynamics to nearby $L_t^{(i,j)}$. |
| $\boldsymbol{E}_t$ | Convex combination of distances from mode $i$, given time-rescaled modes, excluding modes $i$ and $j$. |
| $\text{Proj}_{||}(\boldsymbol{v})$ | Projection onto vector parallel to $\boldsymbol{v}$. |
| $\text{Proj}_{\perp}(\boldsymbol{v})$ | Projection onto vector orthogonal to $\boldsymbol{v}$. |
| $\boldsymbol{\delta_w}$ | Effect of $N - 2$ modes on orthogonal dynamics to nearby $L_t^{(i,j)}$. |
| $d^*$ | Mode which for which the trajectory has minimum distance from one of the two selected (nearby) modes $i$ and $j$. |
| $\mathcal{O}, \Omega, \Theta$ | Used for standard asymptotic notation. |
| $M$ | Defines maximum non-selected mode distance. |
| $R$ | Defines maximum Gaussian mixture weights w.r.t. $d^*$, excluding $\pi_i, \pi_j$. |
| $\theta$ | Used to rescale reverse time into forward time, i.e. $\theta = T - t$. |
| $M, Q$ | For martingale $M$, $Q$ is its quadratic variation. |

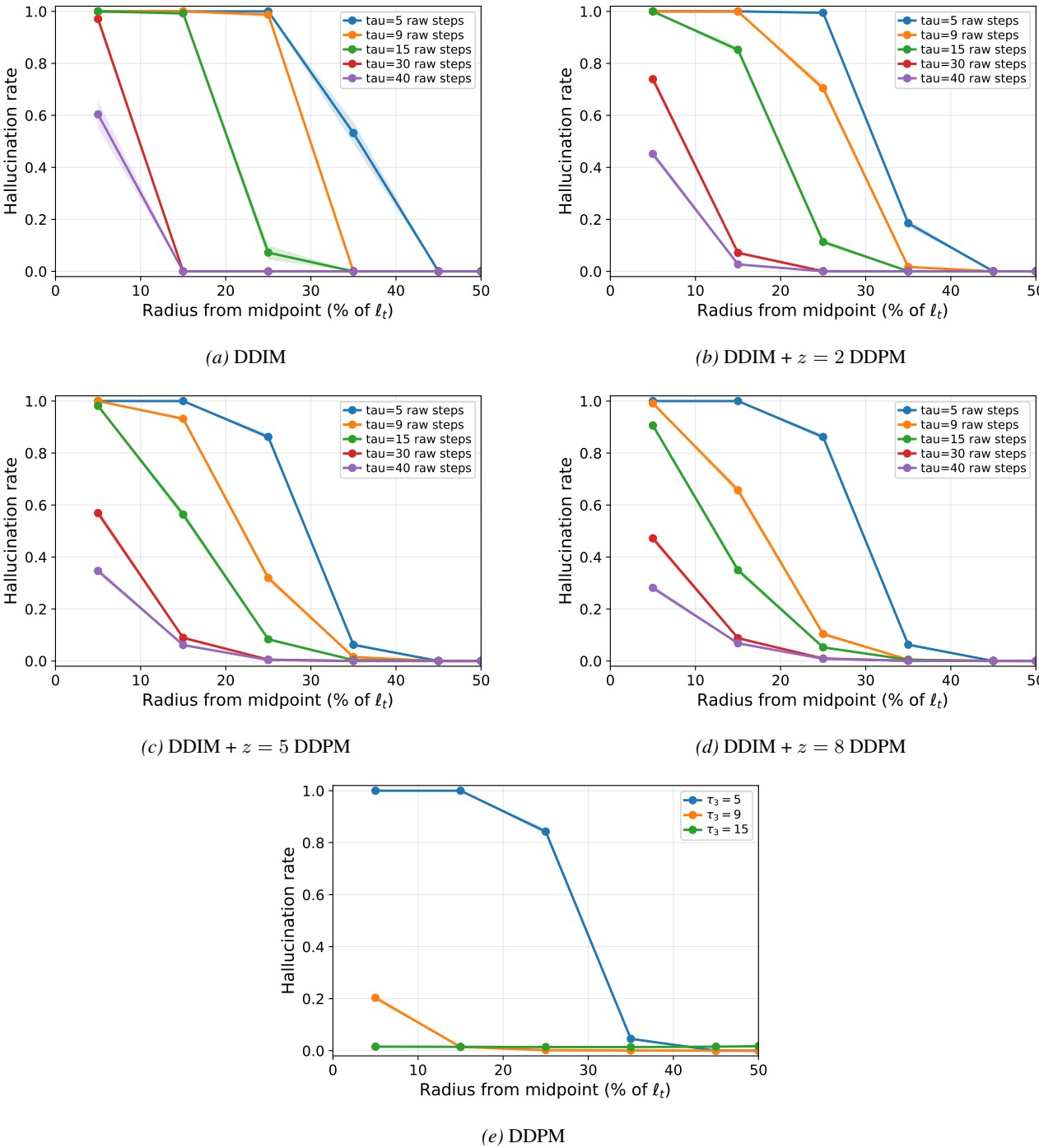

*Figure E.11.* Ablation on $\tau_3$ across various DDIM-DDPM hybrid samplers, starting from the midpoint neighborhood and evaluating hallucination rate. Uncertainty is quantified by the standard error across mode pairs.

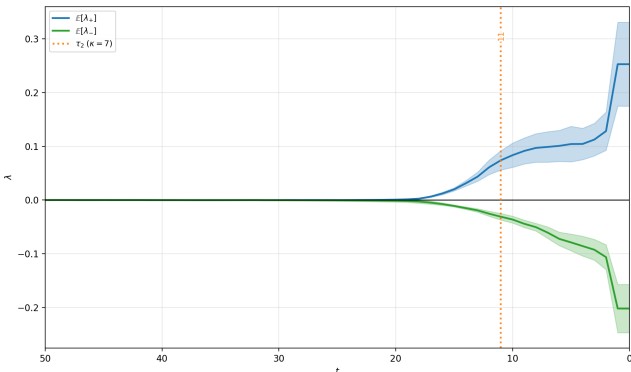

*Figure E.12.* Average DDIM Jacobian eigenvalues evaluated at the midpoint joining mode segments, $\lambda_t$, grouped into positive (blue) and negative (green) groups, during the reverse process. A vertical dotted orange line represents $\tau_2$. We find that there exists both a positive and corresponding negative eigenvalue after time $\tau_2$. Notably, there were no eigenvalues excluded from this plot; all midpoints had one positive eigenvalue and one negative eigenvalue. In particular, the midpoint remains a hyperbolic saddle equilibria with appropriate eigenvalues. This indicates that despite score error and our study of the approximate dynamics, the slowdown region does not greatly shift for a well-trained diffusion model. Finally, note that the Jacobian eigenvalues are identical (up to numerical error) for DDIM and DDPM, since Eq. (3) and Eq. (2) have the same Jacobian.

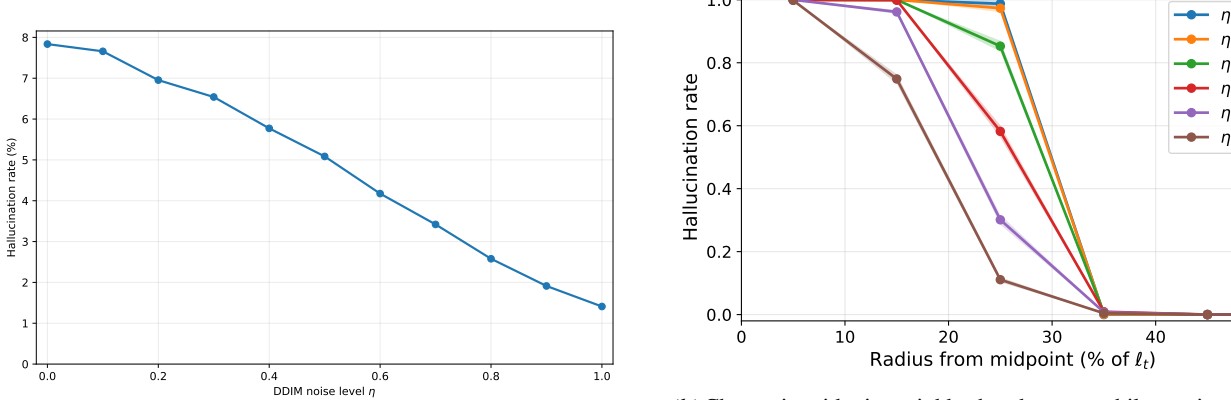

*(a)* Change in hallucination rate while varying DDIM noise level $\eta$

*(b)* Change in midpoint neighborhood escape while varying DDIM noise level $\eta$

*Figure E.13.* Increasing DDIM noise level $\eta$ decreases mode interpolation hallucination rate (Figure E.13a) while helping DDIM trajectories escape the midpoint trapping region (Figure E.13b), demonstrating that SDE noise helps with reducing hallucinations, in line with our theoretical results.

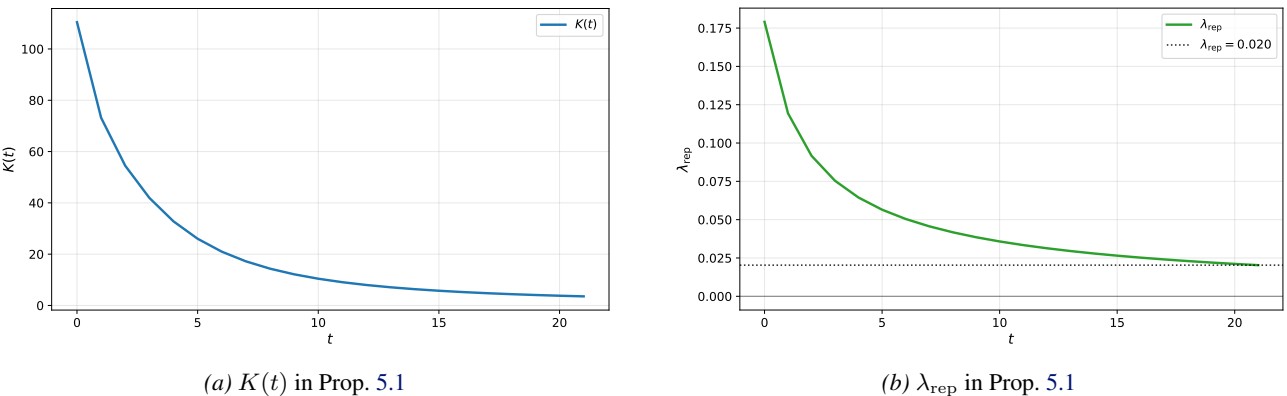

*(a)* $K(t)$ in Prop. 5.1

*(b)* $\lambda_{\text{rep}}$ in Prop. 5.1

*Figure E.14.* $K(t)$ (Figure E.14a) and $\lambda_{\text{rep}}$ (Figure E.14b) computed across a total of 400k samples (10k samples per line segment) under the exact score. We find that $K(t)$ exists and $\lambda_{\text{rep}} > 0$ as desired. Details on how to compute these are included in Sec. F.

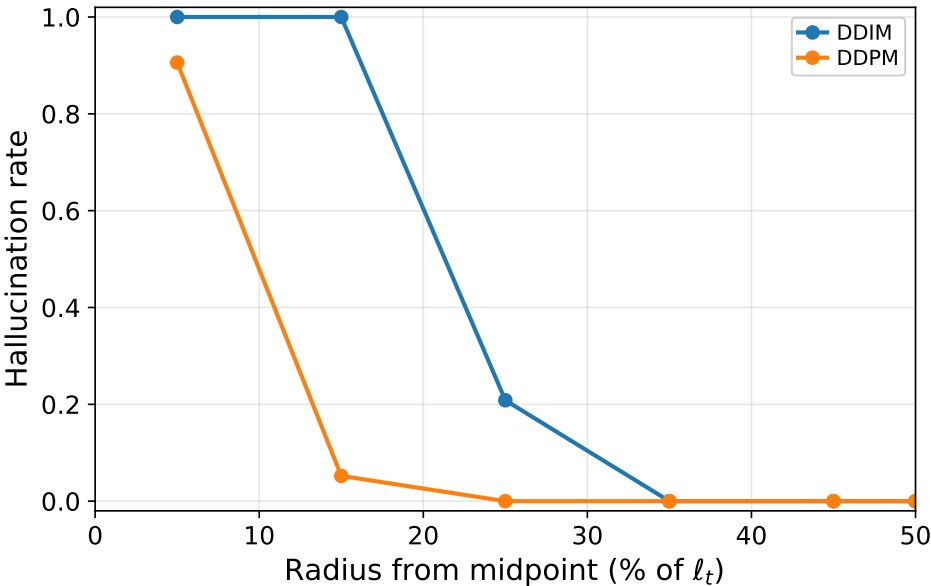

*Figure E.15.* Exact score analogue of Figure 5. We find that at $\vartheta = 0.15\ell_t$ and $\tau_3 = 3$, DDIM gets stuck in the midpoint neighborhood while DDPM escapes, confirming our theoretical results in Prop. 4.7 and Prop. 5.1.

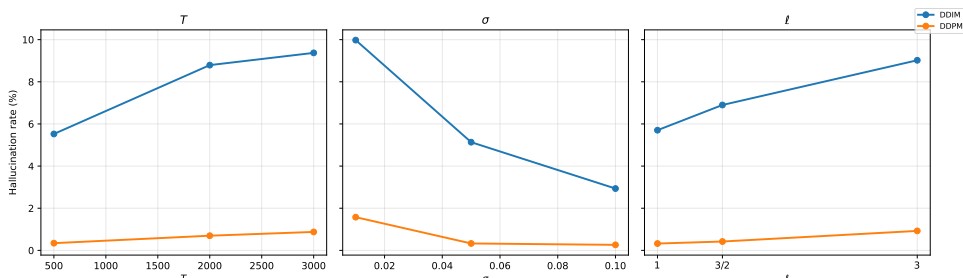

*Figure E.16.* Ablations on hallucination rate gap between DDIM and DDPM across various choices of $\sigma$, $T$, and $\ell$. We find that the hallucination rate gap between DDIM and DDPM persists across several different hyperparameter configurations.

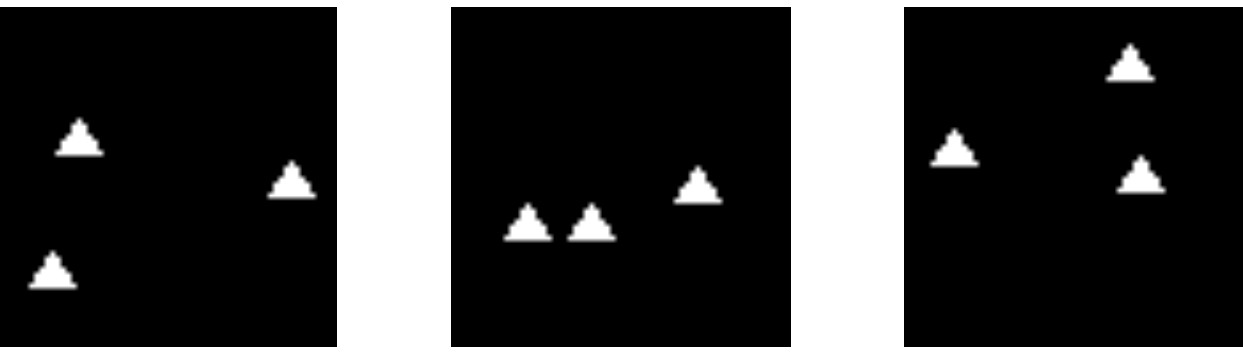

*Figure F.17.* Samples chosen from the dataset of $10,000$ elements containing 3 triangles. The positions are chosen such that the triangles are non-overlapping.

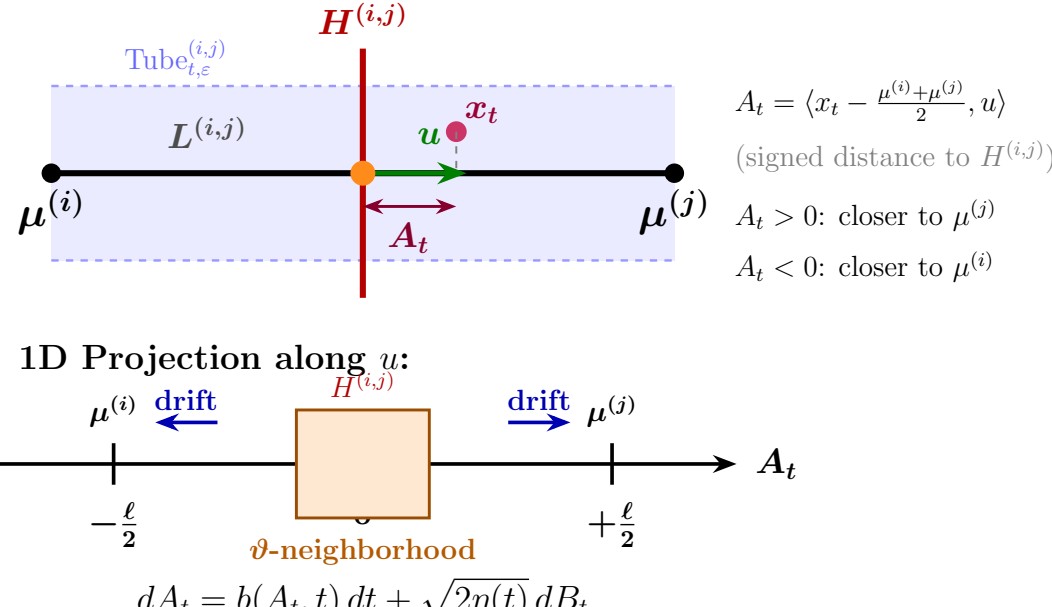

$$A_t = \langle x_t - \frac{\mu^{(i)}+\mu^{(j)}}{2}, u \rangle$$

(signed distance to $H^{(i,j)}$)

$A_t > 0$: closer to $\mu^{(j)}$

$A_t < 0$: closer to $\mu^{(i)}$

$$dA_t = b(A_t, t)\, dt + \sqrt{2\eta(t)}\, dB_t$$

*Figure F.18.* Reduction to 1D dynamics along the bisector normal. **Top**: Inside $\mathrm{Tube}_{t,\varepsilon}^{(i,j)}$, the bisector hyperplane $H^{(i,j)}$ (red) is orthogonal to $L^{(i,j)}$. The signed 1D coordinate $A_t = \langle x_t - \frac{\mu^{(i)}+\mu^{(j)}}{2}, u \rangle$ measures displacement from $H^{(i,j)}$ along the unit normal $u$. **Bottom**: The 1D reduction with modes at $\pm\frac{\ell}{2}$, the $\vartheta$-neighborhood around the midpoint, and unstable drift directions away from the bisector.

