# OpenReview forum: "Why DDIM Hallucinates More Than DDPM: A Theoretical Analysis of Reverse Dynamics"
_ICML.cc/2026/Conference — ICML 2026 regular_

### Official Review · Reviewer_Fvf7 · 2026-02-18

**Soundness:** 3
**Presentation:** 3
**Significance:** 3
**Originality:** 4
**Overall Recommendation:** 5
**Confidence:** 4

**Summary:**

This paper examines why deterministic DDIM sampling tends to produce more “hallucinations” (which are mode-interpolation samples) compared to stochastic DDPM sampling, using an NNN-mode Gaussian mixture as a controlled example. The authors explain that the reverse dynamics happen in two phases: a quick attraction to a nearby line segment between two modes, followed by slower movement along that segment. DDIM can get stuck near the midpoint and fail to reach a true mode. On the other hand, DDPM’s random component makes ending at the midpoint very unlikely under certain drift consistency assumptions, so it usually reaches a true mode instead of interpolating. The paper backs this up with theory, like midpoint trapping for DDIM and a probability bound for DDPM, along with experiments showing that DDIM’s problem is not mainly due to step-skipping.

**Compliance With Llm Reviewing Policy:**

Affirmed.

**Key Questions For Authors:**

See the weaknesses

**Limitations:**

Yes

**Strengths And Weaknesses:**

Strengths

* The theoretical explanation is clear: DDIM tends to get stuck near a midpoint (mode interpolation), while DDPM noise helps it escape. The paper clearly formalizes both aspects with propositions.

* The experiments directly test a common belief among practitioners (that “DDIM hallucinates because of step skipping”) and show that the hallucination rate stays about the same even when increasing DDIM steps. This serves as a useful sanity check.

Weaknesses

1. About Proposition 4.7. The midpoint is described as an instantaneously unstable equillibrium along the segment (positive eigenvalue \lambda_t), yet the proposition claims there is a radius \vartheta neighborhood where trajectories cannot leave within a limited remaining time. This makes sense because the instability is limited by the short time horizon and contraction from other directions, but as it stands, it can seem counterintuitive. Also, the way \vartheta depends on \lambda, \bar\lambda, and \tau_3 is quite specific and might be fragile. The main text should include a more intuitive explanation of why instability does not conflict with time-limited trap.

2. The paper admits that proposition 4.7 uses approximated dynamics and leave out the term caused by the other modes, saying it mostly just shifts the trap location a bit. But, since the key issue is being stuck near a special point on the segment, even small systematic drifts might effect whether the trajectory escapes in finite steps. I would have prefered a clearer bound showing how the \varepsilon perturbation actually changes the hallucination probability, rather than just a qualitative comment.

3. The DDPM “terminal midpoint bound” is a good result, but conditions (5) and (6) are quite abstract, and it’s not clear how often they apply to the actual drift caused by the mixture score. Proposition 5.1’s bound depends on having a function K(t) so that |b(a,t)| ≤ K(t)|a| in a slab, plus a repulsion condition with parameter \lambda_{\text{rep}}. These are reasonable local lipschitz and repelling drift conditions, but the paper doesn’t show how to calculate or estimate \lambda_{\text{rep}} for their mixture case, or how sensitive the result is if these conditions only hold approximately. This makes the theorem sound like: “if the drift behaves well, then noise helps,” which is something we already expected.

4. Empirical evaluation is somewhat narrow relative to the claimed motivation (hallucinations in modern diffusion models). The strongest experiments are on a 2D grid Gaussian mixture, and the DDIM-vs-DDPM gap is clear there, but it’s still a stylized definition of hallucination (mode interpolation). The paper mentions additional image-related evidence elsewhere, but in the main experiments the connection to “semantic constraint violation” is indirect, and it’s not obvious that midpoint interpolation is the dominant failure mode in real generative models.

5. There are some minor presentation and notation issues that make the paper harder to read. At times, the narrative is dense and key quantities like \tau_1, \tau_2, \tau_3, \vartheta, and the reduction to the 1D coordinate A_t appear quickly. I had to reread the trapping statement and the role of the remaining time \tau_3 a few times. Also, a few lines have small typos or formatting glitches, like misplaced hats or spacing. These aren’t serious problems but do make it harder to follow the math.

---

> ### Author Rebuttal · Authors · 2026-03-31
>
> We thank the reviewer Fvf7 for their thorough and constructive review. We especially appreciate their recognition of our work's originality, that our theoretical explanation clearly formalizes both the trapping and escape mechanisms, and that our step-skipping experiment serves as a useful sanity check against a common practitioner belief. Below, we address the reviewer’s questions:
>
> ### > **Q1** In Prop. 4.7, does instability conflict with the midpoint trapping mechanism?
>
> No. Instability is an asymptotic statement, while trapping is a finite-horizon one. The midpoint repels trajectories, but at a bounded rate. If a trajectory starts close enough to the midpoint, the bounded growth over the short remaining time budget $\tau_3$ cannot push it outside the neighborhood. This is stated precisely in Prop. 4.7 item 2 (line 261): a necessary condition for exit is that the remaining time satisfies $\tau_3 \geq \frac{1}{\Lambda_+} \log(\vartheta / |\xi_{\tau_3} - \frac{1}{2}|)$. When this condition is not met, the trajectory can be trapped despite the positive eigenvalue.
>
> ### > **Q2** Prop. 4.7 uses the approximate parallel dynamics. How does the perturbation from the N-2 other modes quantitatively affect the result?
>
> It does not significantly affect it. Empirically, Prop. 4.7 under the exact dynamics is verified in Fig. 5. In this case, trajectories that are within $\vartheta = 0.25\ell_t$ of the midpoint are trapped for $\tau_3 = 11$. From a theoretical standpoint, we provide the key ingredients for this in Appendix B. In particular, the exact parallel dynamics differ from the two-mode approximate dynamics by at most $\varepsilon = O(N \exp(-\kappa))$ (Eq. H.312), and Prop. B.1 item 2 (Eq. B.11) shows that the exact saddle shifts an amount smaller than $\varepsilon$ from the approximate one. This is also demonstrated empirically in Fig. E.12.
>
> ### > **Q3** How do the conditions on $K(t)$ and $\lambda_{\text{rep}}$ in Prop. 5.1 connect to the drift under mixture $p_t$?
>
> In our Gaussian mixture setting, the drift on the bisector coordinate $b(a, t)$ can be computed in closed form using the two-mode responsibility (Eq. H.235). We can then compute $\lambda_{\text{rep}}$ and $K(t)$ by evaluating $b(a,t)$ for a grid of $a \in [-2\vartheta, 2\vartheta]$ and $t \in [0, \tau_3]$, then bounding it appropriately. Below, we compute these quantities for $\vartheta = 0.25\ell_t$ and verify that both conditions hold in our experimental setting. Please view our results [here](https://anonymous.4open.science/r/ddpm_ddim_rebuttal_figs_anon-373C/prop_5_1_assumps.pdf).
>
> We will upload exactly how to compute $b(a,t)$ from the two-mode responsibilities in the camera-ready version. We also note a typo for clarity; in Prop. 5.1, $T$ should be replaced with $\tau_3$ systematically.
>
> ### > **Q4** Is mode interpolation a failure mode in real-world generative models?
>
> Yes, we believe it is. This is highlighted by [1], which shows in Fig. 10 that midpoint interpolation arises in the latent space of images as well. Our primary experiments use Gaussian mixtures as a controlled testbed to verify the tractable mechanism underlying hallucinations given in our theoretical results.
>
> Importantly, we do not claim that this mechanism explains all practical hallucinations. However, we believe that extracting the underlying mechanism in a tractable setting is key to understanding and mitigating diffusion hallucinations.
>
> ### > **Q5** How do other types of hallucination connect to mode interpolation?
>
> We thank the reviewer for raising this point. As demonstrated in Fig. 1 and Fig. 10 of [1], counting hallucinations in Shapes and Hands datasets can also be represented by mode interpolation. The theoretical guarantees of the manuscript focus on a Gaussian mixture distribution. A natural extension of this analysis to images is studying latent diffusion [2] with a Gaussian mixture latent, which we intend to explore in future work.
>
> ### > **Q6** Presentation and notation density.
>
> We appreciate the specific feedback. The roles of $\tau_1$, $\tau_2$, $\tau_3$ are summarized in the remarks following each assumption (lines 196, 243) and in the discussion paragraph after Prop. 4.7 (line 296), and the 1D reduction to $A_t$ is illustrated in Fig. F.14 (Appendix F.3). We acknowledge these could be surfaced more prominently and will address the typos and formatting issues identified in the camera-ready version upon acceptance.
>
> We again thank the reviewer for their detailed feedback and are happy to clarify further if any questions remain.
>
> ### References:
>
> [1] Understanding Hallucinations in Diffusion Models through Mode Interpolation. NeurIPS’24.
>
> [2] High-Resolution Image Synthesis with Latent Diffusion Models. CVPR’22.

---

> > ### Author Rebuttal · Reviewer_Fvf7 · 2026-04-02
> >
> > Thank you for the clear and detailed rebuttal. Here is my final feedback based on your responses:
> >
> > * **Prop 4.7 (Instability vs. Trapping):** Your explanation regarding the finite-horizon bounded growth versus asymptotic instability perfectly resolves the counterintuitive aspect of the trapping mechanysm. Please ensure this exact intuition is explicitly articulated in the main text of the camera-ready version, as it greatly aids reader comprehention.
> > * **Approximate Dynamics:** Highlighting the $\mathcal{O}(\epsilon/m)$ displacement bound from Appendix B satisfies my concern regarding the perturbation from the remaining modes. The quantitative limit on the saddle shift resolves the issue.
> > * **Prop 5.1 Conditions ($K(t)$ and $\lambda_{rep}$):** I appreciate that you successfully calculated these constants for the experimental setting to prove the abstract conditions hold in practice. It is crucial that the explicit derivation of these bounds from the two-mode responsibilities, along with the empirical verification provided in your external link, is fully incorporated into the appendix of the final manuscript.
> > * **Real-world Relevance & Presentation:** I accept your justification regarding latent space mode interpolation as a valid theoretical starting point, supported by Aithal et al. I trust you will follow through on improving the narrative flow for the time variables ($\tau_{1}$, $\tau_{2}$, $\tau_{3}$) and correcting the noted typos (such as the $\vartheta$ vs $\theta$ swap in Prop 5.1).
> >
> > I am maintaining my recommendation of 5.

---

### Official Review · Reviewer_WAef · 2026-03-09

**Soundness:** 3
**Presentation:** 3
**Significance:** 3
**Originality:** 3
**Overall Recommendation:** 4
**Confidence:** 3

**Summary:**

The paper theoretically studies why DDIM appears to hallucinate more than DDPM, under a well-separated Gaussian mixture setting.

**Compliance With Llm Reviewing Policy:**

Affirmed.

**Final Justification:**

The authors have addressed my concerns. I remain my positive score.

**Key Questions For Authors:**

1. DDIM actually admits a large class of simulator of the backward process. In general it can be written as
$x_{t_{i+1}}= \alpha_{t_{i+1}}\hat x_{0,i} +\sqrt{\sigma_{t_{i+1}}^2-c_i^2}\hat{\epsilon_i}+c_i z_i,z_i\sim \mathcal{N}(0,I).$ Here $\hat x_{0,i}$ and $\hat{\epsilon_i}$ are estimated clean sample and noise based on the score and the newly-added noise level $c_i$ can be chosen freely. When $c_i=0$, this becomes DDIM (PF-ODE). Can this work be generalized to general $c_i$? And what might be the optimal noise level in terms of low hallucination?
2. How much practical scope do you want readers to infer beyond Gaussian mixtures?

**Limitations:**

No. I think the limitations section should more explicitly say that the current theory is for well-separated Gaussian mixtures.

**Strengths And Weaknesses:**

Strength:
1. The paper has a strong high-level intuition, and the two-stage (get into the pipe, along the pipe) geometric decomposition is a useful clear way to reason about sampler differences.
2. The paper is clear and readable in genearl, with minor notational confusion. The geometric picture is easy to follow and helpful to get the intuition.


Weakness:
1. Either Eq. (4)  or Proposition 5.1 appears mis-stated: Eq. (4)  defines A_0 = 0, but the proposition then bounds P(|A0| ≤ ϑ).
2. Assumption 4.4 assumes the effective variance is small so DDIM dynamics may get trapped in the midpoint. However, the paper doesn't explain why this assumption is reasonable.
3. The scope is still limited: nearly all theory and empirical evidence is in controlled Gaussian mixtures, and the only non-Gaussian/image evidence I found is a toy dataset of 10,000 64×64 images with three non-overlapping triangles. That makes the paper more convincing as a theoretical/mechanistic study than as a broadly validated account of practical image diffusion hallucinations.

---

> ### Author Rebuttal · Authors · 2026-03-31
>
> We appreciate the reviewer WAef’s compliments on the high-level intuition of our paper as well as the geometric picture our theoretical results illustrate. Below, we address the reviewer’s questions:
>
> > ### **Q1**: Is Assumption 4.4, where the effective variance is sufficiently small, reasonable?
>
> Yes. We empirically verify Asm. 4.4 in Fig. E.6 for large margin values across 100k trajectories, and in Figs. E.7–E.8 for dimensions up to 64. Asm. 4.4 requires that the mixture $p_t$ is well-separated, so that mode interpolation is meaningful as a hallucination definition:  under Asm. 4.4, the regions of high probability mass around the modes of the mixture $p_t$ do not overlap after $\tau_2$.
>
> > ### **Q2**: Do the results extend beyond Gaussian mixtures?
>
> Yes.  Fig. 10 in [1] studies mode interpolation in images, demonstrating that modes correspond to regions of sampled instances in a latent space. Our work instead focuses on the Gaussian mixture target case to tractably extract a mechanism underlying hallucinations. A natural extension of this analysis to images is studying latent diffusion [2] with a Gaussian mixture latent, which we intend to explore in future work.
>
> Importantly, we do not claim mode interpolation is the only type of practical hallucination; however, as highlighted by [1], mode interpolation remains a relevant characterization of hallucination in real-world generative models.
>
> > ### **Q3**: Can the results of this work be generalized to DDIM with varying noise levels?
>
> Not exactly. Any nonzero noise introduces a stochastic term, and the escape bound in Prop. 5.1 applies directly. However, our theory does not characterize how different noise levels differ. As a first step towards this, we provide experiments evaluating the midpoint trapping mechanism while varying the noise level across the general DDIM family, finding that adding more noise leads to a stronger escape mechanism. Please view these results [here](https://anonymous.4open.science/r/ddpm_ddim_rebuttal_figs_anon-373C/ddim_eta_overlay_tau11.pdf).
>
> > ### **Q4**:  Can we find the optimal noise level of DDIM in terms of low hallucination?
>
> We appreciate the reviewer for raising an interesting application of our theoretical results. Our theoretical results highlight the benefits of adding noise in avoiding hallucinations (Prop. 5.1). In line with this, we find that $\eta = 1.0$ has the lowest hallucination rate. That is, in the context of mode interpolations, more noise is better. Please view these results [here](https://anonymous.4open.science/r/ddpm_ddim_rebuttal_figs_anon-373C/ddim_hallucination_rate_v_eta.pdf).
>
> > ### **Q5**: Is Eq. 4 or Prop. 5.1 misstated?
>
> There is a small typo in Eq. 4; it should be $A_T = 0$, not $A_0 = 0$. Prop. 5.1 correctly bounds the probability that the terminal sample lands near the midpoint. The proof in Sec. H.6 is unaffected. We thank the reviewer for catching this and will fix it.
>
> We again thank the reviewer and are happy to clarify further if any questions remain.
>
> ### References:
> [1] Understanding Hallucinations in Diffusion Models through Mode Interpolation. NeurIPS’24.
>
> [2] High-Resolution Image Synthesis with Latent Diffusion Models. CVPR’22.

---

> > ### Author Rebuttal · Reviewer_WAef · 2026-04-04
> >
> > The authors have addressed my concerns. I remain my positive score.

---

### Official Review · Reviewer_nk7x · 2026-03-12

**Soundness:** 1
**Presentation:** 3
**Significance:** 1
**Originality:** 2
**Overall Recommendation:** 2
**Confidence:** 4

**Summary:**

The paper attempts to formalize "hallucination" for mixtures of Gaussians in terms of getting stuck in segments connecting modes of the mixtures during sampling. This is done in the context of diffusion models, comparing stochastic sampling methods that follow a reverse SDE path with deterministic ones that follow an ODE path. They conclude that deterministic/ODE approaches are prone to hallucination.

**Compliance With Llm Reviewing Policy:**

Affirmed.

**Final Justification:**

I appreciate the authors' response. Regarding the overlap in high dimensions, please note that other modes put mass in space and, depending on their distance compared to the noise level, the modes "interact" with each other. This can lead to quite complicated interactions (see Figs. 9 and 10 in *Neural Empirical Bayes* [1]). This is a fundamental concern unless the proposed framework is only for problems with two mixtures, which is extremely limiting.

Therefore, my main objection to the method stands. I encourage the authors to revisit their approach.

Regarding $W_2(\mathcal{L}(X), \mathcal{L}(\hat{X}))$, which I maintain to be the proper/rigorous hallucination metric, one can also study the dynamics of "how hallucinations emerge" by examining $W_2(\mathcal{L}(X), \mathcal{L}(X_t))$ over the whole sampling path for ODE- vs. SDE-based methods.

**Key Questions For Authors:**

Minor comments:

 - For readability, I suggest changing $\varpi$ (denoting the dimension) to $d$, which is the standard notation. I noticed $d_\perp$ was used in one place to denote distance. Perhaps you can change it to $z_\perp$ or $h_\perp$?
- In several places, the term "score field" is used. I suggest changing it to "score function", which is the standard terminology.

**Limitations:**

yes

**Strengths And Weaknesses:**

I find the paper generally well-written and enjoyable to read. Unfortunately, this is the only strength I can point out at the moment.

I have serious reservations about the mathematical formulation and the methodology in this paper. The core idea of regions connecting two modes of a mixture of Gaussians is displayed in Fig. 2. This is a low-dimensional view which does not match the nature of the Gaussian distribution in high dimensions, where the probability mass is not concentrated at the modes but away from them (very far away, in fact, as the dimension increases). This is a known result; see, e.g., the book *High-Dimensional Probability* by Vershynin. In the context of mixtures of Gaussians, this concentration of measure leads to a very different mental picture than the one in Fig. 2 (see, e.g., Fig. 2 in *Neural Empirical Bayes* by Saremi & Hyvärinen).

One can also formally approach hallucination in terms of the Wasserstein distance between two probability distributions $W_2(\mathcal{L}(X), \mathcal{L}(\hat{X}))$, where $\mathcal{L}(X)$ is the data distribution and $\mathcal{L}(\hat{X})$ is the law of samples generated by the generative model. There is no hallucination if this distance is zero. This is the clearest definition in my opinion, and there is now a large body of work studying this distance for SDE-based and ODE-based samplers for diffusion models, all of which go beyond mixtures of Gaussians that have been the focus of this paper.

Along these lines, the idea of adding noise during reverse ODE sample generation has already been proposed and studied in *Restart Sampling for Improving Generative Processes* by Xu et al. This was based on the rates obtained for ODE and SDE samplers (see Theorem 1 in the paper). This is very much related to the contribution 3 that is highlighted by the authors of the paper under review: "Empirically, we invalidate that DDIM hallucinations arise due to step skipping and demonstrate that adding a few DDPM steps after DDIM converges near the midpoint neighborhood can help the trajectory escape, lowering hallucination rate."

---

> ### Author Rebuttal · Authors · 2026-03-31
>
> We thank the reviewer nk7x for their compliments on the writing and readability. We address their questions below:
>
> ### > **Q1**: Does mode interpolation remain meaningful under high-dimensional concentration?
>
> Yes. Asm. 4.4 can be trivially modified by multiplying with $\varpi$ on the RHS. Under the adjusted Asm. 4.4, we have $\frac{\ell_t}{2\sqrt{\varpi}} > \sigma_t$. In this case, per Fig. 2a in [1], high-dimensional regions of high probability mass **do not overlap** between modes of the Gaussian mixture $p_t$ after $\tau_2$. Critically, this adjustment **does not alter our theoretical results**.
>
> We verify the adjusted Asm. 4.4 across dimensions, finding that we can still have reasonably sized $\tau_2$ for large values of $\kappa$. Please view the results [here](https://anonymous.4open.science/r/ddpm_ddim_rebuttal_figs_anon-373C/tau2_v_kappa_varpi.pdf).
>
> Thank you for raising this important topic; we will update Asm 4.4 and the experimental results in the revised version.
>
> ### > **Q2**: Is mode interpolation a key source of hallucinations?
>
> Yes. Note that mode interpolation hallucinations have been empirically reported and studied in multiple papers; please check Aithal et al. and Triaridis et al. from our paper. We emphasize that our work focuses on **identifying a concrete mechanism** behind **differences in hallucination rates** between DDIM and DDPM, and we only use mode interpolations as a testbed to study this.
>
> ### > **Q3**: Can the Wasserstein distance explain the mechanism of mode interpolation hallucinations?
>
> No. Studying the Wasserstein distance highlights the differences in the *terminal distributions* (${\mathcal{L}}(\hat{X})$) of DDIM and DDPM. This does not provide a clear picture of how the reverse process evolves over time and *how hallucinations emerge*. To make this mathematically precise, consider the bounds on the Wasserstein distance for ODE and SDE samplers in recent work [2]. As stated on pg. 18 of [2], the ODE is bounded by:
>
> $$W_2\\bigl(\mathcal{L}(X), {\mathcal{L}}(\hat{X})\bigr) \lesssim \frac{1}{T+\tau}+ h + \int_0^T \sqrt{\frac{\tau}{t+\tau}}\ \varepsilon_{\text{score}}(t)dt$$
>
> and the SDE is bounded by:
>
> $$W_2(\mathcal{L}(X), {\mathcal{L}}(\hat{X}) \lesssim \frac{1}{(T+\tau)^{3/2}} + \sqrt{h} + \int_0^T \frac{\tau}{t+\tau}\ \varepsilon_{\text{score}}(t)dt$$
>
> However, these bounds do not directly isolate the mechanism which causes hallucinations to arise more often in DDPM than DDIM, since they only consider the terminal distribution. While these bounds highlight that discretization and score error are handled differently by the ODE and SDE, they do not formalize how hallucinations *develop* during the reverse process. We believe this understanding is crucial to mitigate them.
>
> Here, $T$ is the same as in our paper; $\tau$ is Gaussian smoothing of the data; $h$ is the discretization error incurred by the sampler and $\varepsilon_{\text{score}}(t)$ is the score error.
>
> ### > **Q4**: How does this work differ from [3]?
>
> Thank you for providing the reference. This actually strengthens our case, since our paper’s core contribution is *not* hybrid sampling but rather a theoretical analysis of the reverse dynamics. In particular, we pinpoint a key distinction between DDIM and DDPM in a midpoint region between modes. We include the hybrid sampler of adding a few DDPM steps in the midpoint neighborhood only to support our theoretical results in Prop. 4.7 and 5.1. That is, we **do not aim to propose an entirely new sampler**. In addition, our work adds a new perspective on why DDPM has a lower hallucination rate than DDIM aside from score error contraction, which is not covered or implied in [3]. We will revise the introduction and related work to make this distinction explicit.
>
> ### > **Q5**: What does Fig. 2 represent?
>
> Fig. 2 is a high-level schematic used to describe key objects in our theory like $\mathrm{Tube}_{t,\varepsilon}^{(i,j)}$. $\varepsilon$ arises because of the reduction from $N$ modes to modes $i$ and $j$ per Asm. 4.1. Crucially, $\varepsilon$ is *not* intended to describe the boundary between true and interpolated samples: in our experiments, we use a $\mathcal{O}(\sigma \sqrt{d})$ boundary for this in line with concentration of measure [1].
>
>
> We again thank the reviewer and are happy to clarify further if any questions remain.
>
> ### References:
> [1] Neural Empirical Bayes. JMLR’19.
>
> [2] Convergence of Deterministic and Stochastic Diffusion-Model Samplers: A Simple Analysis in Wasserstein Distance. ArXiV’25.
>
> [3] Restart Sampling for Improving Generative Processes. NeurIPS’23.

---

> > ### Author Rebuttal · Reviewer_nk7x · 2026-04-05
> >
> > I appreciate the authors' response. Regarding the overlap in high dimensions, please note that other modes put mass in space and, depending on their distance compared to the noise level, the modes "interact" with each other. This can lead to quite complicated interactions (see Figs. 9 and 10 in *Neural Empirical Bayes* [1]). This is a fundamental concern unless the proposed framework is only for problems with two mixtures, which is extremely limiting.
> >
> > Therefore, my main objection to the method stands. I encourage the authors to revisit their approach.
> >
> > Regarding $W_2(\mathcal{L}(X), \mathcal{L}(\hat{X}))$, which I maintain to be the proper/rigorous hallucination metric, one can also study the dynamics of "how hallucinations emerge" by examining $W_2(\mathcal{L}(X), \mathcal{L}(X_t))$ over the whole sampling path for ODE- vs. SDE-based methods.

---

> > > ### Author Response · Authors · 2026-04-06
> > >
> > > We thank the reviewer for their response. We **strongly believe** that a) our analysis does hold under complex interactions and b) our analysis is more refined than the Wasserstein analysis.
> > >
> > > Firstly, we respectfully disagree that the complex interactions between the $N$ modes is a fundamental concern for our analysis. Our analysis is for $N$ modes, and Asm. 4.1 is introduced to ensure that the rest $N - 2$ modes are not overlapping with a pair of modes $i, j$. Then, Theorem 4.2 bounds the influence of the remaining $N - 2$ modes by a negligible $\varepsilon$. Thus, once Asm. 4.1 holds, the many-mode interactions raised by the reviewer are not a missing part of the analysis; they are already shown to be negligible after $\tau_1$. We validate Asm. 4.1 across dimensions in Fig. E.7. Interactions between the regions of high probability mass of modes $i$ and $j$, which are relevant for the dynamics along the segment, are then handled by Asm. 4.4 as discussed in Q1 above.
> > >
> > > These separation assumptions (Asm. 4.1, 4.4) are not limitations unique to our work, but are natural for any rigorous analysis of mode interpolation hallucinations as studied in [1]; if regions of high probability mass overlap, then there are no mode interpolations since all samples are in these valid regions. We note that other papers include assumptions on well-separatedness of Gaussian mixture components (e.g., Asm. 14 in [2]) as well.
> > >
> > > Secondly, we also respectfully disagree with the time-dependent Wasserstein perspective. **There is no such work or derivation on mode interpolation with Wasserstein distance**. If the reviewer means comparing separate bounds on $W_2(\mathcal{L}(X), \mathcal{L}(\hat{X}_t^{\mathrm{ODE}}))$ and $W_2(\mathcal{L}(X), \mathcal{L}(\hat{X}_t^{\mathrm{SDE}}))$, such bounds measure **global, distribution-level error** of each sampler to the target law. This is complementary, but does not isolate the **local, sample-level** event of the midpoint neighborhood studied in our paper. Our contribution is not merely that one sampler is globally closer to the target in Wasserstein distance but rather **where in state space** and at **what phase of the reverse process** hallucinations arise in order to explain the DDIM and DDPM hallucination rate gap. This also holds for a study of  $W_2(\mathcal{L}(\hat{X}_t^{\mathrm{ODE}}), \mathcal{L}(\hat{X}_t^{\mathrm{SDE}}))$ [3], which does not clearly isolate this mechanism either.
> > >
> > > Thirdly, since the reviewer mentions repeatedly on the utility of Gaussian mixtures, let us emphasize to the reviewer that Gaussian mixtures are a key testbed to theoretically analyze diffusion models across modern works [4, 5]. Furthermore, mode interpolations are a clear and concrete hallucination definition under Gaussian mixtures at the sample level, which is critical to delineate hallucinations versus true samples and analyze them theoretically.
> > >
> > > To sum up the main point, **there is no such "time-dependent" analysis from the Wasserstein perspective and we do not believe that it should influence the validity of our work**. We are thankful to the reviewer for improving our work (Asm. 4.4) and questioning the Wasserstein perspective. We will correct the assumption and also provide a related paragraph about the Wassterstein perspective in the revised version.
> > >
> > > Lastly, we recommend the reviewer to have a look at the following related works [4, 5] for theoretical analyses of diffusion models under Gaussian mixtures.
> > >
> > > References:
> > >
> > > [1] Understanding Hallucinations in Diffusion Models through Mode Interpolation. NeurIPS’24.
> > >
> > > [2] Learning Mixtures of Gaussians Using the DDPM Objective. NeurIPS’23.
> > >
> > > [3] Closing the ODE-SDE Gap in Score-Based Diffusion Models Through the Fokker-Planck Equation. Philosophical Transactions of the Royal Society A’25.
> > >
> > > [4] On the Edge of Memorization in Diffusion Models. NeurIPS’25.
> > >
> > > [5] Generalization of Diffusion Models Arises with a Balanced Representation Space. ICLR’26.

---

### Official Review · Reviewer_bF8a · 2026-03-13

**Soundness:** 3
**Presentation:** 3
**Significance:** 3
**Originality:** 3
**Overall Recommendation:** 5
**Confidence:** 4

**Summary:**

This work addresses an important and interesting question to theoretically understand the difference between DDPM and DDIM -- the mainstream popular algorithms in generating the new samples in the framework of score-based diffusion.  The angle to compare is from the so-called *hallucination* -- characterized by mode interpolation between modes in the data (Gaussian mixture model, specifically here).
+ Ideally, the generated samples should center around the mean of each mode with the right weight. But hallucination occurs if they get stuck in the midpoint of two neighboring modes, instead of converging to the modes' centers.
+ The difference of the DDPM and DDIM is Eqn(2) and (3) -- whether the diffusion coefficient is the orignal schedule $\beta_t$ in the forward training process, or simply zero.

This investigation of the practical effect of the diffusion coefficient in the reserved SDE for generating has attracted lots of works. Most of them focus on the emperical investigations only. The first theoretic work to theoretically study the impact of the diffucision coefficient on the data generatioin quality is perhas by Cao etal 2023  "Exploring the Optimal Choice for Generative Processes in Diffusion Models: Ordinary vs Stochastic Differential Equations" NeuRIPS. But the quality is measure by the KL divergence under the perturbation of the score function.

This work focused on the unique *hallucination* effect and provides the dynamical analysis (in contrast to looking at the distribution) for the *dynamics* near the line-segment of two centers in the neighboring modes, and demonstrated the 'saddle-type' trap near the midpoint for the (determinstic )  DDIM ODE, while the more stochastic DDIM with non-zero diffusion $\beta_t$ has a small probability of such trapping. This conclusion of 'self-correction' effect of noise in the diffusion coefficient for the revesed dynamics align with the conclusion of the above  Cao's paper in 2023.
The numerical examples and tests look  supportive and illuminating.

But to compare the difference between DDPM and DDIM surely involves a lot of other factors to control. The authors here assume that the score function -- certianly the same shared by DDPM and DDIM -- is **exact**.   This immediately raises a puzzling question that the marginal distributions at any time $t$ for the DDPM and DDIM, $p_t$,  are actually the same!  Then in theory,  there should be no hallucination since the distributions are the same.  Some clarifications are expected to shed light on the connection between dynamics and the distribution in terms of characterizing the hallucination.

**Compliance With Llm Reviewing Policy:**

Affirmed.

**Ethical Review Concerns:**

N.A.

**Final Justification:**

The new comments and explanation showed the justification and signifance of adopting the conditonal dynamical analysis used in this work.

**Key Questions For Authors:**

1. Please resolve my above puzze between the dynamical system and the probability curve equivalence,  under the exact score assumption (Figure 1a) since the pdfs  $p^{DDPM}(t)= p^{DDIM}(t)$ are the same. Both conceptuall and numerically, better theoreticaly too.

2. Use more rigorous experiments to exclude the othe possible reason for halluciation  from score error, the initial distribution, the  truncation of total time interval $T$.

## Minors
1.  Is it should be $A_{T}=0$ in Eq(4).
2.  What is the meaning of the shade area in Figure 4. ? The confidence interval ?

**Limitations:**

The authors adequately discussed the technical limitations.

**Strengths And Weaknesses:**

## Strength

1. Rigorous analysis for the dynamics with the focus on the hallucination for the Gaussian mixture model.
2. The presentation is overally fine to follow.  The figures and diagrams are instructive.
3. The  result is of interests and helpful to deepen the understanding the various dynamics for the generating process.

## Weakness
1.  The *exact score function* Assumption means the pdf $p_t$ generated with different dynamics (DDIM, DDPM) are the same:  $p^{DDPM}(t)= p^{DDIM}(t)$ since both starts with the same Gaussian distribution.  In addition, the time-discretization for these $p_t$ seems also to be excluded by Figure 3.   Then the hallucination illustrated by the Figure 1(a) is difficult to undertstand, since the two distributions are the same.

2.  To follow the above 1,  I checked Appendix F.1 and see the score is in fact training by the one-hidden-layer neural network, which is numerical. Why not use the analytical score function from the known Gaussian mixture pdf  $p_t$ ( You have shown it in the end of Section 2)?
How to make sure your MLP score is exact or at least sufficiently accurate ? In addition, there are many different score-matching ways to train this  MLP score, for example, different weigtht $w_t$  to control the score errors. Did you test your score training  with such different methods?


3. The dynamical perspective is quite unique and wonderful here. But the time-reverse principle in the score-based diffusion models is to have a curve in the distribution space which is realized by (an infinite number of ) dynamics (by tuning the diffusion coefficient), some of which are not stable or strange. But it should only appear when the system is *perturbed*: like the Lyapunov exponent is about the response to the perturbation.  How to still identify  the hallucination without any perturbation here ?

---

> ### Author Rebuttal · Authors · 2026-03-31
>
> We thank the reviewer bF8a for their insightful feedback. We appreciate their acknowledgement of our work’s importance in theoretically understanding the differences between DDIM and DDPM through the dynamical perspective. Below, we address the reviewer’s questions:
>
> ### > **Q1**: Since the DDIM and DDPM marginals coincide under the exact score, do the theoretical results of the paper help understand the differences in their hallucination rates?
>
> Yes, they do. Line 107 already verifies that the marginals $p_t$ coincide; note though that the *trajectory dynamics* $\mathbf{x}_t$ are different for DDIM/DDPM, since the ODE/SDE differ. We assume the exact score as regularity, studying the dynamics in the infinite sample limit. In this limit, we show that $\mathbf{x}_t$ under the ODE can get stuck (Prop. 4.7) while $\mathbf{x}_t$ under the SDE can escape this trapped region (Prop. 5.1). We study this limit in our main paper because it provides the cleanest presentation of this vulnerability.
>
> Our experiments then demonstrate that **the trapping region becomes useful and predictive in the realistic learned score scenario**. In particular, we find that most (~89%) DDIM interpolations are in a neighborhood of radius $\vartheta = 0.25\ell_t$ with $\tau_3$ = 11 remaining. Then, our Fig. 5 shows that DDIM trajectories get stuck in this case while DDPM ones escape, as predicted by our theory. Additionally, we provide a preliminary extension of our theoretical results to the learned score setting in Appendix C, demonstrating that for bounded score error DDPM noise functions in a similar way (Prop. C.6). Thank you for raising this important point.
>
> ### > **Q2**: Can you use the exact score in the experiments?
>
> Not for general distributions. One usually does not have access to the exact score but rather the *learned score*. We thus design our experiments (Sec. 6) to mimic this more realistic learned score setting and show that our theory is predictive in this case (see Q1 above).
>
> Additionally, note that Prop. 4.7 and 5.1 hold empirically in the exact score case. Please view these results [here](https://anonymous.4open.science/r/ddpm_ddim_rebuttal_figs_anon-373C/ddim_overlay_exact_score.pdf).
>
> We will revise the manuscript to make the distinction between the exact score in our theory and the learned score in our experiments explicit.
>
> ### > **Q3**: Is the learned score sufficiently accurate?
>
> Yes; we use a held-out test set of 1M samples across the reverse-time range and find low test error. Also, note that the DDIM/DDPM comparison in Fig. 1 uses the same pretrained score network for both samplers; thus the observed gap in hallucination rates is not explained by using different learned score models.
>
> ### > **Q4**: What does the shaded area in Fig. 4 mean?
>
> The 95% confidence interval of the distance to the nearest $i, j$-mode segment, $d_\perp(\mathbf{x_t}, L_{t,\varepsilon}^{(i,j)})$ across 100k trajectories sampled with DDIM (Fig. 4a) or DDPM (Fig. 4b).
>
> ### > **Q5**: Is there a still a difference in DDIM and DDPM hallucinations across hyperparameters e.g. $\sigma$ and $T$?
>
> Yes. We provide an ablation on the hallucination rates for DDIM and DDPM across $T$, $\sigma$, and $\ell$. The hallucination rate gap stays stable for $T$ and $\ell$. It decreases as $\sigma$ increases, since regions of high probability mass begin to overlap between modes. Please view these results [here](https://anonymous.4open.science/r/ddpm_ddim_rebuttal_figs_anon-373C/ablations_halluc_rate.pdf).
>
> We once again thank the reviewer and are happy to clarify further if any questions remain.

---

> > ### Author Rebuttal · Reviewer_bF8a · 2026-04-03
> >
> > I appreciate if  the author could further  clarify the first point of  Weakness I pointed out before.
> >
> > let me take an analogy. Suppose to generate the standard 2D Gaussian by two dynamics:  direct $X$ and $RT^{-1}RTX$  where $R$ is an any rotation and $T$ is translation  -- surely the dynamics are different.  What is the numerical effect -- you will never tell the difference like in Figure 1(a) . Even the variance is the same (since the empirical distribution has the uncertainty related to the variance  -- which can explain the deviation of the realized finite number of samples), right ?
> >
> > Dynamics  can place any initial state in the middle of the generative process  (i.e., near the line segment), so the dynamical analysis surely assume the initial is in the desired region as an implicit condition to study the fate of the trajectory, -- but the probability of getting that initial state for the dynamics in concerns would be different too,  which justifiy the overall effect then should be that the final distirbution $p_t$  remains the same despite of the dynamics.
> >
> > I guess  the dynamics analsyis here is surely of some merit of its own. But please help address this logic gap.

---

> > > ### Author Response · Authors · 2026-04-06
> > >
> > > We honestly appreciate the rigor from the Rev bF8a on their response. We agree with your analogy, but our motivation to move to the learned score makes the difference. Concretely, in the example mentioned by the reviewer, even though the trajectory dynamics differ, the marginals are the same. Thus, there should be no hallucination rate difference.  Under the *exact score*, this carries over to DDIM and DDPM. To make this mathematically precise, we have $p_t^{\mathrm{DDIM, exact}} = p_t^{\mathrm{DDPM, exact}}$. Let $H :=$ {mode interpolation occurs}. Then, we have $\Pr^{\mathrm{DDIM, exact}}(H) = \Pr^{\mathrm{DDPM, exact}}(H)$.
> > >
> > > However, **we are interested in explaining the observed differences in hallucination rates in the realistic, practical learned score setting**: $\Pr^{\mathrm{DDIM, learned}}(H) \gg \Pr^{\mathrm{DDPM, learned}}(H)$ (Fig. 1a).
> > >
> > > To bridge this gap, we must look beyond unconditional marginals and examine the *conditional trajectory dynamics* $p(\mathbf{x}\_0 | \mathbf{x}\_{\tau_3})$, which differ entirely. We find that **we can extract a difference in DDIM (Prop. 4.7) and DDPM (Prop. 5.1) under the exact score which uncovers the trapping mechanism driving $\Pr^{\mathrm{DDIM, learned}}(H) \gg \Pr^{\mathrm{DDPM, learned}}(H)$**.
> > >
> > > To make this mathematically concrete, let $M_{\vartheta, \tau_3}: =$ {trajectory enters the neighborhood of radius $\vartheta$ around the midpoint during the final $\tau_3$ steps}. Then:
> > >
> > >
> > > $$\Pr(H) = \Pr(H|M_{\vartheta, \tau_3}) \Pr(M_{\vartheta, \tau_3}) + \Pr(H|M_{\vartheta, \tau_3}^c) \Pr(M_{\vartheta, \tau_3}^c)$$
> > >
> > > The contributions of our exact score theory in Props. 4.7, 5.1 are twofold:
> > > 1. The midpoint neighborhood and $M_{\vartheta, \tau_3}$ are driving the differences in hallucination rates between DDIM and DDPM, i.e. $\Pr(H|M_{\vartheta, \tau_3}^c)$ is small and the first term dominates so that $\Pr(H) \approx \Pr(H|M_{\vartheta, \tau_3}) \Pr(M_{\vartheta, \tau_3})$. This is done in Prop. 4.7.
> > >
> > > 2. Demonstrate that $\Pr^{\mathrm{DDIM, exact}}(H|M_{\vartheta, \tau_3}) \gg  \Pr^{\mathrm{DDPM, exact}}(H|M_{\vartheta, \tau_3})$. This is done in Prop. 5.1.
> > >
> > > These results arise due to the differences in (conditional) dynamics, even though the marginals are the same under the exact score. The exact score assumption in our theory allows us to demonstrate this cleanly. Still, $\Pr^{\mathrm{DDIM, exact}}(M_{\vartheta, \tau_3}) = \Pr^{\mathrm{DDPM, exact}}(M_{\vartheta, \tau_3}) = 0$.
> > >
> > > But, in the practical learned score setting where $\Pr^{\mathrm{learned}}(M_{\vartheta, \tau_3}) \neq 0$ for both samplers, the fundamental difference $\Pr^{\mathrm{DDIM, learned}}(H|M_{\vartheta, \tau_3}) \gg \Pr^{\mathrm{DDPM, learned}}(H|M_{\vartheta, \tau_3})$, which persists in the learned score case but is uncovered via our exact score analysis, helps explain $\Pr^{\mathrm{DDIM, learned}}(H) \gg \Pr^{\mathrm{DDPM, learned}}(H)$. Empirically, we find that the first term remains dominant while $\Pr^{\mathrm{DDIM, learned}}(H|M_{\vartheta, \tau_3}) \gg  \Pr^{\mathrm{DDPM, learned}}(H|M_{\vartheta, \tau_3})$. Please view our results [here](https://anonymous.4open.science/r/ddpm_ddim_rebuttal_figs_anon-373C/halluc_prob_table_learned_score.pdf).
> > >
> > > The bound in Prop. 5.1 is also proved for the learned score setting in Prop. C.6.
> > >
> > > **To summarize**: the marginals are indeed identical under the exact score.  Crucially,  we leverage the exact score only to understand *how DDIM and DDPM behave differently along the line segment joining two modes*, since this is precisely where mode interpolation occurs. We study the exact score in our theoretical analysis because it lets us **cleanly** expose how DDIM and DDPM differ along the line segment, specifically near the midpoint (Props. 4.7, 5.1). This highlights $M_{\vartheta, \tau_3}$ as the event driving hallucination differences. This then helps explain the observed differences in hallucination rates in the realistic, practical *learned score setting*  (Fig. 1a), where the marginals are perturbed and need not be equal.
> > >
> > > We will clarify the exact motivation in our camera-ready manuscript: the hallucination rate gap is established on the learned score. We will also clarify that the learned score is essential for studying realistic hallucinations.
> > >
> > > Once again, we are thankful to the reviewer for holding our manuscript to a high standard.

---

### Decision · Program_Chairs · 2026-04-30

**Decision:**

Accept (regular)

**Comment:**

This paper studies why DDPM reduces hallucination compared to DDIM in a Gaussian mixture setting. The authors argue that deterministic DDIM trajectories can get trapped near a midpoint between two modes, while the stochastic noise in DDPM helps the dynamics escape and reach a true mode.

Reviewers agreed the question is interesting and the explanation is appealing. However, the presentation is the main weakness. The mechanism by which DDPM noise helps is not explained clearly enough. Some technical statements are hard to interpret, and the narrative is not always transparent. One reviewer remained unconvinced, which highlights the need for better exposition.

As a result, I recommend to accept the paper if there is room in the program, but if accepted the presentation must be substantially improved. The authors should better motivate the setting, clarify the intuition behind the main results, and explain the significance and limitations more clearly.